# Pantothenate biosynthesis is critical for chronic infection by the neurotropic parasite *Toxoplasma gondii*

Matteo Lunghi [1,5], Joachim Kloehn [1,5], Aarti Krishnan [1], Emmanuel Varesio [2,3], Oscar Vadas [1,4] & Dominique Soldati-Favre [1✉]

Coenzyme A (CoA) is an essential molecule acting in metabolism, post-translational modification, and regulation of gene expression. While all organisms synthesize CoA, many, including humans, are unable to produce its precursor, pantothenate. Intriguingly, like most plants, fungi and bacteria, parasites of the coccidian subgroup of Apicomplexa, including the human pathogen *Toxoplasma gondii*, possess all the enzymes required for de novo synthesis of pantothenate. Here, the importance of CoA and pantothenate biosynthesis for the acute and chronic stages of *T. gondii* infection is dissected through genetic, biochemical and metabolomic approaches, revealing that CoA synthesis is essential for *T. gondii* tachyzoites, due to the parasite's inability to salvage CoA or intermediates of the pathway. In contrast, pantothenate synthesis is only partially active in *T. gondii* tachyzoites, making the parasite reliant on its uptake. However, pantothenate synthesis is crucial for the establishment of chronic infection, offering a promising target for intervention against the persistent stage of *T. gondii*.

[1] Department of Microbiology and Molecular Medicine, University of Geneva, CMU, Rue Michel-Servet 1, 1211 Geneva, Switzerland. [2] Institute of Pharmaceutical Sciences of Western Switzerland (ISPSO), University of Geneva, CMU, Rue Michel-Servet 1, 1211 Geneva, Switzerland. [3] Mass Spectrometry Core Facility (MZ 2.0), University of Geneva, 1211 Geneva, Switzerland. [4] Protein and peptide purification platform, University of Geneva, CMU, Rue Michel-Servet 1, 1211 Geneva, Switzerland. [5]These authors contributed equally: Matteo Lunghi, Joachim Kloehn. ✉email: Dominique.Soldati-Favre@unige.ch

The Apicomplexa phylum comprises thousands of parasites that infect humans and animals, such as *Plasmodium spp.*, the causative agents of malaria, and *Cryptosporidium spp.* responsible for diarrheal disease. *Toxoplasma gondii* is the most ubiquitous member of the phylum, infecting all warm-blooded animals including an estimated third of the human population[1]. Infection typically occurs through the accidental intake of oocysts from contaminated food and water, or consumption of tissue cysts from infected meat. Primary infection during pregnancy can cause miscarriage or stillbirth following placental infection of the fetus[2]. During an effective immune response, the fast replicating tachyzoites are cleared, while some parasites convert into slow growing bradyzoites that persist within cysts, predominantly in the brain and muscle tissues, for the lifetime of the host[3]. This chronic infection is generally asymptomatic but poses a severe risk of toxoplasmosis recrudescence in case of immunosuppression[4,5]. Clinically available drugs are effective against tachyzoites, but fail to eradicate the encysted, quasi-quiescent bradyzoites.

As an obligate intracellular parasite, *T. gondii* and other apicomplexans rely on the uptake of essential nutrients from their host, as well as on the de novo synthesis of metabolites which cannot be sufficiently salvaged. The metabolic needs and capabilities of *T. gondii* bradyzoites are poorly characterized due to the technical challenges associated with studying this parasite state, particularly in its natural niche. The identification of salvaged metabolites or synthesis pathways that are essential for the establishment of chronic stage is a critical step towards the treatment of chronic toxoplasmosis.

Coenzyme A (CoA) is a ubiquitous and essential hub metabolite found in all organisms, acting in gene regulation, post-translational protein modification and several metabolic pathways, including the tricarboxylic acid (TCA) cycle as well as heme and fatty acid synthesis. In *T. gondii*, the most abundant CoA-derivative, acetyl-CoA, is produced by four different enzymes, each of which localizes to one of three metabolically active subcellular compartments: the mitochondrion, the cytosol/nucleus, and the apicoplast, a relict plastid organelle found in many apicomplexans[6–8]. Aside from acetyl-CoA, synthesis of other critical metabolites such as malonyl-CoA, succinyl-CoA and acyl-CoAs also depend on CoA availability. The synthesis of CoA involves five conserved enzymatic steps and uses pantothenate (vitamin B$_5$, Pan) as a precursor. Mining of the *T. gondii* genome confirmed the presence of a complete pathway for CoA biosynthesis, including the previously unannotated gene for the dephospho-CoA kinase (DPCK)[9,10]. The pathway is conserved in all apicomplexans (Fig. 1a, b, Supplementary Data 1). Intriguingly, unlike their human and animal hosts, *T. gondii* and other coccidians also possess the genes encoding enzymes to synthesize the CoA precursor Pan (Fig. 1a, b, Supplementary Data 1). Here, we scrutinized the parasite's ability to synthesize and/or salvage intermediates of the Pan/CoA pathway and the importance of several biosynthesis steps for the clinically relevant life cycle stages of *T. gondii*. We uncover that CoA synthesis is essential in *T. gondii*, while Pan synthesis is dispensable in tachyzoites due to an efficient salvage pathway. Remarkably, Pan synthesis becomes critical during chronic infection, with disruption of the pathway resulting in a marked reduction in cyst numbers. These findings are invaluable for the development of much needed drugs to treat acute and chronic toxoplasmosis.

## Results and discussion
### Initiation of CoA biosynthesis from Pan is essential for *Toxoplasma gondii* and relies on a heteromeric PanK-complex. In order to probe the importance of CoA synthesis in *T. gondii*, the

role of several putative enzymes of the pathway was characterized, namely, two pantothenate kinases (PanK1 + 2), the phospho-pantothenoylcysteine decarboxylase (PPCDC) and DPCK (Fig. 2a), catalyzing the first, an intermediate and the last step of CoA synthesis, respectively. The initial enzymatic step consists in the phosphorylation of Pan to form 4′-phosphopantothenate (PPan) and is catalyzed by PanKs[9,11]. *T. gondii*, as most apicomplexans, possesses two *PanK* genes: *PanK1* (TGME49_307770) and *PanK2* (TGME49_235478) (nomenclature based on sequence similarity and existing literature[11]). To examine the function of the *PanKs*, both proteins were C-terminally tagged with haemaglutinin (HA-) epitope and fused to a mini auxin degron-haemagglutinin-tag (mAID-HA, Supplementary Fig. 1a) in an RH *ku80-ko hxgprt-ko*::Tir1 strain (Tir1 parental) at the endogenous locus by CRISPR/Cas9 editing[12,13]. PanK1-mAID-HA and PanK2-mAID-HA migrated close to the predicted molecular weight (MW) of 144 kDa and 190 kDa, respectively, and were efficiently downregulated upon addition of auxin (indole-3-acetic acid, IAA)[14] as shown by western blot (Fig. 2b).

Both enzymes were localized by immunofluorescence assays (IFAs), with PanK1 and PanK2-mAID-HA tagged strains presenting a faint, dotty cytoplasmic staining (Fig. 2c). Addition of IAA to the culture medium over 24 h growth of the parasite resulted in a marked loss of the dotty signal in the two strains (Fig. 2c), confirming efficient downregulation of PanK1 and PanK2 as observed by western blot (Fig. 2b). Crucially, downregulation of either PanK1 or PanK2 over 24 h was accompanied by severe loss of parasite morphology, as seen by staining with actin (Fig. 2c) and GAP45, a marker of the parasite pellicle (Fig. 2d).

As expected, given the severe morphological defects after only 24 h of PanK1 or PanK2 downregulation, PanK1-mAID-HA and PanK2-mAID-HA parasites failed to form lysis plaques in host cell monolayers (human foreskin fibroblasts, HFFs) in the presence of IAA over 7 days (Fig. 2e), quantified in Fig. 2f. These findings reveal that PanK1 and PanK2 are individually essential for the parasite's lytic cycle.

The fitness conferring nature of PanK1 and PanK2 is consistent with data from a genome-wide fitness screen, which assigned a highly negative fitness score to both enzymes (Fig. 1a)[15] and led us to hypothesize that the enzymes are not functionally redundant but instead both enzymes may be required to form a functional PanK-complex. In *Escherichia coli*, PanKs have been shown to form homodimers[16], although heteromeric PanK complexes have not been reported until recently[17]. Co-immunoprecipitation experiments of dually tagged strains PanK1-mAID-HA/PanK2-Ty and PanK2-mAID-HA/PanK1-Ty revealed that PanK1 and PanK2 form a heteromeric complex, causing PanK1 to be pulled down with PanK2 and vice versa (Fig. 2g). These results are consistent with a recent study reporting that *P. falciparum* and *T. gondii* PanKs function in a heteromeric PanK complex[17]. Of relevance, IAA-induced destabilization of either *T. gondii* PanK1 or PanK2 did not lead to the degradation of the other PanK (Supplementary Fig. 1b). Given that both PanKs are required to form an active heteromeric complex, the following phenotypical characterization was carried out focusing on parasites depleted in PanK1 only.

To identify how quickly upon initiation of PanK-downregulation parasite fitness deteriorates, as judged by cell morphology, additional IFAs were performed on PanK1-mAID-HA parasites treated for 0, 8 and 12 h with IAA. The onset of cell deformation of varying types and severity was revealed, using a parasite pellicle marker, already 12 h after IAA-treatment. Parasites appeared bloated, likely due to failed cytokinesis (Fig. 2h3, for normal morphology of non-dividing and dividing parasites, see Fig. 2h1 and 2h2, respectively), were distributed

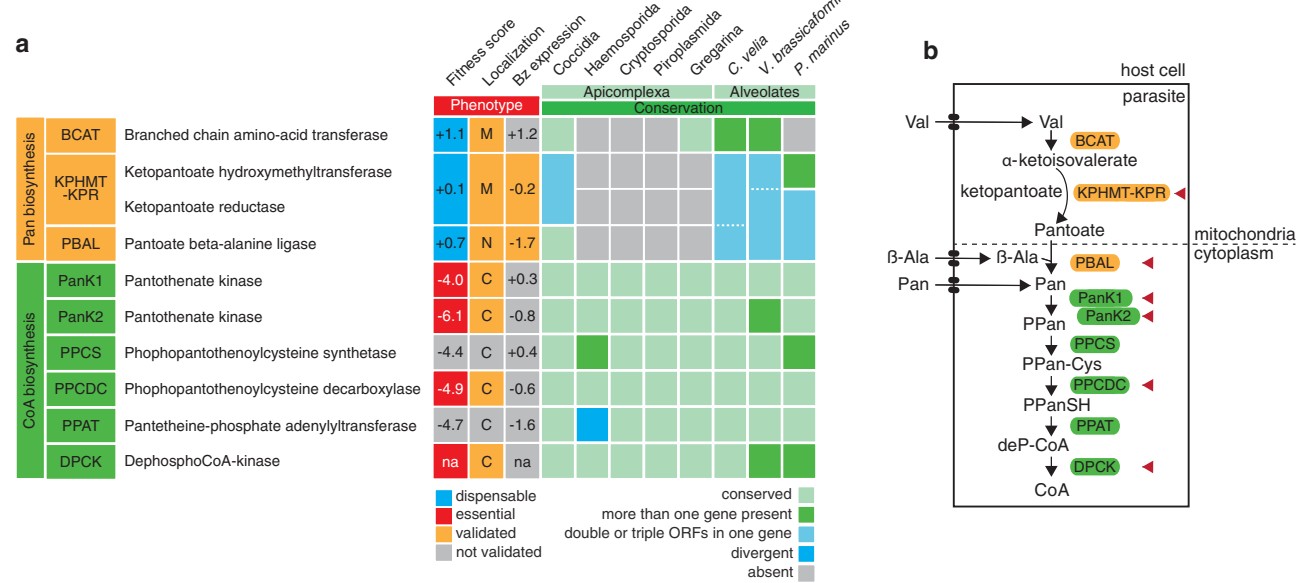

**Fig. 1 Pan and CoA biosynthesis pathways in *T. gondii*. a** Table representing the fitness cost, localization, and stage expression of enzymes for pantothenate and coenzyme A biosynthesis in *T. gondii*, and conservation of the genes in other alveolates. Fitness scores from a genome-wide fitness screen are given (positive values indicate dispensability, while negative values suggest essentiality)[15], together with dispensability/essentiality determined from this and previous studies (blue - dispensable, red - essential). Localizations, experimentally determined in this or previous studies, are given for all enzymes involved in pantothenate (Pan) or coenzyme A (CoA) biosynthesis (M mitochondrion, N nucleus, C cytosol), in gray when predicted but not experimentally validated. Log$_2$ fold change in RNA levels from an RNA-seq analysis[75] are provided (bradyzoite (Bz) expression levels relative to tachyzoites) and complemented by protein expression levels determined in this study (orange – validated experimentally). Conservation is determined by a BLAST search (e-value < E−04). Dotted lines represent the situation where single/fused genes are present, as well as one open reading frame (ORF) spanning all 3 genes. **b** Cartoon scheme of the Pan and CoA biosynthesis pathways. Valine (Val), β-Alanine (β-Ala) and Pan are likely salvaged through unknown transporters. Enzymes are in boxes. Red arrowheads highlight the enzymes under scrutiny in this manuscript. Abbreviations: PPan, 4'-phosphopantothenate; PPan-Cys, 4'-phospho-N-pantothenoyl cysteine; PPanSH, 4'-phosphopantetheine; deP-CoA, dephospho-CoA; CoA, coenzyme A.

abnormally in the vacuole (Fig. 2h4), or displayed a complete loss of inner membrane complex (IMC) integrity (Fig. 2h5).

Additionally, employing markers for the two key metabolic organelles, which harbor metabolic pathways that critically depend on CoA, the apicoplast and the mitochondrion, parasites displayed no aberrant organelle morphology after 8 h IAA (Fig. 2i, j). The absence of deformed parasites at 8 h IAA, while predominant at 24 h IAA, was quantified by an intracellular growth assay (Fig. 2k).

To examine whether the PanKs catalyze the expected reaction in CoA synthesis, targeted metabolomic analyses were performed, measuring the abundance of CoA and three of its precursors (Pan, 4'-phospho-*N*-pantothenoyl cysteine, PPan-Cys and dephospho-CoA, deP-CoA) in parasite extracts by liquid chromatography-mass spectrometry (LC-MS). Based on the above phenotyping, quantitative LC-MS analyses were performed after 8 h IAA treatment (Fig. 2l). Consistent with PanK1 catalyzing the phosphorylation of Pan, its downregulation led to a significant increase in its substrate, Pan. PPan-Cys was not detected in the parental line and the PanK1 depleted parasites. deP-CoA was significantly decreased in PanK1-mAID parasites upon addition of IAA compared to the parental Tir1 line +IAA but was similarly reduced in PanK1-mAID parasites −IAA. Instead, CoA was significantly (~2-fold) reduced specifically upon downregulation of PanK1 compared to its controls. The accumulation of its substrate, Pan, and reduction in the end-product, CoA, are consistent with the expected function of PanKs.

We conclude that *T. gondii* possesses two PanKs which function in a heteromeric complex that catalyzes the phosphorylation of Pan towards CoA synthesis. Obstruction of this step causes severe morphological defects and results in parasite death.

***T. gondii* cannot rely on the salvage of CoA or intermediates of the CoA synthesis pathway.** Following enzymatic activity of the PanKs, cysteine is added to PPan through the phospho-pantothenoylcysteine synthetase (PPCS), and the subsequent decarboxylation catalyzed by PPCDC leads to the generation of 4'-phosphopantetheine (PPanSH). Critically, the function of PPCDC can be bypassed by the promiscuous activity of some PanKs, which catalyze the phosphorylation of pantetheine (PanSH)[18,19]. In *Plasmodium yoelii* and *Plasmodium berghei* blood stages[20,21], this salvage pathway was shown to be active, bypassing the CoA synthesis enzymes PPCS and PPCDC (Fig. 3a). In contrast, direct PPanSH salvage has been proposed in *Drosophila* and mammalian cell culture models and could explain dispensability of both PanKs in *P. berghei*[19,22,23]. PPanSH in culture medium can be generated by CoA hydrolysis through serum hydrolases. To determine whether either salvage pathway can contribute to CoA synthesis in *T. gondii*, PPCDC (TGME49_242880), the enzyme upstream of the potential PanSH salvage, was characterized.

C-terminal mAID-HA tagged *PPCDC* was generated (Supplementary Fig. 1c) but could not be detected by western blot, whereas a C-terminal 3xTy-tagged fusion in the endogenous locus confirmed that PPCDC-Ty migrates close to the predicted molecular weight by western blot (34 kDa, Fig. 3b). By IFA, PPCDC-Ty showed a cytoplasmic and nuclear signal (Fig. 3c). The signal detected by IFA for PPCDC-mAID-HA was weak but sufficient to confirm its efficient downregulation following addition of IAA for 24 h (Fig. 3d). As described above for the PanKs, PPCDC downregulation led to a dramatic loss of cell morphology based on the actin staining (Fig. 3d) and pellicle marker, 24 h after IAA treatment (Fig. 3e). Consistent with this

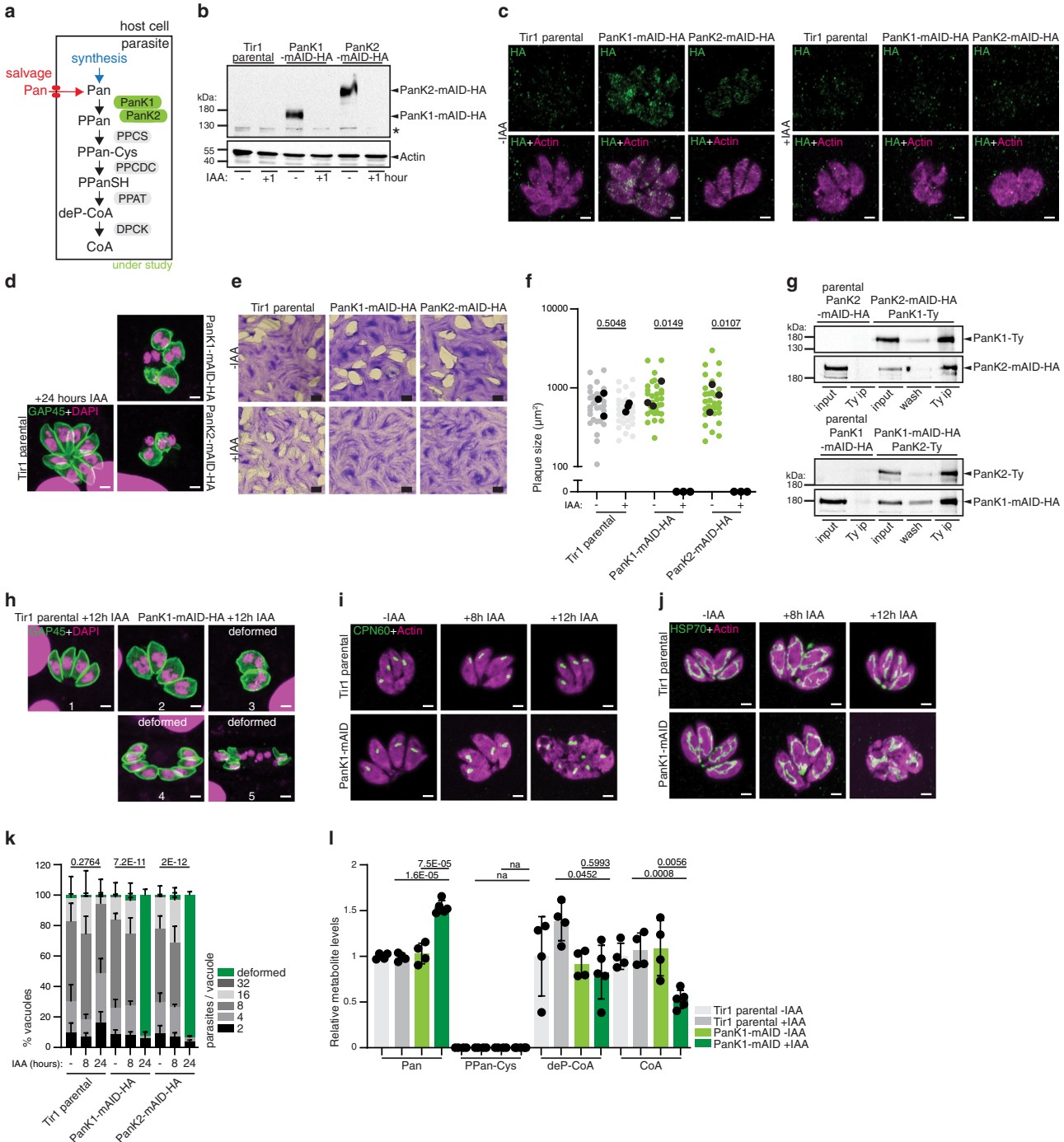

**Fig. 2 Initiation of CoA biosynthesis from Pan is essential for *T. gondii* and relies on a heteromeric PanK-complex. a** Scheme of the CoA biosynthesis pathway, highlighting the *T. gondii* pantothenate kinases (PanKs). See Fig. 1 for abbreviations of enzymes and metabolites. **b** Western blot of endogenous C-terminally mAID-HA tagged PanK1 and PanK2 (expected MW 144 kDa and 190 kDa, respectively) in presence of indole-3-acetic acid (IAA). Anti HA, anti actin as loading control ($n = 4$). **c** Immunofluorescence assay (IFA) of PanK1-mAID-HA and PanK2-mAID-HA. Antibodies: anti HA and anti actin marking the parasite cytoplasm ($n = 5$). **d** IFA of 24 h IAA treated PanK1- and PanK2-mAID-HA. Antibodies anti GAP45 for parasite pellicle, DAPI for nuclear staining ($n = 3$). **e** Plaque assays of PanK1-mAID-HA and PanK2-mAID-HA in the presence of IAA ($n = 4$). **f** Quantification of plaque size. Black dots present the average of each independent experiment. ($n = 3$, 2-sided *t*-test). **g** Western blot of co-immunoprecipitation (IP) assay of dually tagged PanK1-Ty/PanK2-mAID-HA (top, $n = 3$) and PanK1-mAID-HA/PanK2-Ty (bottom, $n = 4$). Antibodies for IP: anti-Ty; anti-HA and anti-Ty for revelation. **h** IFA of PanK1-mAID-HA with IAA for 12 h. Antibodies: anti GAP45 for parasite pellicle, DAPI for nuclear staining ($n = 3$). **i**, **j** IFAs of apicoplasts and mitochondria, respectively, at the indicated timepoints of IAA addition in PanK1-mAID-HA. Antibodies: anti actin for parasite cytoplasm, anti CPN60 for apicoplasts, anti HSP70 for mitochondria ($n = 3$). **k** Intracellular growth assay of the PanK1- and PanK2-mAID-HA. Deformed parasites were scored as indicated in panel h ($n = 3$-6, mean and SD, 2 tailed *t*-test on the "deformed" category). **l** LC-MS analysis of indicated metabolite abundances in PanK1 depleted parasites (8 h +IAA, $n = 4$-5, mean and SD, 2-sided Student's *t* test). Equal number of parasites were analyzed, and metabolite levels normalized to an internal standard ($^{13}C_6/^{15}N$-isoleucine) and quantified relative to Tir1 parental −IAA (abundance = 1). *n* number of independent biological replicates. Source data are provided as a Source Data file. White scale bar 2 μm, black scale bar 1 mm. Black asterisk indicates unspecific signal.

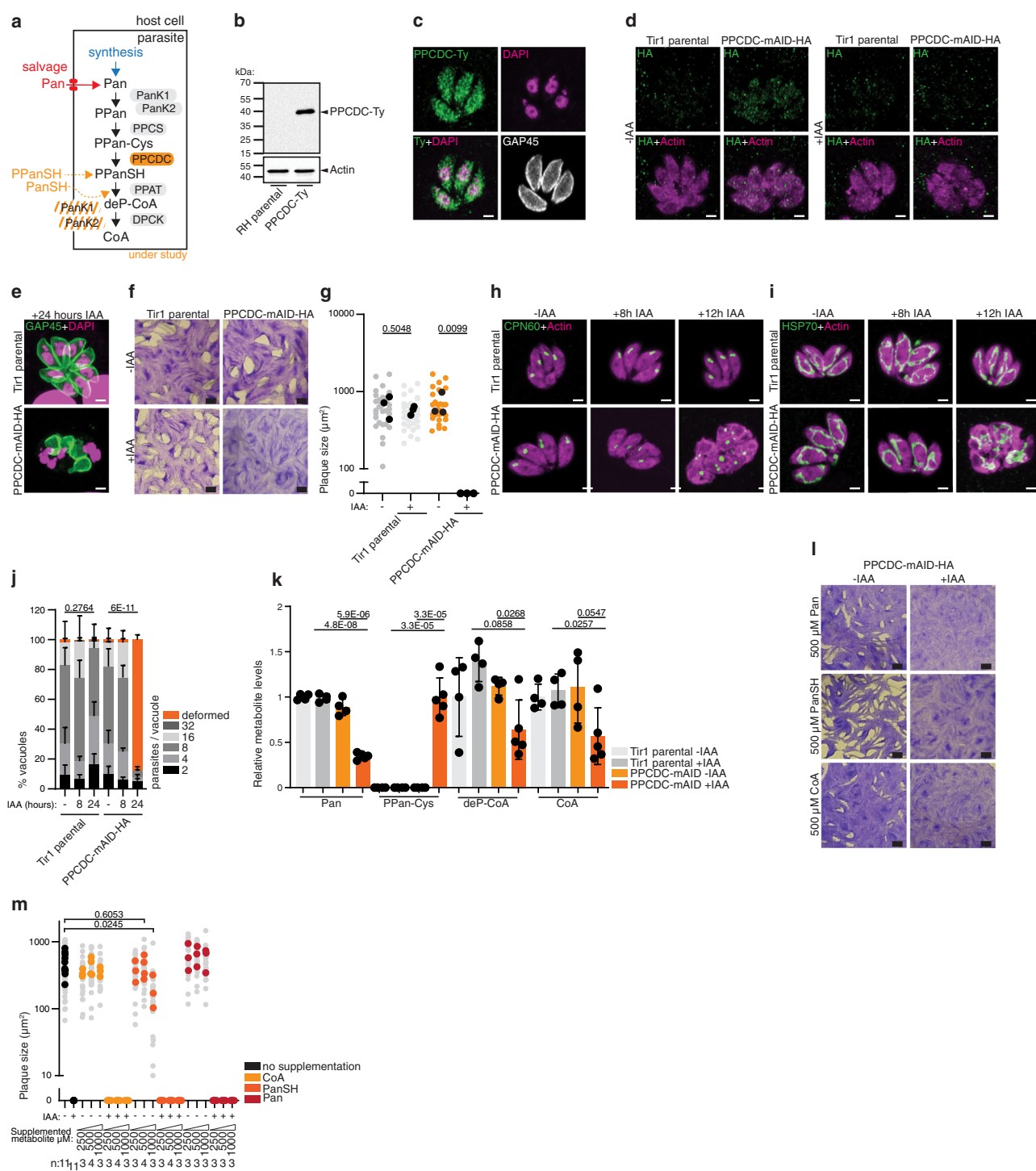

observed cell deformation, PPCDC was found to be essential for the lytic cycle of *T. gondii*, as PPCDC-mAID-HA parasites failed to form plaques in a HFF monolayer over 7 days in the presence of IAA (Fig. 3f, g). These findings indicate that salvage of PanSH or PPanSH does not contribute to CoA synthesis in *T. gondii* under standard culture conditions. As observed for PanK1, cell morphology, as well as the apicoplast and mitochondrion stained with anti-CPN60 and anti-heat shock protein HSP70, respectively, appeared intact and normal after 8 h of IAA-treatment, but the onset of deformation began at 12 h of IAA-treatment (Fig. 3h, i). Additionally, intracellular growth assays following 24 h of growth in absence of IAA or following 8- or 24-h of IAA-

treatment of PPCDC-mAID-HA parasites were performed. No deformation or growth defects were observed after 8 h of IAA treatment, while PPCDC-mAID-HA parasites were largely deformed after 24 h of IAA treatment (Fig. 3j).

To demonstrate that PPCDC catalyzes the expected decarboxylation of 4'-phospho-N-pantothenoylcysteine (PPanCys) participating in CoA synthesis, targeted metabolomic analyses were performed by LC-MS on parasite extracts collected 8 h after IAA-treatment. Consistent with the expected function of PPCDC, its substrate PPan-Cys accumulated but was not detected in control strains (Fig. 3k). Remarkably, Pan levels were significantly decreased, indicating that accumulating PPan-Cys, or the

**Fig. 3 *T. gondii* cannot rely on the salvage of CoA or intermediates of the CoA synthesis pathway. a** Scheme of the CoA biosynthesis pathway, highlighting putative salvage pathways of pantetheine (PanSH) and phosphopantetheine (PPanSH) and the *T. gondii* phosphopantothenoylcysteine decarboxylase (PPCDC). See Fig. 1 for abbreviations. **b** Western blot of endogenously tagged RH PPDCD-Ty. Antibodies anti Ty, anti actin as loading control ($n = 4$). **c** Immunofluorescence assay (IFA) of RH PPCDC-Ty parasites. Antibodies anti Ty, anti GAP45 as marker of parasite pellicle ($n = 3$). **d** IFA of PPCDC-mAID-HA. Antibodies anti HA, and anti actin marking the parasite cytoplasm ($n = 4$). **e** IFA of 24 h IAA treated PPCDC-mAID-HA. Antibodies anti GAP45 for parasite pellicle, DAPI for nuclear staining ($n = 3$). **f** Plaque assay of PPCDC-mAID-HA parasites in presence of IAA ($n = 4$). **g** Quantification of plaque size. Black dots present the average of each independent experiment. ($n = 3$, 2-sided *t*-test). **h, i** IFAs of apicoplasts and mitochondria, respectively, at the indicated timepoints of IAA addition in PPCDC-mAID-HA. Antibodies anti actin (cytoplasm), anti CPN60 (apicoplast), anti HSP70 (mitochondrion) ($n = 3$). **j** Intracellular growth assay of PPCDC-mAID-HA. Deformed parasites scored as indicated in Fig. 2h ($n = 3–6$, mean and SD, 2 tailed *t*-test on the "deformed" category). **k** LC-MS analysis of metabolite abundances in PPCDC depleted parasites (8 h +IAA, $n = 4–5$, mean and SD, 2-sided Student's *t* test). Metabolite levels normalized to internal standard ($^{13}C_6/^{15}N$-isoleucine) and quantified relative to Tir1 parental −IAA. **l** Plaque assay of PPCDC-mAID-HA parasites in presence of IAA, with the supplementation of 500 µM Pan, pantetheine (PanSH), or CoA ($n = 3$). **m** Quantification of plaque size. Colored dots represent the average of each independent experiment ($n$ indicated in the panel, 2-sided *t*-test). The Tir1 parental controls of Fig. 3d–k are shared with Figs. 2c–f, i–l. *n* number of independent biological replicates. Source data are provided as a Source Data file. White scale bar 2 µm, black scale bar 1 mm.

upstream intermediate PPan (not measured) may provide a feedback signal to reduce Pan acquisition. Crucially, the metabolites downstream of PPCDC, deP-CoA and CoA were significantly reduced upon PPCDC-downregulation compared to the control strains. Of note, the values for the Tir1 parental control correspond to those shown in Fig. 2m, as these samples were analyzed in a single batch. Lastly, we investigated whether exogenously supplemented CoA, or its precursors, could facilitate their salvage and overcome the essentiality of PPCDC. However, neither supplementation of Pan, nor PanSH or CoA (each provided at 250 µM, 500 µM and 1 mM) were able to rescue the parasite's inability to form lysis plaques upon downregulation of PPCDC (Fig. 3l, m, Supplementary Fig. 1d). At the highest concentration tested here, 1 mM, PanSH supplementation caused the formation of smaller plaques (Fig. 3m, Supplementary Fig. 1d), pointing towards toxicity of this intermediate at high concentrations. Thus, neither PanSH salvage pathway, nor direct salvage of CoA or PPanSH, following CoA hydrolysis, contribute (sufficiently) to the CoA pool of *T. gondii* even when provided in excess. Whether this is due to inefficient salvage by the host, the parasite, or less promiscuous PanKs compared to *P. yoelii* and *P. berghei*, remains unclear.

In conclusion, PPCDC is an essential enzyme in the CoA synthesis pathway of *T. gondii* that generates PPanSH, and its function cannot be bypassed by the salvage of downstream metabolites. Lack of PPCDC leads to accumulation of its substrate and a drop in CoA levels, which causes severe morphological defects and eventually parasite death.

**The final step of CoA biosynthesis is essential in *T. gondii* and relies on a membrane-bound cytoplasmic DPCK.** The gene coding for the final CoA synthesis enzyme, *DPCK* was recently identified upstream of the alanine dehydrogenase gene in the *T. gondii* genome (TGME49_315260)[9]. DPCK catalyzes the phosphorylation of dephospho-coenzymeA (deP-CoA) (Fig. 4a). The products of a second copy C-terminally myc-tagged DPCK under the constitutive tubulin promoter (cDPCK-myc) and the C-terminally spaghetti monster-myc[24] tagged DPCK (DPCK-SMmyc) at the endogenous locus migrated by western blot close to the expected sizes of 37 kDa and 72 kDa, respectively (Fig. 4b). In accordance with the presence of two predicted transmembrane domains, cDPCK-myc and DPCK-SMmyc behave as integral membrane proteins based on fractionation experiments (Fig. 4c, d). cDPCK-myc and DPCK-SMmyc were detected in the pellet fraction of parasite lysates upon sodium carbonate ($Na_2CO_3$) extraction, but in the soluble fraction upon extraction with the detergents Triton X-100 (Tx-100) or sodium dodecyl sulfate (SDS). In contrast to *P. falciparum* DPCK, which localizes to the

apicoplast[25], *Tg*DPCK revealed a dotty cytoplasmic localization by IFA (Fig. 4e), clearly distinct from the apicoplast (see Fig. 4j for comparison) and in concordance with the localization of all other tagged genes in the CoA synthesis pathway. Depletion of DPCK was efficiently achieved using the tet-repressive promoter system, in an RH *ku80-ko hxgprt-ko* (RH parental) in presence of anhydrotetracycline (ATc) as demonstrated by western blot (Fig. 4f, Supplementary Fig. 1e)[26]. Important for the interpretation of these results, stage specificity is not an issue with these minimal tet promoters[26,27]. Protein downregulation, using the promoter repression system, takes effect considerably slower than the rapid IAA-induced protein degradation system, with residual protein detected after 24 h (Fig. 4f). Depletion of DPCK over 48 h, resulted in dramatic loss of parasite morphology (Fig. 4g). Consistent with this, prolonged ATc-treatment of DPCK-iKD parasites prevented the formation of lysis plaques in a HFF monolayer over 7 days of growth (Fig. 4h), quantified in Fig. 4i. Critically, this defect in the lytic cycle was rescued by the expression of a second copy DPCK-myc (DPCK-iKD::cDPCK-myc, Fig. 4h, i). Parasite morphology and growth were assessed following ATc-treatment for varying durations including 0, 32, 48 and 72 h of ATc (Fig. 4j–l). While aberrant morphology was observed after 48 h of ATc-treatment for the total cell, the apicoplast and the mitochondrion appeared intact after 32 h of treatment. However, a slight defect in growth and a small percentage of deformed parasites were observed after 32 h ATc, which aggravated over longer durations of ATc-treatment (48 and 72 h) (Fig. 4l). As observed for the plaque assay, defects in intracellular growth and morphology were rescued by expression of a second copy DPCK-myc (Fig. 4l).

To assess DPCK activity, metabolite extracts of parental and DPCK depleted parasites strains were analyzed by LC-MS on extracts collected after 32 and 72 h of ATc-treatment. Upon 32 h ATc-induced DPCK downregulation, Pan levels were unaltered, while deP-CoA levels increased significantly (3~fold), consistent with a block of the expected enzymatic function, the phosphorylation of deP-CoA (Fig. 4m). CoA levels were unaltered at this relatively early time-point of DPCK-downregulation, whereas at the time point of 72 h ATc-treatment, DPCK depleted parasites presented significantly higher Pan and 15-fold increased levels of the substrate deP-CoA but markedly reduced levels of CoA (~10-fold) and the active metabolite acetyl-CoA (2–3-fold) (Fig. 4n). While DPCK-iKD parasites are expected to be unfit and partially deformed at this stage based on our phenotyping described above, these experiments validate the enzymatic function of DPCK. The high levels of accumulating Pan and deP-CoA measured in these parasite extracts support that these parasites still have intact membranes and at least a partially functional metabolism.

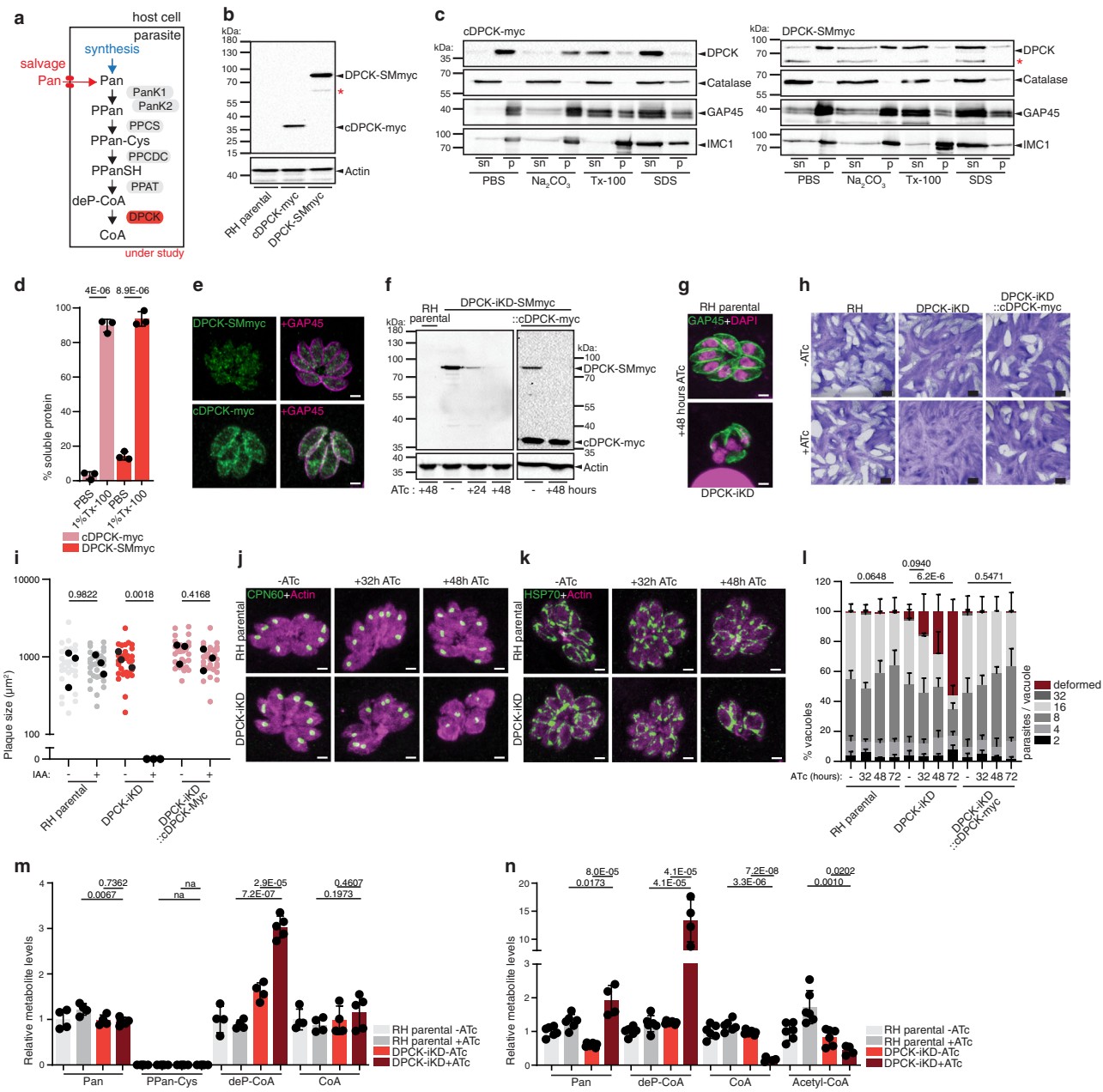

**Fig. 4 The final step of CoA biosynthesis is essential in *T. gondii* and relies on a membrane-bound cytoplasmic DPCK. a** Scheme of CoA biosynthesis pathway, highlighting the final CoA synthesis enzyme dephospho-CoA kinase (DPCK). See Fig. 1 for abbreviations. **b** Western blot of cDPCK-myc (expected MW 37 kDa, $n = 3$) and spaghetti-monster (SM)myc tagged DPCK (expected MW 72 kDa, $n = 3$). Antibodies anti myc and anti actin as loading control. **c** Western blot of solubility assay of cDPCK-myc ($n = 3$) and DPCK-SMmyc ($n = 3$). Antibodies anti myc/ anti-catalase, anti GAP45 and anti IMC1 as markers for PBS-, Triton- and SDS-soluble fraction, respectively. sn supernatant, p pellet. Solubility quantified in **d** ($n = 3$, mean and SD). **e** IFA of cDPCK-myc ($n = 3$) and DPCK-SMmyc tagged strains ($n = 3$). Antibodies: anti myc, anti GAP45 as marker of parasite pellicle. **f** Western blot of DPCK-iKD-SMmyc and cDPCK-myc protein levels upon anhydrotetracycline (ATc) addition. Antibodies anti myc, anti actin as loading control ($n = 3$). **g** IFA of 48 h ATc treated DPCK-iKD. Antibodies: anti GAP45 (parasite pellicle), DAPI (nucleus) ($n = 3$). **h** Plaque assay of DPCK-iKD and DPCK-iKD::cDCPK-myc ($n = 3$). **i** Quantification of plaque size. Black dots present the average of each independent experiment. ($n = 3$, 2-sided *t*-test). **j, k** IFAs of apicoplasts and mitochondria in DPCK-iKD, at the indicated timepoints of ATc addition. Antibodies: anti actin (cytoplasm), anti-CPN60 (apicoplast), anti-HSP70 (mitochondrion) ($n = 3$). **l** Intracellular growth assay of RH and DPCK-iKD parasites over 24 h. Deformed parasites scored as indicated in Fig. 1h ($n = 3-6$, mean and SD, 2 tailed *t*-test on "deformed" category). **m, n** LC-MS analysis of metabolite abundances in DPCK depleted parasites (32 h ATc, m; 72 h ATc, n). $n = 4-5$ for **m** and 5-6 for **n**, mean and SD, 2-sided Student's *t* test. Metabolite levels normalized to internal standard ($^{13}C_6/^{15}$N-isoleucine for n and $^{13}C_3/^{15}$N-pantothenate for m) and quantified relative to RH − ATc. *n* number of independent biological replicates. Source data are provided as a Source Data file. White scale bar 2 μm, black scale bar 1 mm. Red asterisks indicate degradation products of the tagged protein.

Taken together, the combination of reverse genetics and metabolomic analyses firmly confirms the enzymatic function of four (PanK1 + 2, PPCDC, DPCK) of the six *T. gondii* CoA biosynthesis enzymes, catalyzing five steps. Critically, a block at any of the three investigated steps in CoA synthesis pathway results in severe depletion of essential metabolites, such as CoA and acetyl-CoA, ultimately causing parasite death. Several studies have highlighted the druggability of the pathway in Apicomplexa, due to the inhibition of acetyl-CoA generating enzymes[28–30]. Our results uncover that specific inhibition of *T. gondii* CoA synthesis enzymes could be an effective treatment against toxoplasmosis and cannot be overcome by salvage of CoA or other intermediates.

**The Pan synthesis pathway is dispensable due to Pan uptake in tachyzoites**. While CoA can be produced by all organisms, humans and animals are unable to synthesize its precursor Pan, which is synthesized by plants, fungi and most bacteria[31]. Some protists (mainly Alveolata and Stramenopiles) encode in their genome the required enzymatic pathway. *T. gondii* encodes the three proteins required for de novo synthesis of Pan (Fig. 5a): i) the branched chain amino acid transaminase (BCAT), previously localized to the mitochondrion and shown to be active but dispensable in tachyzoites[32]; ii) the bifunctional enzyme keto-pantoate hydroxymethyltransferase - ketopantoate reductase, (KPHMT-KPR) and iii) the pantoate β-alanine ligase (PBAL, also known as Pan synthetase, encoded by *panC* in *E.coli*)[9,10,33,34]. In contrast to model organisms, alveolates encode the pathway with various gene duplications and rearrangements, like for the bifunctional KPHMT-KPR in *T. gondii* (Fig. 1a, b, Supplementary Data 1).

To investigate the localization of Pan synthesis in *T. gondii*, KPHMT-KPR (TGME49_257050) and PBAL (TGME49_265870) were C-terminally epitope- tagged at the endogenous locus with SMmyc[24] and 3xTy, respectively. Both tagged proteins migrated on western blots close to the predicted sizes of 144 kDa and 93 kDa, respectively (Fig. 5b). KPHMT-KPR-SMmyc co-localized with the mitochondrial marker HSP70, while PBAL-Ty localized to the nucleus (Fig. 5c). These findings point towards pantoate, as has been described for ketopantoate in plant model organisms[35], being synthesized in the mitochondrion and transported into the cytosol/nucleus, where it is converted into Pan by PBAL. Furthermore, these observations, together with a previous study of BCAT[32], reveal that all Pan synthesis genes are expressed in tachyzoites. To assess the importance of Pan synthesis for *T. gondii* fitness, *KPHMT-KPR* and *PBAL* genes were deleted by Cas9-mediated double homologous recombination (Supplementary Fig. 2a). Consistent with a published genome-wide fitness score suggesting dispensability[15], parasites lacking Pan synthesis enzymes exhibited no defect in the lytic cycle compared to the parental RH strain (Fig. 5d, e). These results stand in contrast to a previous pharmacological study[33], where the efficacy of *Mycobacterium tuberculosis* Pan synthetase inhibitors was tested against *T. gondii* tachyzoites in vitro. The authors reported growth inhibition in the nanomolar range for the compounds SW413 and SW404, which was partially rescued by supplementation with high levels of exogeneous Pan[33]. However, efficient growth reduction through inhibition of *T. gondii* PBAL is incompatible with the dispensability of the *PBAL* gene reported here and supported by the genome-wide fitness screen[15]. In consequence, the reported efficacy of SW413 and SW404 against *T. gondii* is possibly owed to off-target effects. Competitive inhibition of the PanKs rather than PBAL by the compounds, could potentially explain the growth inhibition and the partial rescue through excess Pan.

Of relevance, the dispensability of the Pan synthesis enzymes implies that *T. gondii* tachyzoites are capable of salvaging Pan. Pan may be taken up from the host and also directly from the culture medium during extracellular periods where it is present at 8.3 or 0.5 μM (Dulbecco's Modified Eagle Medium, DMEM; Roswell Park Memorial Institute, RPMI, respectively). To examine whether tachyzoites can take up exogenous Pan, stable isotope labeling followed by LC-MS was performed. Isolated extracellular parasites were incubated for 15 min in medium containing 40 μM $^{13}C_3/^{15}N$-Pan and for reference 1 mM $^{13}C_6/^{15}N$-isoleucine (ILE), before quenching the metabolism and washing cells rigorously. Both metabolites were readily taken up within this short time frame leading to significant increase in the abundance of the M7 and M4 isotopologues of ILE and Pan, respectively, as measured by LC-MS (Fig. 5f).

To further scrutinize whether *T. gondii* takes up exogenous Pan, stable isotope labeled Pan ($^{13}C_3/^{15}N$-Pan, 100 μM) was added to the culture media for 40 h to infected HFFs. This led to pronounced labeling (80 %) of the intracellular Pan pool, as measured by gas chromatography-mass spectrometry (GC-MS) analysis of parasite extracts, indicating efficient uptake of the vitamin (Fig. 5g, Supplementary Fig. 2b). These findings demonstrate that *T. gondii* tachyzoites, as *P. falciparum*[36] and as expected for other apicomplexans, is capable of salvaging Pan. While essentiality of Pan uptake in *Plasmodium spp.* was proposed several decades ago[37] and a putative Pan transporter was proposed in *P. falciparum*[38], subsequent studies provided evidence that this putative Pan transporter localizes to secretory organelles and is implicated in exocytosis both in *P. berghei* and in *T. gondii*[39,40]. Specifically, *T. gondii* depleted in the closest homolog (Transporter Facilitator Protein 1, TFP1) of the putative *P. falciparum* Pan transporter (*Pf*PAT) displayed no intracellular growth defect, which is highly atypical for proteins involved in metabolism, but rather presented impaired host cell attachment, invasion and egress[40]. Thus, the apicomplexan Pan transporter remains to be identified, as a putative target for intervention against these important group of human pathogens. To investigate the importance of Pan salvage in *T. gondii*, tachyzoites were grown in custom-made culture medium lacking Pan and supplemented with dialyzed fetal bovine serum (dFBS). The absence of Pan in the culture medium including 5% dFBS was confirmed by GC-MS analysis (Supplementary Fig. 3). Unexpectedly, neither wildtype (RH parental) nor *pbal-ko* parasites cultivated in medium depleted of Pan displayed reduced fitness or altered growth rates, as shown by plaque assay and intracellular growth assay (Fig. 5h–j). To investigate how *T. gondii* parasites cope under these Pan-limiting conditions, GC- and LC-MS analyses were employed to directly measure the intracellular levels of Pan and its downstream metabolites, deP-CoA, CoA and acetyl-CoA. Parasites cultured in Pan-depleted medium for two or three passages in pre-depleted host cells indeed presented a 2–3-fold decrease but were not completely devoid of Pan (Fig. 5k, l). From the second to the third consecutive passage of parasites in host cells pre-depleted of Pan, intracellular Pan levels did not decrease further (Fig. 5k). Strikingly, while CoA and its precursors were significantly decreased in parasite extracts, acetyl-CoA levels were unchanged or even increased upon Pan-depletion, indicating that even low levels of Pan are sufficient to synthesize ample amounts of this critical metabolite (Fig. 5k).

The parasite's ability to grow in medium deficient of Pan was previously reported, with the authors concluding that *T. gondii* relies on Pan synthesis[28]. However, depletion of PBAL had no impact on intracellular Pan-levels in parasites cultivated either in regular or Pan-depleted culture media (Fig. 5l), as measured by

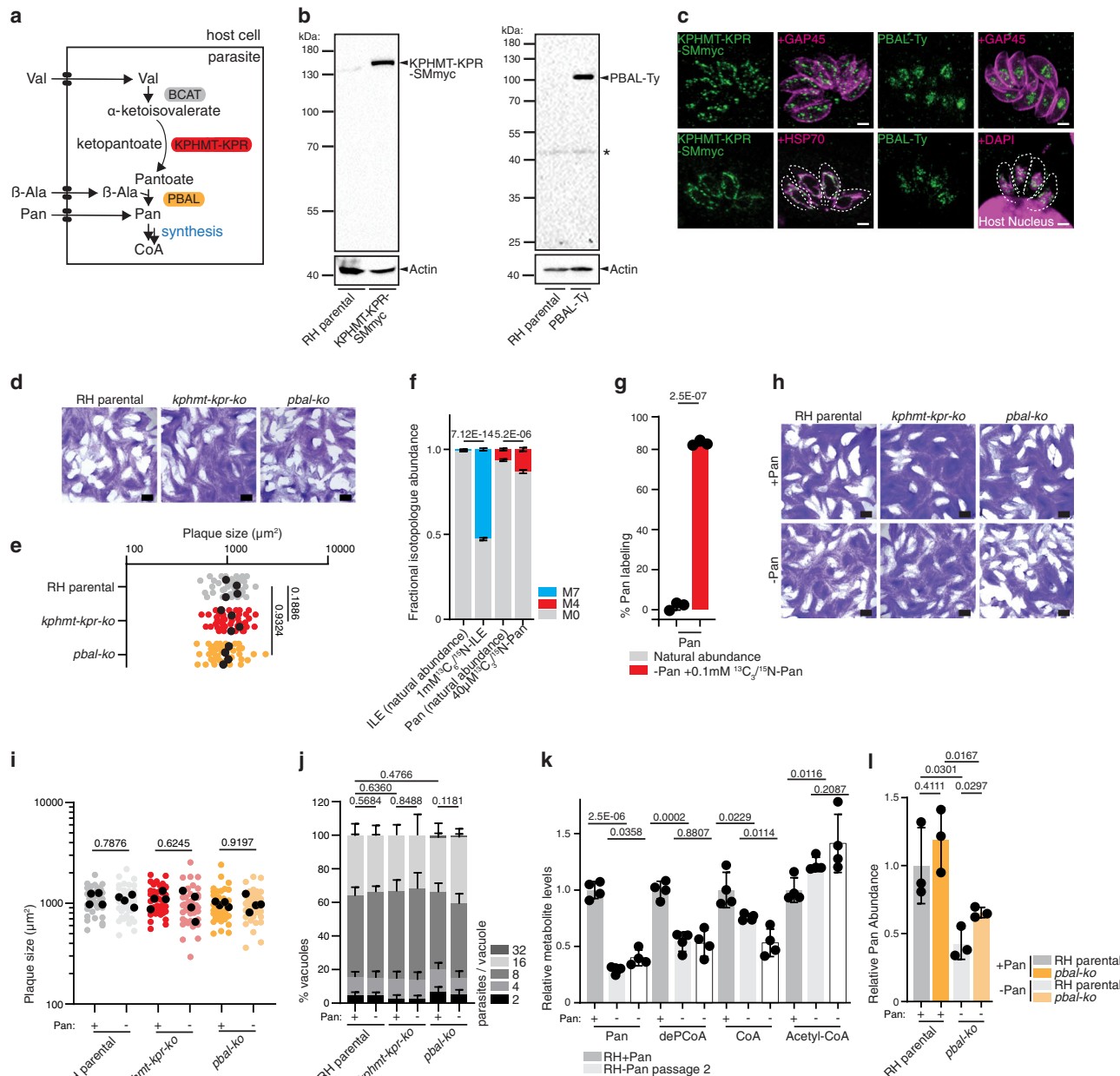

**Fig. 5 The Pan synthesis pathway is dispensable due to Pan uptake in tachyzoites. a** Scheme of the Pan biosynthesis pathway, highlighting the ketopantoate hydroxymethyltransferase - ketopantoate reductase (KPHMT-KPR) and the pantoate beta alanine ligase (PBAL). See Fig. 1 for abbreviations. **b** Western blots of endogenous C-terminally spaghetti-monster (SMmyc) tagged KPHMT-KPR (MW 144 KDa, $n = 3$) and endogenous C-terminally Ty tagged PBAL (MW 93 KDa, $n = 3$), respectively. Antibodies anti-myc and anti-Ty, anti actin as loading controls. **c** Immunofluorescence assay of KPHMT-KPR-SMmyc ($n = 3$) and PBAL-Ty ($n = 5$). Antibodies anti-myc and anti-Ty, anti GAP45 (pellicle), anti HSP70 (mitochondrion), DAPI (nucleus). **d** Plaque assay of RH parental, *kphmt-kpr-ko* ($n = 4$) and *pbal-ko* ($n = 6$). **e** Quantification of plaque size. Black dots present the average of each independent experiment. ($n = 4$, 2-sided *t*-test). **f** LC-MS analysis of Pan (M0, M4) and isoleucine (M0, M7) isotopologues, following 15 min incubation of isolated RH parasites in medium $+40 \mu M$ $^{13}C_3/^{15}N$-Pan and 1 mM $^{13}C_6/^{15}N$-isoleucine (ILE) ($n = 3–6$, mean and SD, 2-sided Student's *t* test). **g** GC-MS analysis of RH parental derived Pan upon culture in medium containing $^{13}C_3/^{15}N$-Pan (100 µM) for 40 h ($n = 4$, mean and SD, 2-sided Student's *t* test). **h** Plaque assay of mutant parasites in ±Pan culture medium ($n = 4$). **i** Quantification of plaque size. Black dots present the average of each independent experiment. ($n = 4$, 2-sided *t*-test). **j** Intracellular growth assay of mutant parasites in ±Pan culture medium ($n = 3–6$, mean and SD, 2 tailed *t*-test on the "16 parasites/vacuole" category). **k** LC-MS measurement of intracellular Pan, dephospho-CoA (deP-CoA), coenzyme A (CoA) and acetyl-CoA (AcCoA) in RH parental parasites grown for 2 and 3 passages (P2, P3) in ±Pan culture media. Metabolite levels normalized to internal standard ($^{13}C_3/^{15}N$-Pan) and quantified relative to RH + Pan ($n = 4$, mean and SD, 2-sided Student's *t* test). **l** GC-MS measurement of Pan levels in RH parental and *pbal-ko* parasites grown in ±Pan culture medium. Metabolite levels normalized to signal intensity of the total ion chromatogram and quantified relative to RH + Pan. ($n = 4$, mean and SD, 2-sided Student's *t* test). *n* number of independent biological replicates. Source data are provided as a Source Data file. White scale bar 2 µm, black scale bar 1 mm. Asterisks mark unspecific bands.

GC-MS, indicating that salvage of residual Pan rather than synthesis is responsible for the remaining Pan detected in parasites following depletion. To determine if tachyzoites are able to modulate the expression of host Pan recycling enzymes such as pantetheinases and CoA pyrophosphatases under Pan depletion conditions[41], RNA-seq analyses were performed on parasites sequentially passaged in Pan-depleted HFFs. A total of 621 transcripts were significantly changed (409 genes up- and 212 downregulated with $p$ value < 0.01) in the parasite transcriptome (Supplementary Data 2), but none displayed changes greater or lesser than 4-fold (>2 $\log_2$ fold). In the transcriptome of parasite-infected HFFs, infection with Pan-deprived parasites compared to parasites grown in rich-media, significantly altered the levels of 1691 transcripts (871 genes up- and 820 downregulated with $p < 0.01$, from Supplementary Data 2), 92 of which differentially regulated more than 4-fold (Supplementary Data 2). The genes participated in diverse cellular processes (GO terms in Supplementary Data 2), but none that could indicate an increased salvage mechanism or any alteration to the Pan/CoA metabolism within the host.

Taken together, these results reveal that a complete depletion in intracellular Pan cannot be readily achieved when cultivating *T. gondii* tachyzoites in HFFs. As long as the host cell viability is not compromised, the parasite is able to salvage sufficient Pan from its host. Due to this efficient salvage, Pan synthesis is dispensable in cultured *T. gondii* tachyzoites.

**T. gondii tachyzoites salvage Pan synthesis intermediates and uses these for Pan synthesis.** To interrogate further whether the *T. gondii* Pan synthesis pathway can contribute to the parasite's metabolite pool under Pan-limiting conditions, Pan synthesis was promoted by supplementing the culture medium with excess of PBAL substrates in the absence of exogenous Pan. Valine (Val), an essential amino acid for *T. gondii* and the substrate of BCAT (Fig. 6a), is abundantly present in culture medium (0.8 mM in DMEM) and is readily detected in parasite extracts[42] (Supplementary Fig. 4a). In contrast, the non-proteinogenic amino acid beta-alanine (β-Ala) is absent in standard culture medium, including DMEM, but is also detected at low levels in extracts of parasites cultured under standard conditions using GC-MS and LC-MS (Supplementary Fig. 4a, b). It is unclear whether *T. gondii* is capable of synthesizing β-Ala since at least 6 different β-Ala synthesis pathways have been described in model organisms, some of which involve potentially promiscuous transaminases and aldehyde dehydrogenases[43]. Since mammalian host cells are capable of producing β-Ala[44], *T. gondii* might rely on its salvage. The parasite's ability to take up exogenous β-Ala was clearly demonstrated in experiments described below. In bacteria, β-Ala has been shown to be a limiting factor for Pan synthesis, with elevated β-Ala levels leading to increased Pan synthesis[45]. However, supplementation of Val and β-Ala at 3 mM and 2 mM, respectively, to the culture medium failed to restore or increase Pan levels in parasites depleted of exogenous Pan, as measured by GC-MS (Fig. 6b). These results suggest that these precursors cannot be the only limiting factors for Pan synthesis and provide further evidence that residual intracellular Pan in depletion experiments is derived from salvage rather than its de novo synthesis. To investigate more specifically whether Pan synthesis from Val and β-Ala (Fig. 6a) is functional and active in *T. gondii* by means of a more sensitive approach, stable isotope labeling combined with LC-MS was employed. Parasites were incubated in medium containing stable isotope labeled $^{13}C_3/^{15}N$-β-Ala (3 mM) or $^{13}C_3/^{15}N$-Pan (100 μM) for 48 h. During active Pan synthesis, stable isotopes from $^{13}C_3/^{15}N$-β-Ala are expected to be incorporated into de novo synthesized Pan and further into CoA (see cartoon in Fig. 6c). However, intracellular Pan and its derived

metabolites, CoA and acetyl-CoA, remained unlabeled following growth of parasites in medium containing $^{13}C_3/^{15}N$-β-Ala (Fig. 6d). In contrast, incubation of the parasites in medium containing $^{13}C_3/^{15}N$-Pan resulted in labeling of the intracellular Pan pool, and the labeled Pan was readily incorporated into CoA and acetyl-CoA (Fig. 6d), confirming efficient Pan salvage, as described above (Fig. 5f, g), and revealing that salvaged Pan but not β-Ala is readily used for CoA synthesis. Importantly the lack of Pan synthesis from β-Ala is not due to insufficient uptake of $^{13}C_3/^{15}N$-β-Ala since stable isotope labeled β-Ala was abundantly detected in parasites cultured in its presence (Supplementary Fig. 5). It presumably passes the plasma membrane through an as of yet unidentified specific or promiscuous amino acid transporter.

In consequence, de novo Pan synthesis is inactive in tachyzoites, raising the question whether these enzymes are functional. Although the catalytic residues are clearly conserved in PBAL (Supplementary Data 3), the protein is divergent and considerably larger compared to the enzymes described in other organisms (88 kDa, vs 31 kDa for *E. coli* PBAL (also called pantothenate synthetase) and 35 kDa for *Saccharomyces cerevisiae* PBAL). Recombinant expression of full length PBAL in either bacteria or Sf9 insect cells did not yield any folded enzyme, and a construct composed of the bacterial maltose binding protein (MBP) fused to the predicted catalytic domain of PBAL (PBALcd, residues 109–492) could be purified to high purity from *E. coli* (Supplementary Fig. 6) but did not show reliable enzymatic activity when incubated with its substrates β-Ala and pantoate in vitro, in contrast to the bacterial PanC enzyme. Of relevance, this PBAL construct likely forms large soluble oligomers as it elutes in the void volume when analyzed by size exclusion chromatography. It is thus conceivable that proper folding of active recombinant PBAL requires additional partners or chaperones.

To interrogate which step(s) of Pan synthesis is/are inactive, preventing its de novo synthesis from Val and β-Ala, mutants of each Pan synthesis pathway enzyme were employed (*bcat-ko*[32], *kphmt-kpr-ko*, *pbal-ko*). Wildtype parasites (RH parental) and the mutants were cultured in the presence or absence of various Pan precursors (all 0.8 mM) in medium lacking Pan, while Val was present in all conditions at 0.8 mM (default concentration in DMEM). Pan and its precursors were identified in parasite extracts by LC-MS based on their retention time, *m/z* of the parental ion and MS² spectra of authentic standards (Supplementary Fig. 7a). The activity of each Pan synthesis enzymes was probed by supplementing the culture medium with stable isotope labeled $^{13}C_3/^{15}N$-β-Ala (0.8 mM) for 40 h, but, as above, presence of Val and β-Ala was insufficient to trigger Pan synthesis, as judged by the absence of labeled Pan (Fig. 6e, Supplementary Fig. 9a). In order to bypass the initial enzyme (BCAT), α-ketoisovalerate was supplemented in addition to $^{13}C_3/^{15}N$-β-Ala (both 0.8 mM). Interestingly, α-ketoisovalerate was detected at very low levels in extracts of non-supplemented parasites, indicating that BCAT is active, as previously reported[32] (Supplementary Fig. 7b). When supplemented at 0.8 mM, the signal intensity of intracellular α-ketoisovalerate only increased marginally (Supplementary Fig. 7c), highlighting that its acquisition by the parasite or the host is inefficient and/or salvaged α-ketoisovalerate is rapidly metabolized. Crucially, provision of exogenous α-ketoisovalerate and $^{13}C_3/^{15}N$-β-Ala did not result in the formation of labeled Pan (Fig. 6e, Supplementary Fig. 8a). Taken together, these findings point towards inactivity of the second enzyme, KPHMT-KPR, or low levels of intracellular α-ketoisovalerate are not sufficient to promote Pan synthesis. Consistent with inactivity of KPHMT-KPR, pantoate was not detected in parasites cultured in non-supplemented culture medium (Supplementary Fig. 7b). Lastly, BCAT as well as the

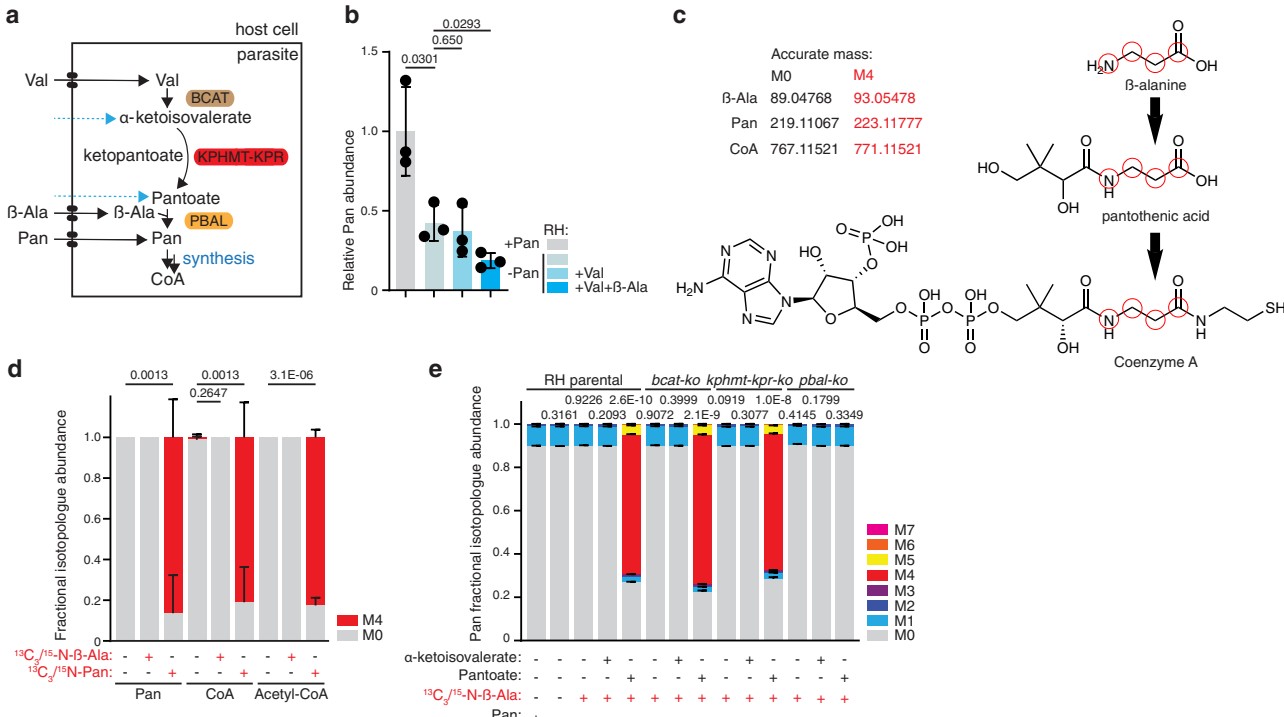

**Fig. 6 *T. gondii* tachyzoites salvage Pan synthesis intermediates and use these for Pan synthesis. a** Scheme of the Pan biosynthesis pathway, highlighting the ketopantoate hydroxymethyltransferase - ketopantoate reductase (KPHMT-KPR) and the pantoate beta alanine ligase (PBAL). Salvage of α-ketoisovalerate and pantoate as precursors for Pan synthesis is shown. See Fig. 1 for abbreviations. **b** GC-MS Measurement of relative intracellular Pan levels in RH parental parasites following incubation in medium without Pan supplemented with valine (Val, 3 mM) and β-alanine (β-Ala, 2 mM) ($n = 3$, mean and SD, 2-sided Student's *t* test). **c** Metabolite structures highlighting the theoretical incorporation of labeled atoms (red circles) from stable isotope labeled $^{13}C_3/^{15}N$-β-Ala into Pan and into CoA. Exact masses of the parental ion (M0), and heavy stable isotope labeled (M4) are presented in black and red, respectively. **d** LC-MS analysis of the relative isotopologue (M0, M4) abundance of RH parental derived metabolites following incubation in medium with 3 mM $^{13}C_3/^{15}N$-β-Ala or 100 μM $^{13}C_3/^{15}N$-Pan ($n = 4$, mean and SD, 2-sided Student's *t* test). **e** LC-MS analysis of Pan isotopologue (M0 to M7) distribution of RH parental, *bcat-ko*, *kphmt-kpr-ko*, *pbal-ko* derived metabolites following growth over 40 h in regular DMEM, DMEM − Pan +5 % dialyzed FBS supplemented with metabolites as indicated: α-ketoisovalerate, pantoate, $^{13}C_3/^{15}N$-β-Ala (each at 0.8 mM). Statistical analysis was carried out on the percentage of labeling shown in supplementary data ($n = 3$, mean and SD, 2-sided Student's *t* test). *n* number of independent biological replicates. Source data are provided as a Source Data file.

penultimate Pan synthesis enzyme, KPHMT-KPR were bypassed by providing pantoate, also salvaged by the parasite (Supplementary Fig. 7c), and $^{13}C_3/^{15}N$-β-Ala (both 0.8 mM) to the culture medium. Remarkably, provision of these two Pan precursors led to the formation of labeled Pan in all strains except *pbal-ko* (Fig. 6e, Supplementary Fig. 8a), providing clear evidence that *T. gondii* PBAL is functional and active. In conclusion, KPHMT-KPR is an inactive enzyme in cultured *T. gondii* tachyzoites, preventing the de novo synthesis of Pan from Val and β-Ala.

Aside from the availability of precursors, Pan synthesis in *E. coli* is also regulated through inhibition of KPHMT activity by accumulation of downstream metabolites, including Pan and CoA[46]. It is plausible that a similar regulation at the level of the KPHMT-KPR enzyme is occurring in *T. gondii* and given that the enzyme is expressed, posttranslational modifications might regulate its activity. In conclusion, despite expressing the enzymes required for Pan biosynthesis, at least two of which are enzymatically active (BCAT:[32], PBAL: this study), tachyzoites do not synthesize Pan unless the final precursors are added in the culture media, even when exogenous Pan levels are low. Thus, the physiological role of Pan synthesis is unclear and was further scrutinized as outlined below.

**De novo Pan synthesis is dispensable and inactive in in vitro differentiated bradyzoites.** To address why coccidian parasites,

but not other Apicomplexa, have retained the capacity to produce Pan, its importance was examined in the cyst-forming bradyzoite stage. While type I RH parasites are highly virulent and cause acute infection, with mice typically succumbing to the infection within 7 days, type II parasites are less virulent, and infected mice commonly overcome the acute phase of infection and develop chronic toxoplasmosis. Of relevance, stage conversion from tachyzoites to bradyzoites can also be achieved in vitro, using stress triggers such as high pH[47] on *T. gondii* strains that are prone to encystation (ME49, Pru), or by upregulating a recently identified bradyzoite differentiation factor, BFD1[48]. Type II parasites, like the ME49 strain, display a tendency to convert spontaneously to bradyzoites at low levels even under standard culture conditions and have a slower growth rate than type I RH parasites.

To understand the role of Pan synthesis (Fig. 7a) in type II parasites, KPHMT-KPR-SMmyc and PBAL-Ty strains were generated in ME49 parasites, using the same strategy as in Fig. 5. The level of both proteins dropped significantly following conversion to bradyzoites through cultivation in alkaline medium for 7 days (Fig. 7b, quantified in 7c). Efficient tachyzoite to bradyzoite conversion was confirmed by detection of the tachyzoite marker surface antigen 1 (SAG1) and/or the late bradyzoite marker P21[49]. While these reduced protein levels may indicate reduced activity of the pathway, it is unclear if protein levels correspond to enzymatic activity. Thus, the importance of

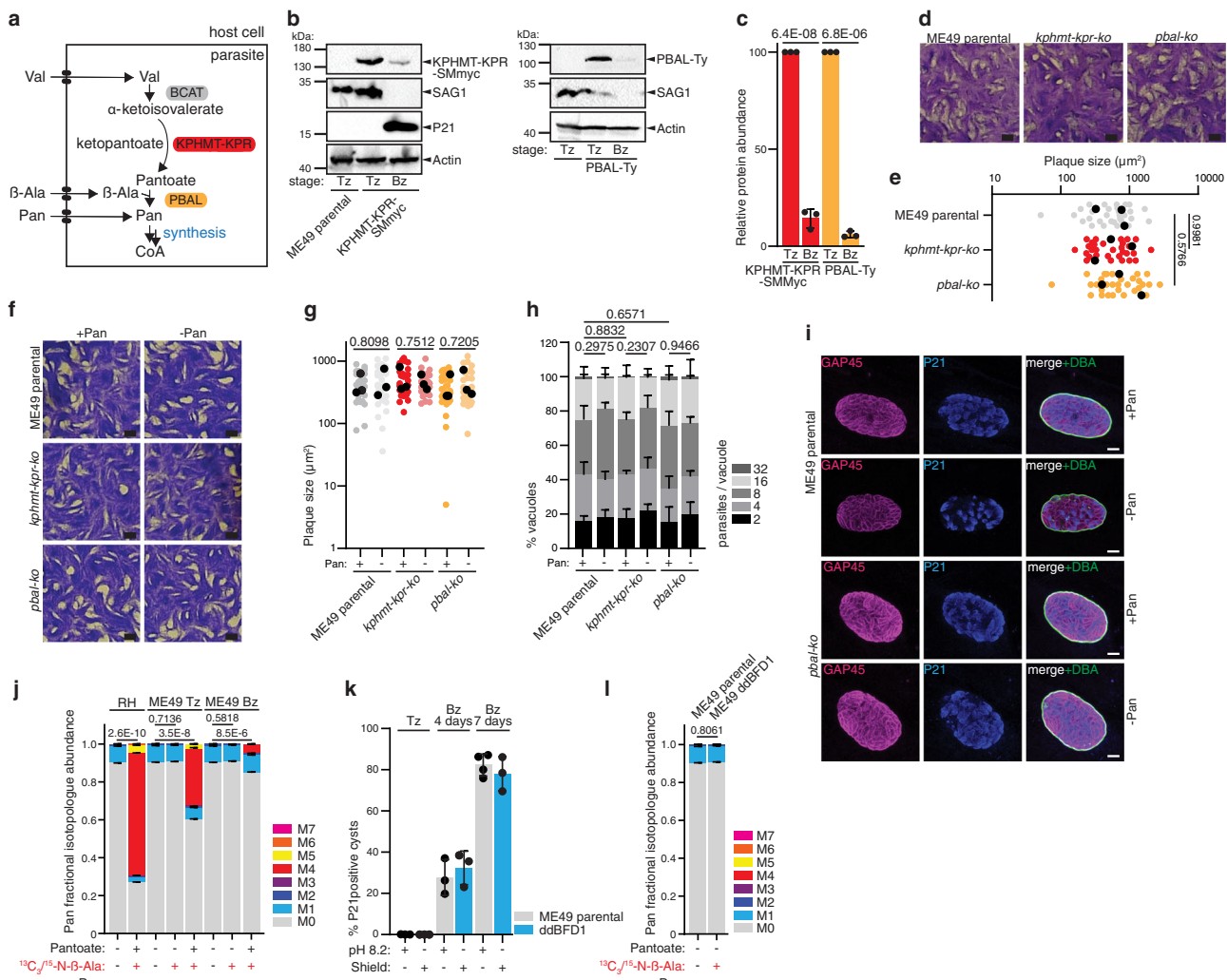

**Fig. 7 De novo Pan synthesis is dispensable and inactive in in vitro differentiated bradyzoites. a** Scheme of the Pan biosynthesis pathway, highlighting the ketopantoate hydroxymethyltransferase - ketopantoate reductase (KPHMT-KPR) and the pantoate beta alanine ligase (PBAL). See Fig. 1 for abbreviations. **b** Western blot of KPHMT-KPR-SMmyc ($n = 3$) and PBAL-Ty ($n = 3$) of 1-week alkaline medium differentiated bradyzoites. Antibodies anti myc and Ty, anti SAG1 (tachyzoite markers), anti P21 (late bradyzoite marker), anti actin as loading control. **c** quantification of protein levels, relative to tachyzoites ($n = 3$, mean and SD, 2-sided Student's $t$ test). **d** Plaque assay of ME49 mutant parasites ($n = 5$). **e** Quantification of plaque size. Black dots represent means of independent experiments ($n = 3$, 2-sided Student's $t$ test). **f** Plaque assay of ME49 mutant parasites, in ±Pan culture medium ($n = 3$). **g** Quantification of plaque size. Black dots represent means of independent experiments ($n = 3$, 2-sided Student's $t$ test). **h** Intracellular growth assay of mutant parasites in ±Pan culture medium ($n = 3$, mean and SD, 2 tailed $t$ test on the "16 parasites/vacuole" category). **i** Immunofluorescence assay (IFA) of parental ME49 and *pbal-ko* 2 weeks in vitro differentiated bradyzoites in ±Pan culture medium ($n = 3$). Antibodies anti-GAP45 (parasite pellicle), anti P21 (late bradyzoites), DBA (cyst wall). **j** LC-MS analysis of Pan isotopologue (M0 to M7) abundance in RH parental, ME49 parental tachyzoites (Tz) or 5 days alkaline-differentiated bradyzoites (Bz) cultured 40 h in medium ±Pan supplemented with Pan (40 µM), pantoate and or $^{13}C_3/^{15}N$-β-Ala (each at 0.8 mM) as indicated. Statistical analysis was performed on percentage of labeling shown in supplementary data ($n = 4$, mean and SD, 2-sided Student's $t$ test). **k** Quantification of bradyzoite marker P21 of alkaline-differentiated ME49 parental, or ddBFD1 (3 µM Shield). $n = 3$, mean and SD. **l** LC-MS analysis of relative Pan isotopologue (M0 to M7) abundance of 6 days ME49 parental alkaline-differentiated bradyzoites, and ddBFD1 (6 days, 3 µM Shield). ddBFD1 parasites were cultured in medium containing $^{13}C_3/^{15}N$-β-Ala (0.8 mM, 40 h). Statistical analysis was carried out on the percentage of labeling shown in the supplementary data ($n = 4$, mean and SD, 2-sided Student's $t$ test). $n$ number of independent biological replicates. Source data are provided as a Source Data file. White scale bar 6 µm, black scale bar 1 mm.

the pathway was probed further by generating *kphmt-kpr-ko* and *pbal-ko* mutants in the ME49 *hxgprt-ko ku80-ko* strain, referred to as ME49 parental (Fig. 7a, Supplementary Fig. 9a). By plaque assay in standard growth conditions, *pbal-ko* and *kphmt-kpr-ko* showed no lytic cycle defect when compared to the parental ME49 strain (Fig. 7d, e), as documented with RH (Fig. 5d). Furthermore, no plaque assay, or growth defect was detected when wildtype or mutant parasites were grown in −Pan medium (Fig. 7f–h). Given that no plaque assay defect could be detected, no defects are expected in any of the steps of the parasite lytic

cycle. These results are consistent with those described for RH parasites above and point towards inactivity of the pathway in vitro, with parasites relying on highly efficient Pan salvage from the host, which supports growth even when Pan is removed from the culture medium.

To specifically address the importance of the pathway in encysted bradyzoites, ME49 parental and *pbal-ko* parasites were differentiated in alkaline medium, with or without Pan, for 2 weeks and cyst morphology was assessed by IFAs. The *T. gondii* cyst wall is rich in carbohydrates and efficiently stains with

*Dolichos biflorus* agglutinin (DBA), while individual bradyzoites can be stained with the late bradyzoite marker P21. Neither parental ME49 parasites, nor *pbal-ko* displayed a defect in cyst morphology or in tachyzoite to bradyzoite differentiation when cultured in the presence or absence of Pan (Fig. 7i).

To measure whether Pan synthesis is active in bradyzoites in vitro, stable isotope labeling followed by LC-MS analysis was performed on parasite extracts collected 6 days after pH induced differentiation. As for type I RH parasites, no Pan synthesis in type II tachyzoites or bradyzoites was observed, as judged by the absence of labeled atoms incorporated from $^{13}C_3/^{15}N$-β-Ala into Pan (Fig. 7j, Supplementary Fig. 8b). The same result was also obtained through measuring by GC-MS the absence of labeling in Pan of ME49 tachyzoites following incubation of parasites in medium containing 2 mM $^{13}C_3/^{15}N$-β-Ala for 40 h (Supplementary Fig. 8c). To assess whether Pan synthesis occurs in ME49 parasites, when these are supplemented with pantoate and $^{13}C_3/^{15}N$-β-Ala, as described above for RH, ME49 parental tachyzoites and bradyzoites were cultured in the presence of 0.8 mM pantoate in addition to 0.8 mM $^{13}C_3/^{15}N$-β-Ala for 40 h. As for RH parasites, these conditions led to Pan synthesis, highlighting that PBAL is also active in ME49 parasites. Strikingly, the Pan labeling was markedly reduced in ME49 tachyzoites and further in bradyzoites compared to RH parasites, labeled under equivalent conditions (Fig. 7j, data from Fig. 6f is included for comparison, Supplementary Fig. 8b). This is likely due to one or several of the following factors: reduced uptake of Pan precursors, reduced growth rate and overall metabolic activity, as well as reduced expression of *PBAL* at the mRNA (Fig. 1a, Supplementary Data 1) and protein level (Fig. 7b, c). To assess whether a different mode of differentiation would reveal a role for Pan synthesis, we made use of a strain that overexpresses the bradyzoite differentiation factor BFD1, which was recently identified[48]. The bradyzoite differentiation factor was fused to a destabilization domain (ddBFD1), which is stabilized in the presence of the ligand Shield. Efficient differentiation of ddBDF1+Shield parasites, comparable to that through alkaline pH-treatment of parental ME49, was validated by assessing the P21 staining (Fig. 7k).

Similar to when parasites are differentiated through alkaline pH, Pan synthesis remained inactive in ddBFD1+Shield, as judged by the absence of labeling in Pan when parasites were cultured in the presence of $^{13}C_3/^{15}N$-β-Ala (0.8 mM) for 40 h (Fig. 7l, Supplementary Fig. 8d). In summary, Pan synthesis was found to be dispensable and inactive in in vitro ME49 tachyzoites and bradyzoites, as observed for RH tachyzoites in culture.

**Pantothenate synthesis is essential for chronic infection of *T. gondii* in vivo.** Lastly, the relevance of Pan biosynthesis (Fig. 8a) was investigated in vivo during acute and chronic infection in the mouse model of toxoplasmosis. RH *kphmt-kpr-ko* and *pbal-ko* strain mutants exhibited the same virulence during acute infection of mice, as compared to the parental strain, with all mice succumbing to the infection within 10 days (Fig. 8b).

To address the function of PBAL more specifically, ME49 *pbal-ko* parasites were complemented with either a C-terminally myc-tagged full length cDNA of *T. gondii* PBAL (*pbal-ko*::*TgPBAL-myc*), or a C-terminally myc-tagged *E. coli* PBAL enzyme (*pbal-ko*::*cEcPBAL-myc*) (Supplementary Fig. 8b). Both constructs were under the control of the constitutive tubulin promoter. Western blot analysis showed the expected MW for both tagged proteins (Fig. 8c). While IFAs showed nuclear localization for *TgPBAL-myc*, consistent with its endogenous localization, *EcPBAL-myc* was found to be cytosolic (Fig. 8d). To ensure that these strains were not impacted by the complementation, plaque assays were performed under standard growth conditions and no defect or

gain of fitness was associated with the complemented strains (Fig. 8e, f). Finally, virulence of the panel of mutants was tested during chronic infection in mice. While ME49 parasites are less virulent compared to RH, and mice usually survive the acute phase of the disease, in some cases, mice succumb to the infection over the course of the experiment. This may be owed to an intense acute infection (first two weeks following inoculation), or a severe chronic infection (3–4 weeks after infection) associated with very high cyst numbers that can lead to weight loss, loss of motor skills and apathy. The *kphmt-kpr-ko* and *pbal-ko* mutants in ME49 showed a significant reduction in virulence compared to mice infected with the ME49 parental strain, which partially (40 % by day 30) succumbed to the infection over the course of the experiment, many as late as 3–4 weeks after inoculation (Fig. 8g). Five weeks after infection, the surviving mice were sacrificed to determine the cyst burden in the brain. Strikingly, the absence of either KPHMT-KPR or PBAL led to a dramatic drop in cyst numbers in the chronically infected animals (Fig. 8h) including a mouse with zero cysts infected with *pbal-ko* parasites, which was confirmed to be infected by testing for seroconversion (Fig. 8h, Supplementary Fig. 9c). Complementation of *pbal-ko* with either *TgPBAL-myc* or *EcPBAL-myc* led to a minor rescue of virulence, and an incomplete but significant restoration of cyst numbers, demonstrating the activity and essentiality of Pan synthesis in the bradyzoite stage (Fig. 8h). The virulence of the complementing strains in the mice is expected to be reduced with respect to ME49 parental strain, given the only partial complementation of the cyst burden. The few cysts produced during infection with *kphmt-kpr-ko* and *pbal-ko* were collected and shown to recrudesce in vitro by infecting monolayers of HFFs within 2 weeks, to levels comparable to parental ME49 (Fig. 8i), indicating that remaining cysts are viable and can access sufficient Pan from the host cell.

While the Pan synthesis pathway was at least partially inactive in vitro, hampering de novo Pan synthesis, the in vivo model for the chronic stage of disease uncovers a function for the complete pathway during encystation. This specific role of Pan synthesis is consistent with conservation of the pathway exclusively in coccidia amongst apicomplexans.

Taken together, the findings highlight a discrepancy between the metabolism of in vitro and in vivo differentiated bradyzoites, with Pan synthesis being dispensable in vitro but critical for cyst formation in vivo (Fig. 8j). Although in vitro differentiated bradyzoites recapitulate several hallmarks of in vivo bradyzoites such as the formation of cyst wall, accumulation of amylopectin granules and expression of several marker proteins within the plasma membrane (SRS9, PMA1) and in the cytoplasm (LDH2, BAG1)[50], little is known about how closely the metabolism of in vitro bradyzoites resembles that of those in vivo. Undoubtedly, the culture in rich medium at ambient $O_2$ in human foreskin fibroblasts, provides a suboptimal model for the conditions encountered by bradyzoites within the brain. Current metabolomic approaches do not allow the measurement of pure in vivo bradyzoites as they are hampered by low parasite yields, host cell/tissue contamination and long purification protocols, which are expected to result in a distortion of the metabolome.

It remains unclear whether the switch to synthesize Pan occurs specifically in response to sensing altered levels of metabolites, such as Pan and β-Ala, or whether it occurs as part of a larger transcriptional regulatory switch following differentiation in vivo. Pan concentrations may vary in different regions of the brain[51], likely making Pan synthesis essential in only a fraction of the developing cysts. Exogenous Pan is expected to cross the parasite's parasitophorous vacuole membrane and the cyst wall through the molecular sieve (<1 kDa and 1.3 kDa), for which the exclusion size is well over the MW of Pan[52,53]. Nevertheless, how parasites within mature cysts access host metabolites remains

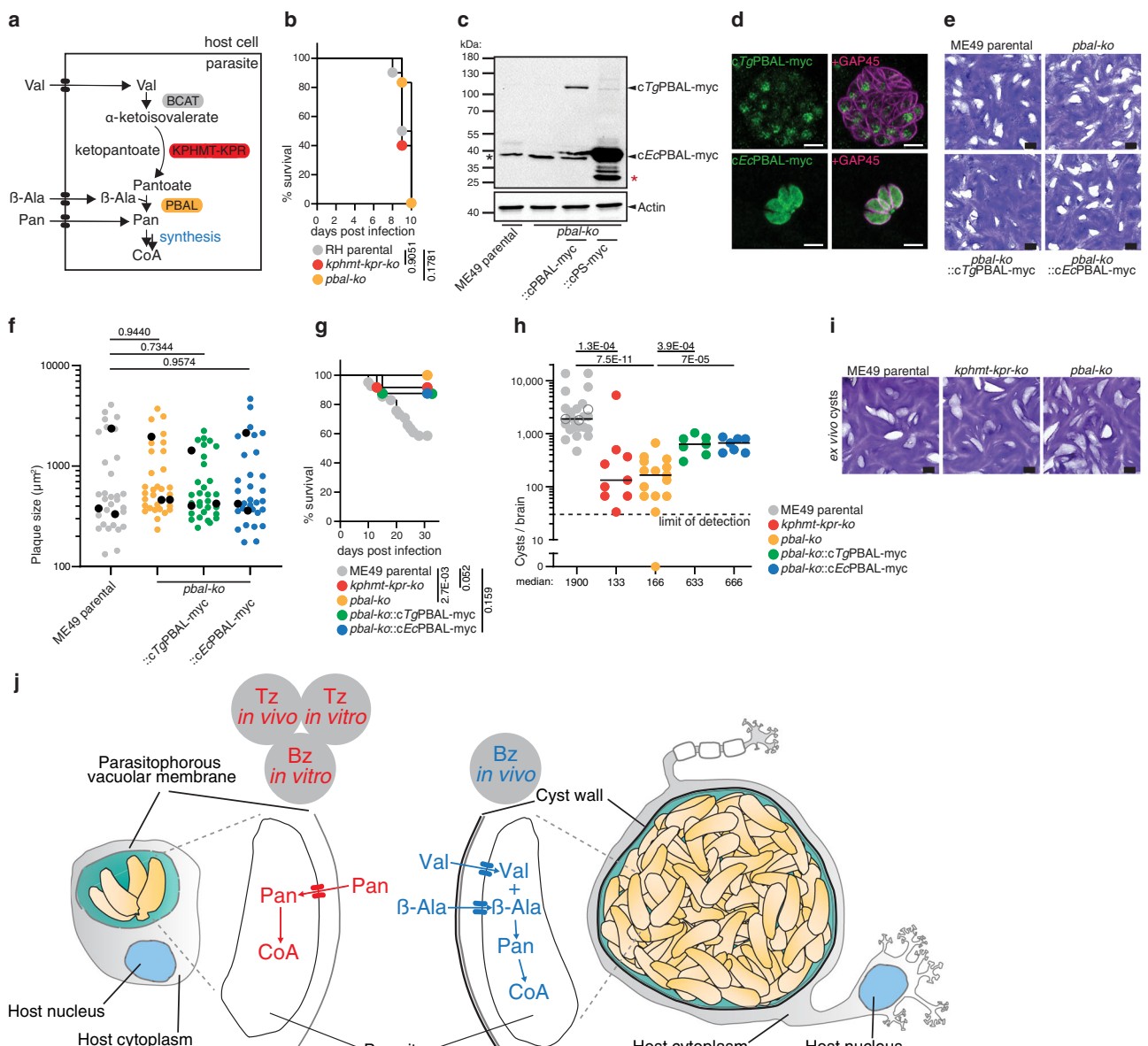

**Fig. 8 Pantothenate synthesis is essential for chronic infection of *T. gondii in vivo*. a** Scheme of the Pan biosynthesis pathway, highlighting the ketopantoate hydroxymethyl transferase - ketopantoate reductase (KPHMT-KPR) and the pantoate beta alanine ligase (PBAL). See Fig. 1 for abbreviations. **b** Survival curve of mice infected with RH parental (*n* = 10), *kphmt-kpr-ko* (*n* = 5) and *pbal-ko* (*n* = 5) parasites (from 2 experiments, logrank Mantel–Cox test). **c** Western blot of complementation strains *pbal-ko*::c*Tg*PBAL-myc (*n* = 3) and *pbal-ko*::c*Ec*PBAL-myc (*n* = 4). Antibodies anti myc, anti actin as loading control. **d** IFA of *pbal-ko*::c*Tg*PBAL-myc (*n* = 5) and *pbal-ko*::c*Ec*PBAL-myc (*n* = 3). Antibodies anti myc, anti GAP45 as marker of parasite pellicle. **e** Plaque assay of ME49 parental, *pbal-ko* and complement strains::c*Tg*PBAL-myc,::c*Ec*PBAL-myc (*n* = 3). **f** Quantification of plaque size. Black dots present mean values of independent experiments (*n* = 3). **g** Survival curves of mice infected with ME49 parental (*n* = 41, 24 survived), ME49 *kphmt-kpr-ko* (*n* = 12, 11 survived), ME49 *pbal-ko* (*n* = 17, 17 survived), ME49 *pbal-ko*::c*Tg*PBAL-myc (*n* = 8, 7 survived), ME49 *pbal-ko*::c*Ec*PBAL-myc (*n* = 8, 7 survived) strains, pooled from 5 independent experiments. **h** Ex vivo cyst count from mouse brains infected for 5 weeks with parental ME49, *kphmt-kpr-ko*, *pbal-ko*, *pbal-ko*::c*Tg*PBAL-myc, and *pbal-ko*::c*Ec*PBAL-myc parasites (*n* = 25, 9,15, 7 and 7, respectively, pooled from 5 independent experiments). Empty circles present counts before the 5 weeks endpoint. Median is represented, 2-sided Mann–Whitney test. **i** Representative plaque assay of brain-derived cysts from ME49 parental, *kphmt-kpr-ko* and *pbal-ko* infected mice (*n* = 3, 3, 4, respectively). **j** Cartoon schematic depicting the different modes of Pan acquisition in tachyzoites (Tz) and bradyzoites (Bz). *n* number of independent biological replicates. Source data are provided as a Source Data file. White scale bar 5 μm. Black scale bar 1 mm. Red asterisk marks degradation product of tagged protein. Black asterisk marks unspecific signal.

unknown. The average β-Ala concentration within the brain, the main site of *T. gondii* persistence, has been estimated to be 30–80 μM[54,55], hence it is sufficiently abundant for salvage by *T. gondii* bradyzoites. Intriguingly, β-Ala is particularly abundant in tissues that function as primary niches for *T. gondii* bradyzoite cysts during chronic toxoplasmosis, such as the central nervous system (CNS), the retina and muscle tissue[55], possibly making it

an essential metabolite for encystation. It is compelling to speculate that β-Ala in these tissues may be a factor contributing to the tissue tropism of *T. gondii* bradyzoites.

The identification of a metabolic pathway that is essential for the persistent stage of toxoplasmosis but is absent in its host paves the way for the development of new intervention. Of interest, an *M. tuberculosis* strain auxotrophic for Pan also

presents a dramatically reduced virulence in the mouse model[56], indicating that Pan synthesis could be an essential feature for chronic infection of various intracellular pathogens.

## Methods

**Parasite strains, culture and selection**. RH *hxgprt-ko ku80-ko* parasites (RH)[57], RH *ku80-ko hxgprt-ko*::Tir1[14], and ME49 *hxgprt-ko ku80-ko* (ME49) were grown in confluent human foreskin fibroblasts (HFF, ATCC SCRC-1041) in DMEM (Gibco 41965039) supplemented with 5% fetal calf serum and 40 μg/ml gentamycin (Gibco 15710-049). Parasites were selected for 1 week with 25 μg/ml mycophenolic acid and 50 μg/ml xanthine for HXGPRT positive selection[58], 1 μM pyrimethamine for DHFR-TS positive selection[59], 5 μM FUDR for UPRT negative selection[60]. Stable parasite populations were cloned by serial dilution.

**Pantothenate depletion and media supplementations**. Custom-made pantothenate depleted medium was purchased from Gibco based on regular DMEM recipe (Gibco 41965039). 5% dialyzed FBS (Pan Biotech P30-2102; 10,000 Da exclusion size membrane) and 40 μg/ml gentamycin (Gibco 15710-049) were supplemented. HFFs were grown for at least one day in Pan-depleted medium prior parasite infection. Before medium change, HFFs were washed once with Pan depleted media to remove residual Pan. Regular DMEM + 5% FCS was supplemented with indicated concentrations of additional pantothenate (Enzo ALX-460-003-G011), PanSH (Sigma 16702) or CoA (Sigma C3144) for experiments described in Fig. 3l, m.

**Cloning and parasite lines generation**. Primers are listed in Supplementary Table 1. PanK1-mAID-HA parasites were generated with primers 9157 (gRNA[12,13]), 9158/9159 (homology template[14]); integration was checked with primer couples 9160/5370, 1629/9161, 9160/9161. PanK1-Ty parasites were generated with primers 9157 (gRNA), 9266/9267 (homology template containing Ty-tag). PanK2-mAID-HA parasites were generated with primers 9162 (gRNA), 9163/9164 (homology template); integration was checked with primer couples 9165/7081, 4609/9263, 9165/9263. PanK2-Ty were generated with primers 9162 (gRNA), 9268/9269 (homology template). PPCDC-mAID-HA parasites were generated with primers 7897 (gRNA), 7798/7799 (homology template); integration was checked with primer couples 6255/7081, 1629/7726, 6255/7726. PPCDC-Ty parasites were generated by cloning KpnII/NsiI the 6253/6254 PCR amplified 3' of the *PPCDC* gene in p3xTy-HX plasmid, linearized in the *PPCDC* sequence and transfected. DPCK-iKD parasites were generated with primers 7960 (gRNA), 8247/8256 (homology template[61]); integration was checked with primer couples 8249/3596, 8250/1935, 8249/8250. DPCK-SMmyc parasites were generated with primers 9153/9154 (homology template[24]). DPCK cDNA was generated with primers 9046/9047 and cloned EcoRI/EcoRV in a modified pT8/UPRT plasmid[60], linearized NruI/ApaI, cotransfected with a gRNA targeting the *UPRT* locus[13] to generate the cDPCK-Myc parasite line. PBAL-Ty parasites were generated by cloning ApaI/SpfI the 7793/7794 PCR amplified 3' of the *PBAL* gene in p3xTy-HX plasmid, linearized NsiI and transfected. *pbal-ko* parasites were generated with primers 7228/7229 (2gRNA[62]), 7230/7231 (homology template); integration was checked with primer couples 7232/2074, 1629/7233, 7232/7628. KPHMT-KPR-SMmyc parasites were generated with primers 9200 (gRNA), 9381/9382 (homology template). *Kphmt-kpr-ko* parasites were generated with primers 9374/9375 (2gRNA), 9376/9377 (homology template); integration was checked with primer couples 9378/2018, 2017/ 9379, 9378/9380. *PBAL* full length cDNA (PCR amplified 10121/9156) and *E. coli PBAL* (PCR amplified 10126/10127) were cloned EcoRI/EcoRV in pT8/UPRT. The plasmids were linearized AatII/KpnI and cotransfected with a gRNA targeting the *UPRT* locuc. Integrations of the constructs in the *UPRT* locus were checked with primers 8414/3979, while loss of the locus with 9318/9319. Integration PCRs were performed once for each strain. Strains involved in animal experimentation were additionally tested for the correct genotype before every experiment.

**Immunofluorescence assays (IFAs)**. Parasites grown in HFF on coverslips were fixed with 4% paraformaldehyde 0.05% glutaraldehyde for 15 min. Fixative was quenched with PBS 0.1 M glycine. Coverslips were permeabilized 10 min with PBS 0.2%Triton X-100, blocked 1 h with blocking buffer (PBS 0.2%Triton X-100 2% bovine serum albumin), incubated with primary and secondary antibodies in blocking buffer. Images were acquired with a confocal microscope (Zeiss LSM700). Images were processed with ImageJ.

**Intracellular growth assays**. Parasites were inoculated on coverslips with confluent HFF. Briefly centrifuged (1 min at 1100 × g) and allowed to infect for 1 h at 37 °C. Non invaded parasites were removed by washing twice with DMEM. Coverslips were fixed for immunofluorescence 24 h post infection. Antibody staining (rabbit anti GAP45 and mouse anti GRA1) was used to quantify the number of parasites per vacuole, for a minimum of 200 vacuoles per condition. Quantification was performed blindly. Datasets were analyzed and plotted with GraphPad Prism.

**Plaque assays**. Freshly lysed parasites were inoculated on confluent HFF monolayers, supplemented or depleted, if indicated. Plaque sizes were quantified using the FindEdges and Segmentation (LevelSets) plugins in ImageJ (FIJI) software. Smaller dots represent each plaque size (10 for each independent experiment), while bigger black dots indicate the mean of each independent experiment.

**Western blot and solubility assays**. Parasite samples were lysed in Laemmli buffer with 10 mM DTT, boiled 10 min and loaded in an SDS-page gel for electrophoresis in reducing conditions. Blots were transferred on nitrocellulose membranes, blocked and incubated with antibodies in blocking buffer (PBS 0.05% Tween20 5% non-fat milk). Freshly lysed parasites were collected, resuspended and freeze-thawed in either PBS, PBS 1%Triton X-100, PBS 1 mM Na₂CO₃, or 0.1% SDS. Soluble and insoluble fractions (sn and p, respectively) were separated by centrifugation at 30,000 × g after 30 min incubation on ice.

**Co-immunoprecipitation**. Freshly lysed parasites were collected, lysed in cold IP buffer (PBS, 0.5%Triton X-100, protease inhibitor), freeze thawed 3 times, sonicated 5 times, centrifugated 30 min at 21,000 × g at 4 °C. The supernatant was collected (a fraction kept as input sample) and incubated 2 h at 4 °C with mouse anti Ty ascite antibody or mouse anti HA antibody, 2 h with protein A sepharose beads. Centrifuged 2 min at 2000 × g at 4 °C (a fraction kept as wash sample), washed 3 times with IP buffer and resuspended in Laemmli buffer.

**Antibodies**. Polyclonal rabbit anti GAP45 (1:10,000)[63], anti HA (1:1,000, Sigma H6908), anti HSP70 (1:1,000)[64], anti CPN60 (1:3,000)[65], anti-catalase (1:2,000)[66], anti IMC1 (1:2,000)[67]. Monoclonal mouse anti actin (1:20)[68], anti GRA1 and anti GRA3 (1:20, gifts of Dr. J. F. Dubremetz), anti myc (1:10, 9E10), anti Ty (1:10, BB2), anti P21 (1:10)[49], anti SAG1 (1:10, gift of Dr. J. F. Dubremetz). FITC conjugated lectin (DBA, 1:500, Vector Laboratories FL-1031-2). Secondary antibodies for immunofluorescence: anti mouse Alexa fluor 405 (1:3000, Invitrogen A31553), 488 (1:3000, Invitrogen A11001), 594 (1:3000, Invitrogen A11005), anti-rabbit Alexa fluor 488 (1:3000, Invitrogen A11008), 594 (1:3000, Invitrogen A11012). Secondary antibodies for western blot: anti mouse HRP (1:3000, Sigma A5278), anti-rabbit HRP (1:3000, Sigma A8275).

**Bradyzoite differentiation**. In vitro bradyzoite differentiation was performed by culturing ME49 infected HFF in alkaline medium (RPMI 1640 (Gibco 51800-019), 50 mM HEPES, 3% fetal calf serum, 40 μg/ml gentamycin (Gibco 15710-049), pH 8.2 with NaOH) starting 24 h post infection[47]. Cultures were incubated at atmospheric CO₂ concentration and medium was changed every 2 days. ME49 ddBFD1 strain was cultured in standard DMEM + 5% FCS with 3 μM Shield from day 1 post infection to differentiate into bradyzoites, as previously described[48]. Bradyzoite differentiation was quantified by immunofluorescence by counting P21 positive cysts (P21⁺, DBA⁺).

**RNA extraction and reverse transcription**. RNA was purified by acid phenol-chloroform extraction and column purified (RNeasy mini kit QIAGEN REF 74104) from HFFs grown in media with or without Pan and/or infected with wild-type RH parasites (grown in regular or Pan-depleted media). All samples were collected in biological triplicate. Samples for Pan-depleted parasites were collected at passage 8 of Pan depletion. For cDNA generation RNA was treated with DNAse (QIAGEN REF 79254) and reverse transcribed (Thermo SuperscriptII Reverse transcriptase REF 18064014) following the manufacturer's instructions. For RNAseq the quantity and quality of the extracted RNA was measured with a Qubit Fluorometer (Life Technologies) and an Agilent 2100 BioAnalyser (Agilent Technologies) respectively. Ribosomal RNA was removed by applying polyA selection. RNA was subjected to 100 bp single read sequencing on Illumina HiSeq 4000 (Illumina, San Diego, CA, USA) at the iGE3 Genomics platform at the University of Geneva (http://www.ige3.unige.ch/genomics-platform.php). Samples of uninfected and RH-infected HFFs (grown in media with or without Pan) were multiplexed and processed together in the sequencing lane of the flow cell.

**RNaseq data processing and analysis**. The quality of the reads was assessed with FASTX-Toolkit (http://hannonlab.cshl.edu/fastx_toolkit/). The Illumina adapter sequences from the raw reads and the poor-quality reads from the dataset were removed (parameters –q 30 and–length 36) with Trim Galore v 0.4.2 (www.bioinformatics.babraham.ac.uk/projects/trim_galore). The resulting curated reads were aligned to the human reference genome (hg38_Ensembl) or *T. gondii* genome (TGGT1, toxodb.org v42) using HISAT2 aligner. Read counts on the genome features were generated using HTSeq-count (https://htseq.readthedocs.io/en/release_0.11.1/count.html). The high-performance computing Baobab cluster at University of Geneva was used to perform all the computations. All samples were collected and processed in triplicates. Differences in gene expression were assessed using edgeR, a Bioconductor package in R (http://www.Rproject.org). Genes with at least a logFC of ±0.5 (condition tested versus the WT) and an adjusted *p* value (FDR) less than 0.05 were treated as differentially expressed, using standard statistical methods. A *t*-test was performed to obtain the *p* value and the adjusted *p* value (FDR) for each gene was calculated using the Benjamini–Hochberg method.

Generally, the combination of the logFC and FDR, represented in a 'volcano plot', is used to identify reliable and strong expression changes. The GeneIDs of the parasites (TGGT1) and the human host (HFFs), along with the fold-changes and p-values are summarized in Supplementary Data 2.

PantherDB (http://pantherdb.org/) was used to obtain the GO Terms for the up- and downregulated genes in HFFs (infected with RH parasites) grown in Pan depleted media (Supplementary Data 2, S2.2 and S2.3). To classify the GO Terms, we used the ENSEMBL gene IDs. "Homo sapiens" and "Functional classification viewed in gene list" were selected for the GO Terms and "Functional Classification viewed in graphic charts: Pie chart" was used for the charts. The distribution of the changes in the following GO Terms can be seen: Molecular Function, Biological Process, Cellular Component, Protein Class and Pathway. To observe the daughter branches of the parental GO Terms, the interactive and clickable online pie charts can be generated.

**Animal experimentation**. All animal experiments were conducted with the authorization numbers GE150-16 and GE121-19, according to the guidelines and regulations issued by the Swiss Federal Veterinary Office. No human samples were used for these experiments.

All animals were housed at the University of Geneva in room with day/night cycle of 12 h/12 h and constant ambient temperature of 22 °C and 35% humidity. Virulence assays were performed by intraperitoneal injection of 100 freshly lysed RH parasites in 7 weeks old female CD1 mice (Charles River). Mice were monitored daily and sacrificed at the onset of signs of acute infection (ruffled fur, difficulty moving, isolation). Cyst burden experiments were performed by intraperitoneal injection of 200 freshly lysed ME49 parasites into 7-week-old female B6CBAf1 mice (Janvier labs). Mice were monitored daily and sacrificed at the onset of signs of infection (ruffled fur, difficulty moving, isolation). 5 weeks post infection surviving mice were sacrificed. Brains were collected and homogenized by sequential passaging in needles with gauge diameter of 18 g, 20 g and 23 g in 1 ml sterile PBS. 10 μl of brain homogenates were counted blindly in triplicate with an optical microscope to determine cyst burden.

**Ex vivo cysts viability assay**. Brains from infected mice were collected in sterile conditions. Cysts were extruded from neurons by sequential needle-passaging as described in the "in vivo experiments" section. Cysts from brain homogenate were counted and 25, 5 and 1 cysts (all dilution performed in technical duplicate) per mouse brain were inoculated on HFF monolayer in DMEM 5%FCS, allowed to settle for 24 h and washed from cell debris. Parasites were allowed to grow for 2 weeks before fixation and crystal violet staining. Replicating parasites were detected in all inoculated wells.

**Enzyme expression and purification**. Bacterial protein expression and purification: *PBAL* coding sequence was PCR amplified from cDNA with primers 9975/9976 and cloned by Gibson assembly into a pET-modified vector containing an N-terminal maltose binding protein (MBP) and a C-terminal TEV recognition sequence followed by a His10-TwinSTREP tag. Catalytic domain of PBAL (PBALcd) was selected based on sequence homology and comparison with published crystal structures (PDB: 2X3F), PCR amplified with primers 10166/10167. PBAL was expressed in Rosetta DE3 pLysS cells. From a preculture, 1.5 l of cells were grown until reaching OD600 of 0.6, maintained on ice for 1 h and induced by addition of 0.5 mM IPTG. Protein expression was done for 15 h at 20 °C in a shaking incubator. Cells were harvested by centrifugation at $3000 \times g$ for 12 min at 4 °C. The pellet was resuspended in 60 ml of lysis buffer (50 mM Tris pH 8, 150 mM NaCl, 3 mM β-mercaptoethanol), lysed by shear forces by passing the sample twice through the Microfluidizer at 15,000 psi. Soluble material was separated from membranes and cell debris by centrifugation at $35,000 \times g$ for 35 min at 4 °C. Supernatant was syringe-filtered at 5 μm and applied to a 5 ml STREP-tactin XT Superflow (iba) column. Column was washed first with 40 ml of lysis buffer supplemented with 250 mM NaCl, then with 40 ml of lysis buffer. Protein was digested on-column usin 0.3 mg of TEV protease. Tag-cleaved protein was eluted from the column by applying 15 ml of lysis buffer. Collected protein was concentrated and applied to a Superdex 200, 10/300 size-exclusion chromatography column equilibrated in PBS. Fractions containing MBP-PBALcd were pooled and concentrated to 500 μl using AMICON 30 MWCO. Protein concentration was estimated by measuring absorbance at 250 nm using a NanoDrop (ThermoFisher Scientific).

**Culture and cell harvest for MS experiments**. *T. gondii* parasites were cultured in complete medium (DMEM, +5 % FCS) or in custom culture medium −Pan (DMEM − Pan, +5 % dialyzed FBS) supplemented with Val (Sigma, V6504), pantoate (Sigma, 16682), α-ketoisovalerate (Sigma, 198994), β-Ala (Sigma, 146064) or stable isotope labeled metabolites: $^{13}C_3/^{15}N$-β-Ala (Cambridge Isotope Laboratories, CNLM-3946) or $^{13}C_3/^{15}N$-Pan (Cambridge Isotope Laboratories, CNLM-7964). Concentrations and durations as indicated for each experiment.

For parasite harvest, excess culture medium was gently aspirated off the monolayer of host cells with freshly egressing parasites and ice-cold PBS was added to quench the metabolism. All subsequent steps were carried out on ice or at 4 °C for centrifugations. Parasites were harvested by scraping the infected host-cell

monolayers and passing host cells through 28 g needles (3x) before removing host-debris through filtration (3 μm pore size, Millipore, TSTP04700). The flow-through was collected on ice and filters were rinsed with additional ice-cold PBS to maximize the yield of parasites. The subsequent centrifugation and washing steps were carried out swiftly at 4 °C to minimize metabolic activity. Parasites were counted using a Neubauer counting chamber and $10^8$ parasites were pelleted by centrifugation ($2200 \times g$, 10 min, 4 °C) and pellets washed with ice-cold PBS (3x) to remove residual medium. The pellets were immediately processed as described below or stored at −80 °C until metabolite extraction.

**GC-MS sample preparation and analysis**. Metabolites were extracted from cell pellets through addition of 50 μl chloroform and vigorous vortexing, followed by 200 μl methanol/ultrapure water (3:1, containing 5 μM scyllo-inositol (Sigma I8132) as internal standard. Insoluble material was pelleted by centrifugation ($21,000 \times g$, 6 min, 4 °C) and the supernatant transferred to a new microtube containing 100 μl ultrapure water. The polar and apolar phases were separated through centrifugation ($21,000 \times g$, 6 min, 4 °C). The polar phase was taken off and dried (50 μl) in a glass vial insert suitable for mass spectrometry vials using a centrifugal evaporator. For GC-MS analysis, samples were methoximated by resuspending the dried extract in 20 μl pyridine (Sigma 270970) containing methoxyamine-hydrochloride (20 mg/ml; room temperature, over-night, Sigma 226904) and trimethylsilylated shortly before analysis through addition of 20 μl N,O-bis(trimethylsilyl)trifluoroacetamide with 1 % trimethylchlorosilane (BSTFA-TMCS, Cerilliant, Sigma B-023). The GC-MS instrumentation and settings (electron ionization mode) were as described in[62]. All metabolites were identified based on the retention time and mass spectra of authentic standards. 3-trimethylsylil-(TMS)-Pan qualifier ions were: $m/z$ 201, 291 and 420, for 2-TMS-Val: $m/z$ 144, 218, 246 and for 3-TMS-β-Ala: $m/z$ 174, 248 and 290 (Supplementary Fig. 4a). Previously, a detailed GC-MS study of the 3-TMS-Pan derivative investigated its fragmentation during electron ionization demonstrating that the ion $m/z$ 420 corresponds to $[M-CH_3]^+$, while ions $m/z$ 291 and $m/z$ 201 derive from complex rearrangements but contain the full backbone of the β-Ala fraction of Pan[69]. Relative concentrations of Pan were determined by quantifying abundance of the parental ion $m/z$ 201 in unlabeled samples relative to the ion $m/z$ 318 of the internal standard (scyllo-inositol) or the total ion chromatogram (TIC), as indicated. Labeling in parasite-derived Pan was determined by quantifying the isotopologue distribution for the ion $m/z$ 201. Parental ions and their isotopologues were quantified using Xcalibur (ThermoFisher Scientific), OpenChrom and Excel (Microsoft) software following correction for natural abundance[70].

**UPLC-MS sample preparation and analysis**. Parasites ($10^8$ cells) were harvested as described above. Metabolites were extracted from cell pellets by adding 200 μl acetonitrile/ultrapure water (4:1) and vortexing vigorously. Insoluble material was pelleted through centrifugation ($21,000 \times g$, 6 min, 4 °C). For reversed-phase UPLC-MS experiments the metabolite extract was sequentially dried as described above and resuspended in 50 μl ultrapure water. For HILIC UPLC-MS analyses, the metabolite extract was injected directly. For metabolite quantification experiments, 10 μM $^{13}C_3/^{15}N$-Pan (Figs. 4n and 5k) or 40 μM $^{13}C_6/^{15}N$-isoleucine (Figs. 2l, 3k, and 4m) were included in the extraction solvent as internal standard. Abundances were normalized to the intensity of the internal standard and presented relative to the control strains as indicated (normalized to relative abundance of 1). The retention time and fragmentation of each metabolite was determined using authentic standards.

*Reversed-phase UPLC-MS analyses*. were performed by parallel reaction monitoring (PRM) experiments using a ThermoFisher Scientific Q Exactive Plus coupled to an UltiMate 3000 RSLC, or by multiple reaction monitoring (MRM) experiments using either a Sciex QTRAP 3200 (Fig. 6d) or a Sciex QTRAP 6500 (Figs. 4n and 5k) coupled to an Agilent Technologies 1290 Infinity LC Series. For all above instrument setups, the UPLC column was a Cortecs T3 column (150 × 2.1 mm × 1.6 μm) equipped with its 5 mm VanGuard cartridge (Waters). Mobile phase composition and gradient as well as MS conditions are given as supplementary information (Supplementary Methods).

*HILIC UPLC-MS analyses*. were performed by parallel reaction monitoring (PRM) experiments using a ThermoFisher Scientific Q Exactive Plus coupled to an Ulti-Mate 3000 RSLC. The UPLC column was a BEH Amide column (150 × 2.1 mm × 1.7 μm) equipped with its 5 mm VanGuard cartridge (Waters). Mobile phase composition and gradient elution conditions, as well as MS parameters are given as supplementary information (Supplementary Methods).

**LC-MS data processing**. XCalibur (ThermoFisher Scientific), Skyline[71], PeakView and MultiQuant (SCIEX) software were used to process the data. Relative metabolite levels and labeling were quantified using Excel (Microsoft). Metabolite levels were determined by quantifying the area under the curve relative to the internal standard. For PRM experiments, the TIC trace was used for quantification, for FS-PRM experiments, the full scan trace was used and for MRM experiments, the first transition was selected for quantification. For labeling analyses shown in Figs. 6d or 5f, the relative levels of M0 and M4 or M7 isotopologues were determined for each

metabolite. For data shown in Figs. 6e and 7j, l, all Pan isotopologues in the full scan were measured (NEG mode) and quantified using El-Maven Software[72]. The relative isotopologue distribution is given and labeling (%) was quantified based on the isotopologue distribution following correction for natural abundance[70].

**Reporting summary**. Further information on research design is available in the Nature Research Reporting Summary linked to this article.

## Data availability

The authors declare that the data supporting the findings of this study are available within the paper and its supplementary information files. Source data are provided with this paper. RNA sequencing data is deposited on the European Nucleotide Archive (ENA) following the ENA guidelines. The files can be accessed and downloaded via the link: http://www.ebi.ac.uk/ena/data/view/PRJEB43225. RNA analysis output: https://doi.org/10.17632/zyrz6dcc85.1[73] Metabolomics datasets can be found in the Yareta (Geneva) repository following the link[74] below. Data are sorted by platform (GC-MS, LC-MS) and figure panel. All files are deposited as original files (RAW) and as generic file formats (mzML or mzXML): https://doi.org/10.26037/yareta:bvz6yrckafdrxmzgn5hpuumkue. Source data are provided with this paper.

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

## Acknowledgements

This work was funded by the Swiss National Science Foundation (CRSII3_160702), and by the European Research Council (ERC) under the European Union's Horizon 2020 research and innovation program under grant agreement no. 695596. M.L. is supported by a PhD salary award granted by the Institute of Genetics and Genomics of Geneva (IGE3). We thank Dr. Pierre-Mehdi Hammoudi for the generation of the ME49 *hxgprt-ko ku80-ko* strain, Dr. Rebecca Oppenheim for the generation of the PPCDC-Ty tagged strain, Dr. Sunil Kumar Dogga for the support with the RNAseq analysis, and Remy Visentin (Protein platform University of Geneva) for assistance in protein purification. We thank Prof. Sebastian Lourido (Whitehead Institute) for sharing the ddBFD1 strain, and Alessandro Bonavoglia for help in characterizing it. We thank Eliane Sandmeier (MZ 2.0 facility) for setting up the GC-MS and Prof. Youssef Daali (University of Geneva) for the use of his QTRAP 6500 mass spectrometer. We finally thank Prof. Erick Strauss (Stellenbosch university) for critical reading and editing of the manuscript, as well as for the continuous discussion and input on the project.

## Author contributions

D.S.F., M.L., J.K. and A.K. conceived the study; M.L., J.K., and O.V. designed, performed and interpreted the experimental work, with the support of A.K. and E.V.; D.S.F. supervised the research; M.L, J.K and D.S.F wrote the paper, with the support of O.V., A.K., and E.V.

## Competing interests

The authors declare no competing interests.
