## [Peer Review File · Nature Communications]

Pantothenate biosynthesis is critical for chronic infection by the neurotropic parasite *Toxoplasma gondii*Reviewers' Comments:

Reviewer #1:

Remarks to the Author:

This manuscript represents a tour de force analysis of the ability of the parasite *Toxoplasma gondii* to synthesize CoA from pantothenate as well as the pantothenate biosynthesis. Using downregulation and rapid protein degradation techniques, the authors found that downregulation or depletion of any of the enzymes involved in CoA biosynthesis from pantothenate resulted in a severe growth defect for the tachyzoite stage. In contrast, they found that the three key enzymes necessary for pantothenate synthesis are dispensable for the acute infection tachyzoite stage. Using ¹³C/¹⁵N-Pantothenate, they show that tachyzoites can salvage pantothenate from the host, likely due to a transport mechanism, but that this salvage is not essential for growth. *T. gondii* has a pantothenate synthesis pathway so the authors deleted enzymes in the pathway to show that it is not active in tachyzoites, but essential for the bradyzoite stage in culture. It is interesting that the pantothenate synthesis pathway knockouts have virulence defects in mice. Overall, this manuscript represents a thorough analysis of these pathways and a monumental amount of lab time. The manuscript is well-written but there are major and minor suggestions for improvement.

Major:

There is a discrepancy between the data in Fig 4C and the model in Fig 4J. It is confusing why Fig4C shows that culture in the presence of labeled-βAla shows no incorporation into Pantothenate, but the model for 4J shows βAla being taken up by bradyzoites and being incorporated into Pantothenate. The model needs to be adjusted to make the data presented.

The authors cannot conclude that pantothenate synthesis pathway knockouts have an additional defect in chronic infection establishment when there is a significant defect during acute infection. According to ToxoDB, both TGME49_257050 and TGME49_265870 are abundantly expressed during both acute and chronic infections. It looks like the parasites may be decreased in their ability to replicate in animals period and that difference is exacerbated by chronic infection. In line 413, the authors also cannot say "no further reduction of cyst numbers in *kphmt-kpr-ko* and *pbal-ko* was observed" because their numbers of mice at these late infection time points is just too small.

Minor:

Were the parasites per vacuole counted on blinded slides to prevent bias? It doesn't say whether they were blinded in the figure legend or the methods section.

While it is good to have a quantification of the number of parasites per vacuole, it would also be good to have a quantification of the plaque numbers in addition to the representative image.

The manuscript would be aided by breaking up some of the thoughts and experiments into paragraphs. For example, the first three paragraphs of the results are 2+ pages long, cover several different steps in the pathways, and should be broken up.

For figure 2, it would be good to note what the molecular weight cutoff was for the FBS dialysis so that it is clear that it did not contain pantothenate.

For lines 240-245, the authors need to make it clear that they are discussing host transcripts not *T. gondii* transcripts as their RNAseq will capture both.

For the Fig 3D legend, the authors need to explain that M0 is the normal mass and that M4 is the mass +4 with the incorporation of the stable heavy isotopes.

Reviewer #2:

Remarks to the Author:

In this manuscript Lunghi et al. examined the pantothenate synthesis pathway in order to investigate the importance and role of CoA metabolism in *Toxoplasma gondii*, the agent of human toxoplasmosis. The molecular biology of the manuscript is very comprehensive and mutants of a near full CoA synthesis pathway has been generated. Using genetic, biochemical and metabolomics approaches, they conclude that Pan synthesis is not essential during the acute life stages of the parasite, rather PAn seemed scavenged from the host. On the other hand, they claim that Pan synthesis is active and essential for the development and survival of bradyzoites, parasite life stages responsible for ecystement and chronic phase of the disease. The topic is of high interest and the results interesting and novel. However, the manuscript is currently a mix of very good or excellent science (especially molecular biology), and poorly controlled experiments, insufficient statistical analyses/consistency, and several conclusions/interpretations not fully supported (see details below). Due to the numerous flaws that the manuscript currently suffers from, it is not possible to conclude that the claims made by the authors are fully supported by the presented data. At this point, I would suggest the manuscript to be rejected and resubmitted when all issues are fixed and claims fully supported.

Figure 1:

-There is almost NEVER a parental control included in the experiments. For the auxin system there is no IAA control for the Ku80 O_sTIR parasites, nor is there a control for anhydro-tetracycline on the Tet inducible DCPK mutant. Almost nothing has been examined on the impact of IAA on toxoplasma metabolism and this control is needed. Similarly, Tetracycline has been known to impact parasite fitness at high doses (usually above 1.0ug/ml) and no control of such potentially impact has been included or tested, especially at the metabolomic level.

-In figures 1d, 1g 1k, the IFAs are poor quality. The authors claim the localization is cytosolic but the signal is barely above background. In 1d and there is no DAPI stain to eliminate the nucleus as a site of localization. Moreover, the IMC is over-saturated and it seems the parasites are sick and "melting". Is this the case? What are the parasites like after disrupting the various enzymes of the pathway? This is a glaring omission of figure 1 in which there is no phenotypic characterisation of the mutants at all. Previous recent work of the team showed a wonderful job characterizing KO mutants (such as in Kloehn et al 2020 for the ACS/ACL KO mutants). This sort of characterization is completely missing in this manuscript. Does the IMC get disrupted? Is the mitochondrion intact? Is the nucleus intact? We don't know.

-The statistics of the paper are poor to say the least. In figure 1n, the replication assay of DPCK, the statistics are done as a one-way ANOVA. Why? Two-tailed tests should always be the norm.

-Minor point: the Tati system requires a fusion promoter swap strategy which would change the promoter. If some of these genes are using the Tet-SAG1 which is a tachyzoite promoter, this would alter the ability of these parasites to convert into bradyzoites. Similarly, the SAG4 promoter is a bradyzoite marker expressed in moderate levels in tachyzoites so would affect any bradyzoite work in figure 4. However, this would be a minor point if the phenotype was similar to the other PANK and PPCDC mutants but there is no phenotypic characterization! So we cannot say for sure.

Why replication assay have been performed after 24h of IAA treatment and not after 12h to be uniform with the WB analysis?

The authors have performed replication assay after 72h of ATc treatment, which is too long and usually done after 24-36h.

Most importantly, the metabolomics of Fig1 have also been performed at 72h of ATc, which is clearly not the good timing as based on replication assay, a lot of the parasites are "abnormal" and probably dead. So it seems normal that all metabolites homeostasis are affected. We cannot conclude that this is specifically due to the loss of each enzymes of the pathway as no control has been performed on other metabolites to see whether this is due to pleiotropic effect and parasite death. Also a complementation with the products of each enzyme is required to demonstrate the the reduction are specifically due to enzyme disruption. Furthermore and this throughout the manuscript there is no

explanation on what is the normalization criteria for metabolomics analyses. This is a crucial point that needs to be mentioned and results re-interpreted based on this. Also there is no mention whether metabolic quenching was performed prior to metabolomics analyses (based on the authors expertise, I guess that this is the case but this also needs to be precised, or metabolomics re-conducted if not properly performed).

Figure 2 and 3:

-Again, as in figure 1, there is zero phenotypic characterization of the mutants KHPMT and PBAL. Do they have an egress defect? This is very important for a new mutant to know how they are defective.

- There is no replication assay of the KHPMT mutant 2b. Why did the authors choose to analyse the PBAL in fig. 2g but not KHPMT?

-For what little phenotypic characterization that has been done, there is seemingly a slight replication defect in figure 2g, but with such a small sample size (prevalent throughout the manuscript) there might be a small growth defect and this should be quantified.

-Also a problem throughout the manuscript is the poor labelling of the y axes with "metabolite labelling". The paper is complex enough already and it would make the paper much easier to read if the metabolite in question was listed in the y axis without having to delve into the figure to find the answer. But the y axis on figure 2e should have the description of the compound, not just "% metabolite labelling". Also, where is the label for natural abundance? Is it so low that it cannot be seen?

-why is figure 3h y axis listed as "Peak Height" instead of mol% or total mass etc.? Why choose this method of quantification?

Quantifying metabolites by peak height is clearly not acceptable. Use of standards and quantification is the only accepted way of conducting proper metabolomics. I hope this is just a labelling issue

-The statistics in figure 3 are especially poor with many experiments of only n=2 sample size. The bare minimum of n=3 is already very low and this is unacceptable. An experiment being difficult is not an excuse for inadequate statistics.

-Line 298-302:

"Intracellular Pan and its derived metabolites CoA and acetyl-CoA 297 remained unlabeled following the incubation with $^{13}C_3/^{15}N$ - β -Ala (Fig. 3d). In contrast, incubation 298 of the parasites in medium containing $^{13}C_3/^{15}N$ -Pan (highlighted in the cartoon Fig. 3d) resulted in 299 labeled intracellular Pan, which was readily incorporated into CoA and acetyl-CoA (Fig. 3d), 300 confirming active Pan salvage and its metabolization as described above (Fig 2d-e). In 301 consequence, Pan synthesis is inactive in *T. gondii* tachyzoites, raising the question whether 302 these enzymes are functional."

So why didn't the authors try and complement the DPCK, PPCDC and PANK mutants with exogenous Pan in figure 1? This seemed systematically done in previous publication from the team (such as with acetate in Oppenheim et al. 2015)

The authors mention in figure 3 that Pan is most likely scavenged, so why no complementation done?

-Despite suggesting that the downstream Pan enzymes for CoA synthesis are all essential in tachyzoites, there is no experiment on the complementation of Pan in tachyzoites. The authors have done replication assays over 24 hours to quantify replication using with or without pan in PBAL KO, but no such control has been done for the other enzymes in cOA synthesis PANK, DPCK, PPCDC etc..

Figure 4:

-There is no quantification of the growth defect in the plaque assays in 4a. Also, what are "tachyzoite growth conditions" (which, by the way, is not listed in the methods). Is there a different media used in figure 4i?

-What is the difference between fig. 4a and 4i? These look like exactly the same experiment. There is no description of the "tachyzoite conditions" described in the manuscript from figure 4a. Are these the same conditions in 4i? This is not specified.

-Similarly, in 4i, the ME49 strain there is no quantification of the growth in the plaque assays. Which is prevalent throughout the manuscript.

-It could already be seen in the figure 2g that the replication of the PBAL KO might have a mild

replication defect (unclear with such a low sample size...). The growth defect might just be more obvious in the ME49 mutant because it simply grows slower.

-Where is the Pan complementation in the ME49 mutants was done in Figure 2g for PBAL? The authors tried to complement the PBAL mutant with an E. coli copy of the enzyme but with no quantification we can't really see the real difference/(or not).

-The conclusion drawn from the central nervous system requires further investigation for a manuscript of this calibre. The cyst burden is an indicator of infection, can the authors rule out the possibility that some other defect has occurred that would make it more difficult for the parasite to traverse the mouse body to reach the brain? This goes back to the poor phenotypic characterization of the PBCL and KPHMT mutants in which there is no phenotypic characterization of motility, egress, invasion or the daughter segregation etc.

Do the PBAL and KHPMT mutants have motility/egress defects and thus cannot REACH the brain?

-The statistics in this figure are quite bad. There are seemingly large enough sample sizes of mouse infections in the brain, but which are all completely different (which is fine), n=25, n=9 n=7 etc. But, the authors feel that performing statistics between these pools is appropriate. In this case, they should all be on separate graphs with their respective individual WT controls in the set -up of a classic paired t test with treated (KO) and untreated (WT) control.

-Minor point, Also, have the authors investigated an in vitro system for CNS infection? Previously in one of their papers, they argued that gene essentiality might be dependent on cell type and I was wondering if they have performed this experiment in a neuronal model?

In their previous publication the authors suggest that genes might be essential in different host cell types. Have they considered such a neuronal in vitro model for the CoA pathway/bradyzoite infection?

-Also, the reduced brain cyst burden is quite mild and could just be due to a reduced fitness/metabolic efficiency that makes the travel to reach the brain more difficult. Again, if there was adequate phenotypic characterization of these mutants we might have an answer.

-Finally, the authors' model states that Valine and alanine can be used a substrates for Pan synthesis, and the authors have tested ME49 complementation with Alaine. Why have they not done so with Valine? This is required to validate their claimed model.

Reviewer #3:

Remarks to the Author:

In the manuscript entitled 'Pantothenate biosynthesis is critical for the establishment of chronic infection by the neurotropic parasite *Toxoplasma gondii*' by Lunghi et al, the authors generate and use several knockout strains and a combination of in vitro and in vivo approaches to investigate the ability of *T. gondii* to synthesize or salvage intermediates of the Pan/CoA pathway. The intent is to establish the importance of several of these biosynthetic enzymes for the fitness of the parasite that causes toxoplasmosis. The authors find that Pan can be taken up by tachyzoites, while de novo synthesis is not detected and is dispensable for growth, under the conditions tested. Conversely, Pan synthesis plays a role in *T. gondii* chronic infection. Some of the findings presented in this manuscript are well supported by the data, but some of the data sets include a low number of replicates. Furthermore, figure legends and experiments are not described in sufficient detail, and this makes the paper hard to follow. Specific comments are below.

1. In Figure 1C, top panel: the IP of Pank2-mAID-HA from the singly tagged Pank2-mAID-HA strain yields a band in the input, but not in the actual IP. The same thing is observed in the bottom panel with Pank1. Which antibody was used for the pull down in each of these cases? This information should be clearly provided in the figure or, at the very least, in the figure legend.

2. Depletion of PPCDC seems to be deleterious to *T. gondii* under the experimental conditions used, however, the conclusion that a salvage pathway does not exist would be better supported by the addition of exogenous pantoic acid to the medium, which needs to be done.

3. The position of affinity tags should be clearly indicate for all the constructs. For example, was the Ty tag in PPCDC is at the C-terminus or N-terminus? What about the tags introduced in DPCK?

4. The effect on DPCK depletion on the parasite morphology (Fig 1n, see number of abnormal parasites) is modest compared to the depletion of PanK1, PanK2 or PPCDC. The authors should comment on this result. What are the residual CoA and acetyl-CoA levels in the strains with depleted PanK1, PanK2 and PPCDC?

5. The fact that the pbal-ko parasites grown in Pan-depleted medium maintain fitness could be the result of the presence of residual Pan in the medium and/or dialyzed FBS. The authors should measure it and report it.

6. In Fig. 2h, the pbal-ko strain has significantly lower intracellular Pan levels even when Pan is exogenously added. How do the authors explain this result? Is this phenotype rescued by complementation with PBAL?

7. What is actually shown in Fig 2i? Is it the RH parental strain or pbal-ko strain? Also, in Fig 2i and 3a, the intracellular Pan levels in the + Pan conditions are much higher than in the -Pan conditions, but this is different than what shown in Fig 2h. Where is the difference among figures coming from? Along the same lines, what strain is analyzed in Fig. 4c? Is it the wild type strain or one of the 2 ko strains generated? And in this case, which one?

8. Fig. 3b. Correct the weight of valine, which is not 1117.07898

9. The observation that the catalytic domain of PBAL can synthesize Pan, yet the tachyzoites do not, is intriguing. Have the authors tried to supply labeled pantoate and b-alanine to their cultures? This would bypass potential other inactive enzymes in the pathway. Alternatively, residues outside of the expressed PBAL catalytic domain (residues 109-492) could have a regulatory role and keep the enzyme inactive under the culture conditions tested by the authors. Have the authors tried to assay dialyzed (to remove potential endogenous small molecule inhibitors) T. gondii lysates for their ability to synthesize Pan?

10. The decrease in KPHMT-KPR and PBAL shown in supplemental Fig 4b needs to be confirmed with more than 2 replicates and an actual quantification.

11. In general, the way the data are presented in Fig 4e and f is confusing and poorly described. Furthermore, some of these data were obtained from only 2-3 mice and, if the authors decide to show them, they need to be repeated on larger cohorts. Complementation of the ME49 pbal-ko strain with the cPBAL and cPS partially increases the number of cysts but does not decrease the overall survival of the mice at 30 days post-infection. Is there a known threshold above which the number of cysts become lethal? The authors should expand the discussion of these results. Also, it is unclear whether the pbal coding sequence used in these experiments encodes for the catalytic domain or for the full length enzyme. In the first case, it should be called cdPBAL for consistency with the previous designation. Second, have the authors tried to increase the expression levels of cPABL/cPS?

Minor:

Fig S1 legend, lines 756: the gel in the Fig S1 seems to be an agarose gel and the labels seem to indicate base pair, not molecular weights

Fig. 1m is missing the label to mark the panel corresponding to the addition of ATc

Reviewer's Comments:

Reviewer #1 (Remarks to the Author)

This manuscript represents a tour de force analysis of the ability of the parasite *Toxoplasma gondii* to synthesize CoA from pantothenate as well as the pantothenate biosynthesis. Using downregulation and rapid protein degradation techniques, the authors found that downregulation or depletion of any of the enzymes involved in CoA biosynthesis from pantothenate resulted in a severe growth defect for the tachyzoite stage. In contrast, they found that the three key enzymes necessary for pantothenate synthesis are dispensable for the acute infection tachyzoite stage. Using $^{13}\text{C}_3/^{15}\text{N}$ -Pantothenate, they show that tachyzoites can salvage pantothenate from the host, likely due to a transport mechanism, but that this salvage is not essential for growth. *T. gondii* has a pantothenate synthesis pathway so the authors deleted enzymes in the pathway to show that it is not active in tachyzoites, but essential for the bradyzoite stage in culture. It is interesting that the pantothenate synthesis pathway knockouts have virulence defects in mice. Overall, this manuscript represents a thorough analysis of these pathways and a monumental amount of lab time. The manuscript is well-written but there are major and minor suggestions for improvement.

Our findings in regard to the salvage of Pan indicate that it is impossible to entirely deplete in this metabolite in viable host cells (based on MS detection of Pan). Residual Pan salvage was observed

even under Pan limiting conditions, suggesting that the low level of Pan available from the host cell is sufficient to support growth in parasites lacking de novo synthesis. This does not imply that salvage is not essential but only that the salvage pathway is very efficient. Similarly, we found Pan synthesis to be inactive in *in vitro* bradyzoites but to play a crucial role during encystation *in vivo*. The essentiality of the salvage pathway could only be assessed by deleting the Pan transporter which has not been identified to date. We have revised the manuscript and hope that these results are presented more clearly now.

Major:

There is a discrepancy between the data in Fig 4C and the model in Fig 4J. It is confusing why Fig4C shows that culture in the presence of labeled- β Ala shows no incorporation into Pantothenate, but the model for 4J shows β Ala being taken up by bradyzoites and being incorporated into Pantothenate. The model needs to be adjusted to make the data presented.

It was important to hear that the model could be a source of misinterpretation. Our data suggest a difference between *in vitro* parasites (tachyzoites and bradyzoites \rightarrow no Pan synthesis) and *in vivo* bradyzoites (active Pan synthesis based on reduced cyst number in *pbal-ko* or *kphmt-kpr-ko* parasites). We have updated the model to make this discrepancy between *in vitro* parasites and *in vivo* bradyzoites clearer.

The authors cannot conclude that pantothenate synthesis pathway knockouts have an additional defect in chronic infection establishment when there is a significant defect during acute infection. According to ToxoDB, both TGME49_257050 and TGME49_265870 are abundantly expressed during both acute and chronic infections. It looks like the parasites may be decreased in their ability to replicate in animals period and that difference is exacerbated by chronic infection. In line 413, the authors also cannot say “no further reduction of cyst numbers in *kphmt-kpr-ko* and *pbal-ko* was observed” because their numbers of mice at these late infection time points is just too small.

We thank the reviewer for pointing out the difficulty to assess defects in acute vs. chronic infection. However, we would like to point out that no defect was observed during the acute infection either with *pbal-ko* or *kphmt-kpr-ko* parasites compared to RH parasites (Fig. 8b), indicating that the acute infection is not affected by the absence of the Pan synthesis enzymes. In the survival graph of ME49 infected mice, it becomes apparent, that many mice succumbed to the infection later than 10 days post infection, as late as 4 weeks following inoculation (Fig. 8g). We therefore conclude that the ME49 parental infected mice succumb to a severe chronic infection due to a very high cyst burden. The reviewer’s suggestion that the longer chronic infection might reveal a more subtle defect is a good point and is hard to address specifically. We would argue that a general *in vivo* defect would manifest in a difference in acute virulence, which was not observed (Fig. 8b).

Based on western blot data of tagged strains (Fig 7b-c) both KPHMT-KPR and PBAL protein levels are decreased upon bradyzoite differentiation *in vitro*. Yet, this is no reason to exclude their critical role during chronic infection in the mouse model. The experiment mentioned by the reviewer in line 413 has been removed from the manuscript, due to insufficient replicates which would be difficult to justify in order to obtain animal experiments authorization, and because it did not add significantly to the key findings.

Minor:

Were the parasites per vacuole counted on blinded slides to prevent bias? It doesn't say whether they were blinded in the figure legend or the methods section.

All quantifications were performed blinded, and the methods section has been updated accordingly.

While it is good to have a quantification of the number of parasites per vacuole, it would also be good to have a quantification of the plaque numbers in addition to the representative image.

Plaque size and plaques numbers are different assays. We have included a quantification of plaque size next to the representative images of all plaque assays, except for Fig. 8i, where a qualitative rather than quantitative experiment was set-up (recrudescence - yes or no). We have chosen to quantify plaque size, as this is representative of the whole lytic cycle of the parasite over the 7 days of culture (roughly 3 cycles). The number of lysis plaques represents the fitness of the parasites (number of live parasites in a population) that was used to control the number of live parasites for infection in mice.

The manuscript would be aided by breaking up some of the thoughts and experiments into paragraphs. For example, the first three paragraphs of the results are 2+ pages long, cover several different steps in the pathways, and should be broken up.

We have considerably remodeled the manuscript by increasing the numbers of figures (8 instead of 4 Figures) and expanding the text accordingly. Specifically, we followed the reviewer's advice and broke up the CoA pathway investigations into 3 Figures and 3 corresponding results sections.

For figure 2, it would be good to note what the molecular weight cutoff was for the FBS dialysis so that it is clear that it did not contain pantothenate.

We thank the reviewer for this pertinent comment, which we have also answered for Reviewer 3, point 5. The membrane pore size was 10,000 kDa, sufficient to dialyze Pan (218 Da). The absence of Pan in the custom-made medium supplemented with dialyzed FCS was also tested quantitatively by gas chromatography-mass spectrometry and the data is shown in Fig. S3.

For lines 240-245, the authors need to make it clear that they are discussing host transcripts not *T. gondii* transcripts as their RNAseq will capture both.

We have fixed this potential source of confusion by changing the text accordingly (lines 232-240, now 454-463).

For the Fig 3D legend, the authors need to explain that M0 is the normal mass and that M4 is the mass +4 with the incorporation of the stable heavy isotopes.

We have revised the figure legend for 3D (now 6c-d) for more clarity, describing M0 as the unlabeled parental ion vs. the heavy stable-isotope labeled ion (M4).

Reviewer #2 (Remarks to the Author):

In this manuscript Lunghi et al. examined the pantothenate synthesis pathway in order to investigate the importance and role of CoA metabolism in *Toxoplasma gondii*, the agent of human toxoplasmosis. The molecular biology of the manuscript is very comprehensive and mutants of a near full CoA synthesis pathway has been generated. Using genetic, biochemical and metabolomics approaches, they conclude that Pan synthesis is not essential during the acute life stages of the parasite, rather PAN seemed scavenged from the host. On the other hand, they claim that Pan synthesis is active and essential for the development and survival of bradyzoites, parasite life stages responsible for ecystement and chronic phase of the disease. The topic is of high interest and the results interesting and novel. However, the manuscript is currently a mix of very good or excellent science (especially molecular biology), and poorly controlled experiments, insufficient statistical analyses/consistency, and several conclusions/interpretations not fully supported (see details below). Due to the numerous flaws that the manuscript currently suffers from, it is not possible to conclude that the claims made by the authors are fully supported by the presented data. At this point, I would suggest the manuscript to be rejected and resubmitted when all issues are fixed and claims fully supported.

Figure 1:

-There is almost NEVER a parental control included in the experiments. For the auxin system there is no IAA control for the Ku80 OStIR parasites, nor is there a control for anhydro-tetracycline on the Tet inducible DCPK mutant. Almost nothing has been examined on the impact of IAA on toxoplasma metabolism and this control is needed. Similarly, Tetracycline has been known to impact parasite fitness at high doses (usually above 1.0ug/ml) and no control of such potentially impact has been included or tested, especially at the metabolomic level.

We are profoundly aware of the importance of including parental controls in experiments. In our view all the critical controls were present at the time of the first submission, including in the crucial metabolomics experiments. We kindly ask the reviewer to note that the parental control lines differ between the strains used (e.g., RH as control for tet-repressive promoter system experiments but TIR1 as control for mini auxin degron system experiments). For the metabolomic experiments in the previous and this submission, we have included measurements of the parental line as well as the modified strain +/- the inducer. Note that not all studies go to this length and commonly analyzes the modified strain +/- inducer (see for example Fig. 2a in Fairweather SJ *et al.*, PLoS Pathog., 2021). Some additional experiments and figure labeling were included for clarity: in Fig. 2b a Tir1 parental control was included in the western blot; in Fig. 2c and 3d a Tir1 parental was included in the immunofluorescence; a parental RH -Atc condition was added in the intracellular growth assay Fig. 4l. We did realize that the labeling of the tetracycline treatment was missing in Fig. 1m (now Fig. 4h). This has now been fixed, and additional labels (- IAA, parental) were added for clarity in the other panels of the figures.

-In figures 1d, 1g 1k, the IFAs are poor quality. The authors claim the localization is cytosolic but the signal is barely above background. In 1d and there is no DAPI stain to eliminate the nucleus as a site of localization. Moreover, the IMC is over-saturated and it seems the parasites are sick and "melting". Is this the case? What are the parasites like after disrupting the various enzymes of the pathway? This is a glaring omission of figure 1 in which there is no phenotypic characterisation of the mutants at all. Previous recent work of the team showed a wonderful job characterizing KO mutants (such as in Kloehn et al 2020 for the ACS/ACL KO mutants). This sort of characterization is completely missing in this manuscript. Does the IMC get disrupted? Is the mitochondrion intact? Is the nucleus intact? We don't know.

We thank the reviewer for pointing out the omissions. In this submission we included immunofluorescences of all the mAID-HA tagged strains compared with Tir1 parental parasites (Fig. 2c and 3d). It can be appreciated that the signal deriving from the HA tagged PanK1, PanK2 and PPCDC is weak, but well above background. Moreover, we included an immunofluorescence of PPCDC-Ty tagged parasites clearly showing that PPCDC is also present in the nucleoplasm of the parasite (Fig. 3c). We conclude that CoA synthesis occurring in the cytoplasm of the parasite.

Numerous phenotypical analyses through immunofluorescence have been added to complete the phenotyping of the generated mutants. Figures 2d, 2h, 3e and 4g show how the IMC and morphology of the parasite are disrupted upon loss of either CoA synthesis genes. In Fig. 2h all the observed deformations are shown, with parasites blocked during cell division, abnormal vacuoles, and complete loss of parasite morphology. Nuclear signal (DAPI staining) is not lost, but fragmented nuclei and polyploid cells can be clearly seen. Apicoplast and mitochondrion, main metabolic compartments of the parasite, are imaged by immunofluorescence as shown in Fig. 2i-j, 3h-i, 4j-k. No defects in the apicoplast or mitochondrion are to be seen prior to the major morphology defect in the IMC.

-The statistics of the paper are poor to say the least. In figure 1n, the replication assay of DPCK, the statistics are done as a one-way ANOVA. Why? Two-tailed tests should always be the norm.

We agree with the reviewer that the ANOVA test used for the statistical analysis in this experiment was inappropriate. In the revised manuscript, statistical comparisons of all intercellular growth assays were made by two-tailed student t-tests. We wish to highlight, however, that appropriate statistical tests were applied in all other cases and p-values reported according to scientific standard.

-Minor point: the Tati system requires a fusion promoter swap strategy which would change the promoter. If some of these genes are using the Tet-SAG1 which is a tachyzoite promoter, this would alter the ability of these parasites to convert into bradyzoites. Similarly, the SAG4 promoter is a bradyzoite marker expressed in moderate levels in tachyzoites so would affect any bradyzoite work in figure 4. However, this would be a minor point if the phenotype was similar to the other PANK and PPCDC mutants but there is no phenotypic characterization! So we cannot say for sure.

Both 7tetOp-SAG1 and the 7TetOp-Tet-SAG4 are based on minimal inactive SAG1 and SAG4 promoters, respectively. These chimeric promoters contain only a 70 bp sequence upstream of the initiation of transcription of SAG1 or SAG4 fused to 7 tet-operator sequences and are tet-responsive but are NOT tachyzoite or bradyzoite specific promoters (Soldati D and Boothroyd J, Science, 1993). In sum, stage specificity is not an issue with these minimal tet promoters.

A detailed phenotypic characterization of each mutant in the pathway is now presented in Figures 2,3 and 4.

Why replication assay have been performed after 24h of IAA treatment and not after 12h to be uniform with the WB analysis?

While samples for western blots have been collected to highlight the minimum time necessary for downregulation of the proteins of interest, the growth assays were performed to clearly show the resulting growth and morphology defect. The new western blot (Fig. 2b) shows quick down-regulation 1 hour post IAA treatment. New intracellular growth assays (Fig. 2k, 3j) have been performed following 8 hours IAA addition and 24 hours total growth informing us for the best time point to perform metabolomic analyses.

The authors have performed replication assay after 72h of ATc treatment, which is too long and usually done after 24-36h.

We appreciate the reviewer's remarks. We have revised several sections in the manuscript (Figure legend, results, and material and methods) to be clearer and more precise. We have

performed new intracellular growth assays at 32, 48 and 72 hours, informing us for the suitable time point for metabolomic analyses. In all of the intracellular growth assays the number of parasites per vacuole were counted 24 h after infection. Pretreatments with ATc led to 32, 48, and 72 hours of total ATc treatment. As by Western blot, we observed considerable residual levels of protein after 24 hours and none left after 48 hours (now Fig. 4f), we did not perform the assay at 24 hours of ATc treatment. Parasites were scored as “deformed” following the examples in Fig. 2h. The suitable duration of Atc treatment varies for each protein/construct depending on RNA and protein stability.

Most importantly, the metabolomics of Fig1 have also been performed at 72h of ATc, which is clearly not the good timing as based on replication assay, a lot of the parasites are "abnormal" and probably dead. So it seems normal that all metabolites homeostasis are affected. We cannot conclude that this is specifically due to the loss of each enzymes of the pathway as no control has been performed on other metabolites to see whether this is due to pleiotropic effect and parasite death.

The reviewer raises a critical point about the timing to perform phenotypic experiments. When characterizing mutants upon downregulation of an essential protein, it is challenging and absolutely critical to identify the best time point for analysis. That time-point must be late enough so that the cells are suffering due to loss of the protein and exhibit a metabolic phenotype, but also early enough, so that parasites are not exhibiting a general death-phenotype. The newly performed quantification of the intracellular growth / morphological defect (Fig. 4l) revealed only minor defects in parasite morphology at 32 hours ATc treatment. Based on this, we performed a new metabolomics analysis at 32 hours of ATc treatment to detect the earliest metabolomics changes (Fig. 4m). After 32 hours of down-regulation no significant changes to the CoA pool was observed, but the substrate dephospho-CoA increased to 3-fold higher levels. This reveals a block in the DPCK-reaction. Crucially, we still consider our metabolomic results at 72 hours of ATc (Fig. 4n) as valid and informative and therefore chose to include this data. Parasites at this time-point show an accumulation of the precursors Pan (2-fold) and dePCoA (14-fold) but a severe decrease in the products (CoA and acetyl-CoA). This is the metabolic phenotype expected with a blockage of DPCK in viable parasites. A general death-phenotype would instead show absence or reduction of all metabolites due to absence of enzymatic activity and leakage of cells. Although showing a strong morphological defect after 72 hours, the continued metabolic activity of these mutant parasites (metabolite accumulation) leads us to conclude that the cells are viable. Accumulation of 2 precursors and reduction of 2 products points to a specific block consistent with the enzyme's function in CoA synthesis rather than a pleiotropic effect. New metabolomics performed for the mAID-tagged strains were also performed at 8 hours IAA treatment, where our quantifications (Fig. 2l, 3k) show almost no morphological defect.

Also a complementation with the products of each enzyme is required to demonstrate the the reduction are specifically due to enzyme disruption.

The essentiality of all CoA synthesis enzymes assessed (PanK1/PanK2, PPCDC and DPCK) indicate that these metabolites cannot be salvaged under standard culture conditions. To test if excess exogenous CoA itself can be salvaged or if CoA is hydrolyzed and phospho-pantetheine can be salvaged, we performed plaque assays of PPCDC mutant supplemented with 250 μ M, 500 μ M, and 1mM CoA, pantetheine and pantothenate (Fig. 3m-l, supplementary Fig. 1d). No supplementation rescued the PPCDC loss phenotype. The same results are expected for the PanK and DPCK mutants. We chose the PPCDC mutant as the enzyme is at the critical position in the pathway for pantetheine salvage (just upstream).

Furthermore and this throughout the manuscript there is no explanation on what is the normalization criteria for metabolomics analyses. This is a crucial point that needs to be mentioned and results re-interpreted based on this.

The abundance of each metabolite was normalized to that of the internal standard added at the time of extraction ($^{13}\text{C}_3^{15}\text{N}$ -pantothenate or $^{13}\text{C}_6/^{15}\text{N}$ -isoleucine for LC-MS analyses and scyllo-inositol for GC-MS analyses), as indicated for each experiment. Equal number of parasites (10^8) were analyzed and metabolite levels are displayed relative to those in the parental control line. This is now stated clearly in the figure legend and in the material and methods section. The metabolite levels in RH -ATc were compared to those in RH +ATc, the mutant (DPCK-iKD) + and – Atc. This is also critical to point out, to address the earlier comment by reviewer 2:

Similarly, Tetracycline has been known to impact parasite fitness at high doses (usually above 1.0ug/ml) and no control of such potentially impact has been included or tested, especially at the metabolomic level.

We would like to point out that this control was included and that no effect of Atc on the level of the measured metabolites were observed in RH parasites in contrast to the dramatic changes in the ATc-inducible mutant upon ATc treatment (see Figures 4m and 4n). These relevant controls were included in all metabolomic analyses in this revised manuscript as well as in the previous submission.

Also there is no mention whether metabolic quenching was performed prior to metabolomics analyses (based on the authors expertise, I guess that this is the case but this also needs to be precised, or metabolomics re-conducted if not properly performed).

We appreciate the reviewer pointing this out. This information was missing in the material and methods section. The infected monolayer was rinsed and scraped on ice with ice-cold PBS and cells filtered, with the filters rinsed/washed with 2 volumes of ice-cold PBS. All subsequent steps were carried out on ice or at 4 °C to minimize metabolic activity while swiftly prepping the cells. Metabolites were extracted rapidly after the harvest using solvents, to denature enzymes and further reduce metabolic activity and samples were stored at -80 °C until the time of analysis. The methods section has been updated accordingly.

Figure 2 and 3:

-Again, as in figure 1, there is zero phenotypic characterization of the mutants KHPMT and PBAL. Do they have an egress defect? This is very important for a new mutant to know how they are defective.

Importantly, the plaque assay recapitulates the whole lytic cycle of the parasite. Both *kphmt-kpr-ko* and *pbal-ko* parasites showed no defect in plaque assays and the quantifications of these experiments are presented (Fig. 5e, 5i). In consequence and logically, these two mutants are not affected in any individual steps of the lytic cycle including invasion, intracellular growth and egress. Given the metabolic role of the genes under scrutiny, we opted to include the intracellular growth assay in the manuscript to confirm the absence of defect (Fig. 5j). We also performed the egress assay for all generated knock-out strains, both in RH and ME49 strains. Since egress is not immediately relevant in this study and a defect in egress would be detectable in plaque assay, we have opted to share the data with the reviewer but we do not see the pertinence to include it in the revision.

RH and ME49 *kphmt-kpr-ko* and *pbal-ko* present no induced egress defect. Indicated strains were grown for 30 hours (RH strains) and 72 hours (ME49 strains) before washing with unsupplemented DMEM (no FBS). Egress was induced for 8 minutes with 20 μ M Bippo, or DMSO control. Quantification was performed by immunofluorescence (antibodies anti GAP45, and anti GRA3). Mean, SD, and single replicates are plotted, two-sided t-test.

- There is no replication assay of the KHPMT mutant 2b. Why did the authors choose to analyse the PBAL in fig. 2g but not KHPMT?

We have filled this gap of information. Plaque assays and intracellular growth assays, all with or without Pan depletion, were performed for all mutants and are now included in this submission (Fig. 5d-e, 5h-j).

-For what little phenotypic characterization that has been done, there is seemingly a slight replication defect in figure 2g, but with such a small sample size (prevalent throughout the manuscript) there might be a small growth defect and this should be quantified.

The intracellular replication assay has been repeated, side by side with the *kphmt-kpr-ko* mutant (Fig. 5j). As in the first submission, no defect in intracellular growth was detected for parasites grown in Pan depletion conditions, or in the Pan-synthesis mutants. We expect a sample size of 3 independent biological replicates sufficient for this experiment. Please note that a minimum of 3 independent replicates has been analyzed throughout the manuscript.

-Also a problem throughout the manuscript is the poor labelling of the y axes with “metabolite labelling”. The paper is complex enough already and it would make the paper much easier to read if the metabolite in question was listed in the y axis without having to delve into the figure to find the answer. But the y axis on figure 2e should have the description of the compound, not just “% metabolite labelling”. Also, where is the label for natural abundance? Is it so low that it cannot be seen?

Following the reviewer’s comment, we have adjusted the labeling of the y axes to make the data presented clearer. Of note, the generic term “metabolite labelling” has been kept in the instances where in the same graph several metabolites are being presented. Metabolites are then indicated just below the graph. The reviewer is correct, the natural abundance M4 is too small to be detected, although in the 4th bar from the left (the M+4 is visible in the absence of labeled metabolites in the medium). The detected peak is barely above noise levels though.

-why is figure 3h y axis listed as “Peak Height” instead of mol% or total mass etc.? Why choose this method of quantification? Quantifying metabolites by peak height is clearly not acceptable. Use of standards and quantification is the only accepted way of conducting proper metabolomics. I hope this is just a labelling issue

This is an important point raised by the reviewer. We had corrected this by normalizing to an internal standard, however, the outcome was identical as in these simple enzyme activity assays (not complex mix of metabolites), the peak height was directly proportional to the relative amount (after normalization to the standard). Kindly note that for all metabolomics experiments presented, the metabolite levels were quantified relative to an internal standard and normalized to the corresponding parental strain, analyzing equal numbers of cells. Of note the data presented in this figure has been removed and replaced by evidence of PBAL enzymatic activity in *T. gondii* cells rather than recombinant enzyme (see comments to Reviewer 3).

-The statistics in figure 3 are especially poor with many experiments of only n=2 sample size. The

bare minimum of n=3 is already very low and this is unacceptable. An experiment being difficult is not an excuse for inadequate statistics.

The experiments the reviewer refers to have been substituted. In the revision all of the experiments presented have been performed in a minimum of 3 independent biological replicates. Replicates are also shown in main figures, or, when not possible, in supplementary figures and described figure legend. Of note, the experiments referred to, have been performed several times throughout the manuscript with different methods (GC-MS and LC-MS, or using different tracers for the labeling). We argue that this use of different platforms and experiments, all of which come to the same conclusion make the data presented here especially convincing and robust. We have deleted independently two enzymes in the Pan synthesis pathway to show compellingly that Pan synthesis is dispensable under standard tachyzoite culture conditions. In addition, the experiments in Fig. 5k, 5l, 6b, 6d and 6e, while addressing different questions and using different approaches, all support the finding that Pan synthesis is inactive in *T. gondii* parasites, contrary to what has previously been reported in the literature.

-Line 298-302:

“Intracellular Pan and its derived metabolites CoA and acetyl-CoA 297 remained unlabeled following the incubation with $^{13}\text{C}_3/^{15}\text{N}$ - β -Ala (Fig. 3d). In contrast, incubation 298 of the parasites in medium containing $^{13}\text{C}_3/^{15}\text{N}$ -Pan (highlighted in the cartoon Fig. 3d) resulted in 299 labeled intracellular Pan, which was readily incorporated into CoA and acetyl-CoA (Fig. 3d), 300 confirming active Pan salvage and its metabolization as described above (Fig 2d-e). In 301 consequence, Pan synthesis is inactive in *T. gondii* tachyzoites, raising the question whether 302 these enzymes are functional.”

So why didn't the authors try and complement the DPCK, PPCDC and PANK mutants with exogenous Pan in figure 1? This seemed systematically done in previous publication from the team (such as with acetate in Oppenheim et al. 2015). The authors mention in figure 3 that Pan is most likely scavenged, so why no complementation done?

Pan is the first precursor of CoA synthesis. The inducible mutants (DPCK, PPCKC and DPCK) all function downstream of Pan salvage and in consequence their block is not expected to be rescued by Pan salvage. Moreover, Pan is present in the culture medium. Nevertheless, we have performed and included the requested Pan supplementation experiment in Fig. 3l-m, as a control for the supplementations of pantetheine and CoA in PPCDC-mAID knock-down. As predicted Pan supplementation (or CoA or pantetheine) did not rescue PPCDC knock-down.

-Despite suggesting that the downstream Pan enzymes for CoA synthesis are all essential in tachyzoites, there is no experiment on the complementation of Pan in tachyzoites. The authors have done replication assays over 24 hours to quantify replication using with or without pan in

PBAL KO, but no such control has been done for the other enzymes in cOA synthesis PANK, DPCK, PPCDC etc..

As mentioned in the response to the previous point, these enzymes function downstream of Pan salvage. Their block is not expected to be overcome through additional Pan. Note that Pan is present in all the experiments carried out when scrutinizing CoA synthesis (present by default in medium). As mentioned in the previous response, rescue with Pan, pantetheine, and CoA has been tested by plaque assay, but failed, as expected (Fig. 3l-m).

Figure 4:

-There is no quantification of the growth defect in the plaque assays in 4a. Also, what are “tachyzoite growth conditions” (which, by the way, is not listed in the methods). Is there a different media used in figure 4i?

Quantification of plaque assays has been performed, and included (Fig. 7e, 7g). We have rephrased the expression “tachyzoite growth conditions” to “standard growth conditions”, as this refers to the media and conditions used throughout the manuscript, if not stated otherwise. The methods section now has a detailed description of the culture conditions.

-What is the difference between fig. 4a and 4i? These look like exactly the same experiment. There is no description of the “tachyzoite conditions” described in the manuscript from figure 4a. Are these the same conditions in 4i? This is not specified.

The mentioned experiments are now shown in Fig. 7d and 8i. While 7d presents a plaque assay of *in vitro* cultured parasites (tachyzoites), in Fig. 8i we tested, by plaque assay, the recrudescence of *in vivo* derived parasites (bradyzoites from brain tissue cysts). We have improved the description for clarity.

-Similarly, in 4i, the ME49 strain there is no quantification of the growth in the plaque assays. Which is prevalent throughout the manuscript.

While all other plaque assays in the manuscript are now quantified, we have not done the same for this particular experiment (Fig. 8i). The readout of the experiment is purely qualitative, as the intent was to test recrudescence *in vivo* tissue cysts derived parasites (they do recrudescence). We have clarified this in the text.

-It could already be seen in the figure 2g that the replication of the PBAL KO might have a mild replication defect (unclear with such a low sample size...). The growth defect might just be more obvious in the ME49 mutant because it simply grows slower.

Although we have shown that in figure 2g (now 5j) there is no defect in intracellular growth, we thank the reviewer for pointing out this missing experiment. Intracellular growth assays have been performed for the ME49 mutants (Fig. 7h), and, as for the RH strains, no defects have been shown.

-Where is the Pan complementation in the ME49 mutants was done in Figure 2g for PBAL? The authors tried to complement the PBAL mutant with an *E. coli* copy of the enzyme but with no quantification we can't really see the real difference/(or not).

In Fig. 2g (now Fig. 5j) Pan depletion experiments were performed, in an intracellular growth assay format. As mentioned previously, the experiments have been performed with the ME49 strains (Fig. 7h). The plaque sizes of the *T. gondii* and *E. coli* PBAL complemented strains (ME49 *pbal*-ko::c*Tg*PBAL-myc and ME49 *pbal*-ko::c*Ec*PBAL-myc) have been quantified in Fig. 8f.

-The conclusion drawn from the central nervous system requires further investigation for a manuscript of this calibre. The cyst burden is an indicator of infection, can the authors rule out the possibility that some other defect has occurred that would make it more difficult for the parasite to traverse the mouse body to reach the brain? This goes back to the poor phenotypic characterization of the PBCL and KPHMT mutants in which there is no phenotypic characterization of motility, egress, invasion or the daughter segregation etc. Do the PBAL and KHPMT mutants have motility/egress defects and thus cannot REACH the brain?

In response to this, and previous comments, we have improved the phenotyping of the mentioned mutants throughout the manuscript. In detail: quantification of plaque assays (Fig. 5e, 5i, 7e, 7g), improved phenotyping of *kphmt-kpr*-ko mutants (5d-e, 5h-j, 7d-h). Moreover, we have included in the rebuttal an egress assay demonstrating that these mutants have no defect in spreading. We would like to point out that in Fig. 8b, an *in vivo* experiment performed with virulent RH parental and mutants clearly shows no defect in virulence by the *kphmt-kpr*-ko and *pbal*-ko parasites. We conclude that there is no indication of a defect of these parasites to colonize the tissues of the host, and therefore drop in cyst numbers is to be attributed to defects in the bradyzoite stage.

-The statistics in this figure are quite bad. There are seemingly large enough sample sizes of mouse infections in the brain, but which are all completely different (which is fine), n=25, n=9 n=7 etc. But, the authors feel that performing statistics between these pools is appropriate. In this case, they should all be on separate graphs with their respective individual WT controls in the set -up of a classic paired t test with treated (KO) and untreated (WT) control.

We appreciate the reviewers concern, and we have included a table (Supplementary table 4) showing all individual *in vivo* experiments. Yet for the purpose of clear representation, we decided to select and pool the experiments into one graph (Fig. 8g-h). Given the high

reproducibility of these experiments, we are confident in performing a joint statistical analysis on them. We chose to perform a non-parametric test (Mann-Whitney test) over t-tests, as the dataset is non-normal (having many outlier values), having different number of replicates, and this test being the most used for this type of data analysis in the literature.

-Minor point, Also, have the authors investigated an *in vitro* system for CNS infection? Previously in one of their papers, they argued that gene essentiality might be dependent on cell type and I was wondering if they have performed this experiment in a neuronal model? In their previous publication the authors suggest that genes might be essential in different host cell types. Have they considered such a neuronal *in vitro* model for the CoA pathway/bradyzoite infection?

We thank the reviewer for this very appropriate comment. It has been demonstrated that cystogenic parasites strains (such as our ME49) are prone to spontaneous differentiation when cultured in *in vitro* differentiated neurons and myoblasts (Lüder CG *et al.*, *Experimental Parasitology*, 1999; Ferreira-da-Silva MDF *et al.*, *International Journal of Medical Microbiology*, 2009). We tried to culture our ME49 strains in differentiated neuronal cell-lines (not shown), and myoblasts, but could detect only a small percentage of differentiation into bradyzoites (as shown by little DBA and P21 staining in the figure below), if compared to our established alkaline differentiation method. We did not further pursue alternative cell lines for parasite differentiation.

To address the point from a different angle, we make use of the recently published manuscript which identified BFD1, a transcription factor that is essential for bradyzoite differentiation, and its overexpression induces strong *in vitro* differentiation (Waldman BS *et al.*, *Cell*, 2020). We have tested the ME49 *Δ*BFD1 differentiation and have found it to be comparable to the pH-induced method in use in the lab (Fig. 7k). Pan synthesis experiments, using ¹³C/¹⁵N-β-Alanine as tracer, have shown that this novel differentiation method does not lead to synthesis of Pan without pantoate supplementation (Fig. 7l). By supplementing both precursors of Pan, namely β-Alanine and pantoate, we were able to demonstrate that *T. gondii* is capable of Pan synthesis. We speculate on the reasons for which this synthesis is critical only during encysted stage, but this will require further investigation and is beyond the scope of this study.

Comparison of ME49 parental differentiation rates in HFF vs myotubes. Parental ME49 were grown for 7 days, either in HFFs in alkaline medium (left), or in 1-week pre-differentiated KD3 myotubes (on the right), in KD3 differentiation medium (pH7.2). Parasite growth was arrested in the alkaline treated HFFs, but not in KD3, as revealed by GAP45 staining by immunofluorescence. Early bradyzoite cyst-wall marker DBA, and late bradyzoite marker P21 reveal only minimal parasite differentiation in the KD3 myotubes. White scale bar is 25 μ m.

-Also, the reduced brain cyst burden is quite mild and could just be due to a reduced fitness/metabolic efficiency that makes the travel to reach the brain more difficult. Again, if there was adequate phenotypic characterization of these mutants we might have an answer.

As mentioned before, an extended phenotypic characterization has been included in the revised manuscript. There is no data pointing at a reduced fitness of the *kphmt-kpr-ko* and *pbal-ko* mutant parasites. Note that the cyst burden is plotted on a log scale and that the defect in the burden is very substantial (11-14-fold).

-Finally, the authors' model states that Valine and alanine can be used as substrates for Pan synthesis, and the authors have tested ME49 complementation with Alanine. Why have they not done so with Valine? This is required to validate their claimed model.

As pointed out by the reviewer, we have not tested $^{13}\text{C}_5/^{15}\text{N}$ -Valine supplementation in the ME49 strain for detection of Pan synthesis. We argue that β -alanine is the better tracer as it is absent in standard medium and enters Pan synthesis at the final step. Our most recent data (Fig. 6e) demonstrates that Pan synthesis is possible by the parasite upon supplementation with the precursors β -Alanine and pantoate, but not when β -Alanine and valine are provided.

We have adapted the model to the most recent findings. While we expect Pan synthesis to occur from β -Alanine and Valine, as in other model organisms (based on *in vivo* bradyzoite phenotype of *pbal-ko* and *kphmt-kpr-ko* parasites), we could not recreate *in vitro* the conditions by which this synthesis is occurring. We have modified the model for *in vitro* vs *in vivo* situation accordingly.

Reviewer #3 (Remarks to the Author)

In the manuscript entitled 'Pantothenate biosynthesis is critical for the establishment of chronic infection by the neurotropic parasite *Toxoplasma gondii*' by Lunghi et al, the authors generate and use several knockout strains and a combination of *in vitro* and *in vivo* approaches to investigate the ability of *T. gondii* to synthesize or salvage intermediates of the Pan/CoA pathway. The intent is to establish the importance of several of these biosynthetic enzymes for the fitness of the parasite that causes toxoplasmosis. The authors find that Pan can be taken up by tachyzoites, while *de novo* synthesis is not detected and is dispensable for growth, under the conditions tested. Conversely, Pan synthesis plays a role in *T. gondii* chronic infection. Some of the findings presented in this manuscript are well supported by the data, but some of the data sets include a low number of replicates. Furthermore, figure legends and experiments are not described in sufficient detail, and this makes the paper hard to follow. Specific comments are below.

1. In Figure 1C, top panel: the IP of PanK2-mAID-HA from the singly tagged PanK2-mAID-HA strain yields a band in the input, but not in the actual IP. The same thing is observed in the bottom panel with PanK1. Which antibody was used for the pull down in each of these cases? This information should be clearly provided in the figure or, at the very least, in the figure legend.

The missing annotation has been added. Anti-Ty antibodies were used in the IP, and the mentioned samples (HA-tagged PanKs) served as control, as they do not present the Ty-tagged PanK1 or PanK2, but only the HA-tagged PanK1 or PanK2. The use of anti-Ty antibodies is now indicated in the figure and legend.

2. Depletion of PPCDC seems to be deleterious to *T. gondii* under the experimental conditions used, however, the conclusion that a salvage pathway does not exist would be better supported by the addition of exogenous pantothenate to the medium, which needs to be done.

We thank the reviewer for recommending this relevant experiment. We performed plaque assays of PPCDC knock-down, with the supplementation of CoA, pantothenate and pantothenate (Fig. 3I-m). In other organisms, pantothenate can be salvaged either as such, with the necessity of being phosphorylated by a pantothenate kinase, or as phospho-pantothenate. The latter is a product of CoA degradation, and for this, we tested supplementation of CoA to the growth media. Pantothenate was supplemented in parallel as control (and upon request of reviewer 2), as we do not expect it to

complement PPCDC knock-down. None of the supplementation experiments rescued PPCDC knock-down, demonstrating that *T. gondii* does not salvage pantoic acid, phospho-pantoic acid or CoA. Uptake of pantoic acid, on the other hand, was demonstrated in this manuscript.

3. The position of affinity tags should be clearly indicated for all the constructs. For example, was the Ty tag in PPCDC at the C-terminus or N-terminus? What about the tags introduced in DPCK?

All of the tagged mutants generated in the study have the epitope positioned at the C-terminus of the proteins of interest. This is now stated clearly in the text.

4. The effect on DPCK depletion on the parasite morphology (Fig 1n, see number of abnormal parasites) is modest compared to the depletion of PanK1, PanK2 or PPCDC. The authors should comment on this result. What are the residual CoA and acetyl-CoA levels in the strains with depleted PanK1, PanK2 and PPCDC?

This important comment has been addressed in multiple ways. Unlike PanK1, PanK2 and PPCDC, DPCK-iKD was generated with the swap of the endogenous promoter with a tetracycline inducible system. The regulation of the knock-down is slower, and for this, the strength of the phenotype is reduced at earlier time-points. We have performed one more timepoint (+32 hours ATc) for the intracellular growth assay. Moreover, we made the quantification of “deformed” parasites more stringent, taking in consideration parasite morphology and vacuole organization. Lastly, new quantitative metabolomics analysis was performed at an earlier time-point (32 hours ATc) of DPCK depletion, and early depletions of PanK1 and PPCDC, which show reduced, but not null levels of CoA and accumulation of the corresponding substrates (see completely revised Figure 2,3 and 4).

5. The fact that the *pbal*-ko parasites grown in Pan-depleted medium maintain fitness could be the result of the presence of residual Pan in the medium and/or dialyzed FBS. The authors should measure it and report it.

The customized medium (-Pan) is expected to be free of Pan according to the manufacturer's specification. The FBS was dialyzed using a membrane with pores of 10,000 Da exclusion size until glucose levels are below 10 mg/dl (Pan Biotech dialyzed FBS, P30-2102). Pantoic acid has a molecular weight (219.23 g/mol) similar to that of glucose (180.156 g/mol) and is much smaller than the exclusion size of 10,000 kDa. We have added this information concerning the exclusion size in the material and methods section.

As recommended by this reviewer, to confirm the absence of Pan in the customized medium + 5% dialyzed FBS, we analyzed the medium using gas chromatography-mass spectrometry. We were unable to detect Pan in this medium (Supplementary Fig. 3). We cannot rule out that extremely low residual levels of Pan remain present in the medium, below the level of detection. Our data suggests that *T. gondii* tachyzoites rely on Pan uptake and survive when cultured in host

cells using medium -Pan (5% dFBS). Therefore, *T. gondii* must be able to access residual Pan, which may be derived from the medium or from a residual pool within the host cells cultured under these Pan-depleted conditions.

6. In Fig. 2h, the *pbal*-ko strain has significantly lower intracellular Pan levels even when Pan is exogenously added. How do the authors explain this result? Is this phenotype rescued by complementation with PBAL?

Thanks to the reviewer's comment we realized that the order of the samples in the legend does not match the one presented in the bar graph, and was the source of this misunderstanding. This has now been fixed in the figure. In Fig. 2h (now 5l) *pbal*-ko has slightly higher Pan levels in respect to the wild-type control in -Pan medium. We are unsure about the biological significance of this. Generation of mutant parasite lines and the associated cloning can lead to the selection of slightly aberrant metabolic phenotypes. But importantly, this experiment tested whether Pan synthesis contributes to the parasite's Pan pool. The fact that PBAL-ko parasites have slightly higher Pan levels than RH, under Pan limiting conditions, is not consistent with Pan synthesis contributing to the pool. This was further scrutinized in other experiments.

7. What is actually shown in Fig 2i? Is it the RH parental strain or *pbal*-ko strain? Also, in Fig 2i and 3a, the intracellular Pan levels in the + Pan conditions are much higher than in the -Pan conditions, but this is different than what shown in Fig 2h. Where is the difference among figures coming from? Along the same lines, what strain is analyzed in Fig. 4c? Is it the wild type strain or one of the 2 ko strains generated? And in this case, which one?

As discussed in point 6, the mis-labeling of figure 2h (now 5l) led to some confusion. Upon Pan depletion, Pan levels are always reduced, but not zero, in all experiments presented in the manuscript (Fig. 5k-l and 6b). In figure 2i (now 5k) and 4c (replaced by other data) wild-type parasites Pan levels were analyzed, RH and ME49, respectively. Additional labels were included in both figures in order to make it more comprehensible.

8. Fig. 3b. Correct the weight of valine, which is not 1117.07898

We thank the reviewer for spotting this typo. The correct accurate mass of valine is: 117.07898. During the experiments performed for the revision of this manuscript, we established that Pan synthesis through PBAL can be observed in *T. gondii* when providing exogenous pantoate and β -Ala (thanks to this reviewer's suggestion – see below). Given the i) absence of β -Ala in culture medium, ii) the fact that it is readily taken up by *T. gondii*, iii) enters the Pan synthesis pathway at the final step and iv) is used for Pan synthesis through PBAL, we have opted to remove the data previously presented in Figure 3b, demonstrating the absence of Pan-labelling when providing stable isotope labeled valine. We argue that the manuscript is more concise now.

9. The observation that the catalytic domain of PBAL can synthesize Pan, yet the tachyzoites do not, is intriguing. Have the authors tried to supply labeled pantoate and β -alanine to their cultures? This would bypass potential other inactive enzymes in the pathway. Alternatively, residues outside of the expressed PBAL catalytic domain (residues 109-492) could have a regulatory role and keep the enzyme inactive under the culture conditions tested by the authors. Have the authors tried to assay dialyzed (to remove potential endogenous small molecule inhibitors) *T. gondii* lysates for their ability to synthesize Pan?

We are grateful the reviewer's suggestion to supplement pantoate in addition to β -alanine, which turned out to be instrumental. Remarkably, parasites readily salvaged pantoate when supplemented in the culture medium, and supplementation of pantoate in combination with β -alanine (labeled) resulted in Pan (labeled) synthesis by *T. gondii*. This adaptation of culture condition provided compelling evidence that PBAL is active in these parasites. We carefully probed the Pan synthesis pathway by also supplementing valine + β -alanine, and α -ketoisovalerate + β -alanine, but these conditions did not lead to labeling of Pan. The detailed analysis of the pathway including all mutants (bcat-ko, kphmt-kpr-ko and pbal-ko) in the pathway is presented in Figure 6e and was further investigated in ME49 parasites (see Figure 7). Thanks to the pertinent recommendation made by this reviewer we could demonstrate that i) *T. gondii* is capable of Pan synthesis, and ii) regulation of the synthesis is likely happening at the level of the KPHMT-KPR enzyme.

During revision, we also hoped to improve the recombinant expression and purification of the TgPBAL by testing various constructs. While expression and purity of the protein was greatly improved (Supplementary Fig. 6), this preparation presented no Pan-synthesis activity. During chromatography, it was eluting in the void volume, indicating the formation of protein aggregates, likely resulting in the inactivity of this construct. More work will be needed to optimize the heterologous expression of PBAL to be able to characterize the enzyme and screen inhibitors in the future. Given our inability to reproducibly generate active PBAL, following improved purifications, we opted to remove the data associated with the recombinant enzyme (previously Figure 3e-h). Instead, we were able to demonstrate that PBAL is active in *T. gondii* parasites thanks to the reviewer's remarks, making the data of the recombinant enzyme inessential and redundant.

10. The decrease in KPHMT-KPR and PBAL shown in supplemental Fig 4b needs to be confirmed with more than 2 replicates and an actual quantification.

A third replicate of the western blots has been performed, and the quantification of expression levels is now presented in Fig. 7c.

11. In general, the way the data are presented in Fig 4e and f is confusing and poorly described.

Furthermore, some of these data were obtained from only 2-3 mice and, if the authors decide to show them, they need to be repeated on larger cohorts. Complementation of the ME49 pbal-ko strain with the cPBAL and cPS partially increases the number of cysts but does not decrease the overall survival of the mice at 30 days post-infection. Is there a known threshold above which the number of cysts become lethal? The authors should expand the discussion of these results. Also, it is unclear whether the pbal coding sequence used in these experiments encodes for the catalytic domain or for the full length enzyme. In the first case, it should be called cdPBAL for consistency with the previous designation. Second, have the authors tried to increase the expression levels of cPABL/cPS?

We thank the reviewer for the constructive comment. We have adjusted Fig. 4 (now Fig. 8) to include only experiments derived from large cohorts of mice. There is no known threshold, after which cyst burden becomes detrimental for mice viability. We speculate that the only partial complementation is to be attributed for this lack of virulence, and have included this in the main text. Both complementation of full-length *T. gondii* PBAL and *E. coli* PBAL (also called PS) have been constructed by regulating transcription with a strong *T. gondii* promoter (tubulin), well characterized and used before. All the information is now highlighted in the text.

Minor:

Fig S1 legend, lines 756: the gel in the Fig S1 seems to be an agarose gel and the labels seem to indicate base pair, not molecular weights

This mistake has been corrected in the figure legends (“base pairs” instead of “molecular weights”).

Fig. 1m is missing the label to mark the panel corresponding to the addition of ATc

This omission has been fixed (Fig. 4h)

Reviewers' Comments:

Reviewer #1:

Remarks to the Author:

This manuscript shows that *Toxoplasma gondii* has two PanK that function as a heterodimer to phosphorylate Pan for CoA synthesis. PPCDC and DPCK are essential for CoA synthesis. Pan synthesis is dispensable in tissue culture tachy and brady conditions, but critical during chronic infection. An important point for the field as there are no effective treatments for the chronic stage of infection.

The authors have done an excellent job addressing all of the concerns from the previous reviews. Greatly improved manuscript. Much new data was added including a more careful analysis of the intracellular growth phenotype after ATc treatment and finding that at the 32 hour timepoint, there were only minor defects in the parasite morphology. They then performed metabolomics at this 32 hour timepoint. Also key new experiments were added that showed the parasites can salvage pantoate when they are also supplemented with beta-alanine.

Specifically, in response to review #2, the authors have added information to the materials and methods section, included and labeled the parental controls with and without treatments, growth assays for all mutants and strains, additional phenotypic characterization as well as high-quality immunofluorescent images for figures 2,3 and 4. All figures are more clearly labeled and all experiments are stated to be performed with a minimum of 3 independent biological replicates. Reviewer 2's concern over an egress defect was covered previously by plaque assay, but the authors have included an egress assay in the rebuttal letter to show there was no defect. Reviewer 2's concern that the minimal promoters for SAG1 and SAG4 drive some stage specificity may be shared by other readers. It would be good to include a sentence which states that stage specificity is not an issue with these minimal tet promoters and cite the reference for that. All other points for reviewer 2 were addressed.

Reviewer's comments (and author's reply)

We thank the editor and the reviewers for their detailed evaluation and constructive comments on the manuscript. We have addressed experimentally the concerns and trust that the manuscript is greatly improved in the data presented as well as in clarity. Before addressing each point raised by the reviewers below (our response in blue), we would like to emphasize that the manuscript was drastically revised with several new pieces of data included in each section and as a result we opted for not highlighting the changes made that were too numerous. The manuscript now contains 8 instead of previously 4 main figures. We have performed extensive additional phenotyping including metabolomics on 3 conditional knockdown mutants in the CoA biosynthesis pathway.

Furthermore, we have performed several supplementation experiments and found that *T. gondii* is indeed able to synthesise Pan when supplemented with pantoate and beta-alanine (a crucial experiment suggested by reviewer 3 and to whom we are grateful for the pertinent comment). The synthesis is dependent on the PBAL enzyme and provides compelling evidence that the pathway is only partially active in tachyzoites. Using other mutants and precursors, we could deduce that KPHMT-KPR is inactive and hence the limiting factor for de novo synthesis of Pan, which becomes critical during the chronic stage of toxoplasmosis.

Reviewer's Comments:

Reviewer #1 (Remarks to the Author)

This manuscript represents a tour de force analysis of the ability of the parasite *Toxoplasma gondii* to synthesize CoA from pantothenate as well as the pantothenate biosynthesis. Using downregulation and rapid protein degradation techniques, the authors found that downregulation or depletion of any of the enzymes involved in CoA biosynthesis from pantothenate resulted in a severe growth defect for the tachyzoite stage. In contrast, they found that the three key enzymes necessary for pantothenate synthesis are dispensable for the acute infection tachyzoite stage. Using $^{13}\text{C}_3/^{15}\text{N}$ -Pantothenate, they show that tachyzoites can salvage pantothenate from the host, likely due to a transport mechanism, but that this salvage is not essential for growth. *T. gondii* has a pantothenate synthesis pathway so the authors deleted enzymes in the pathway to show that it is not active in tachyzoites, but essential for the bradyzoite stage in culture. It is interesting that the pantothenate synthesis pathway knockouts have virulence defects in mice. Overall, this manuscript represents a thorough analysis of these pathways and a monumental amount of lab time. The manuscript is well-written but there are major and minor suggestions for improvement.

Our findings in regard to the salvage of Pan indicate that it is impossible to entirely deplete in this metabolite in viable host cells (based on MS detection of Pan). Residual Pan salvage was observed even under Pan limiting conditions, suggesting that the low level of Pan available from the host cell is sufficient to support growth in parasites lacking de novo synthesis. This does not imply that salvage is not essential but only that the salvage pathway is very efficient. Similarly, we found Pan synthesis to be inactive in *in vitro* bradyzoites but to play a crucial role during encystation *in vivo*. The essentiality of the salvage pathway could only be assessed by deleting the Pan transporter which has not been identified to date. We have revised the manuscript and hope that these results are presented more clearly now.

Major:

There is a discrepancy between the data in Fig 4C and the model in Fig 4J. It is confusing why Fig4C shows that culture in the presence of labeled- β Ala shows no incorporation into Pantothenate, but the model for 4J shows β Ala being taken up by bradyzoites and being incorporated into Pantothenate. The model needs to be adjusted to make the data presented.

It was important to hear that the model could be a source of misinterpretation. Our data suggest a difference between *in vitro* parasites (tachyzoites and bradyzoites \rightarrow no Pan synthesis) and *in vivo* bradyzoites (active Pan synthesis based on reduced cyst number in *pbal-ko* or *kphmt-kpr-ko* parasites). We have updated the model to make this discrepancy between *in vitro* parasites and *in vivo* bradyzoites clearer.

The authors cannot conclude that pantothenate synthesis pathway knockouts have an additional defect in chronic infection establishment when there is a significant defect during acute infection. According to ToxoDB, both TGME49_257050 and TGME49_265870 are abundantly expressed during both acute and chronic infections. It looks like the parasites may be decreased in their ability to replicate in animals period and that difference is exacerbated by chronic infection. In line 413, the authors also cannot say “no further reduction of cyst numbers in *kphmt-kpr-ko* and *pbal-ko* was observed” because their numbers of mice at these late infection time points is just too small.

We thank the reviewer for pointing out the difficulty to assess defects in acute vs. chronic infection. However, we would like to point out that no defect was observed during the acute infection either with *pbal-ko* or *kphmt-kpr-ko* parasites compared to RH parasites (Fig. 8b), indicating that the acute infection is not affected by the absence of the Pan synthesis enzymes. In the survival graph of ME49 infected mice, it becomes apparent, that many mice succumbed to the infection later than 10 days post infection, as late as 4 weeks following inoculation (Fig. 8g). We therefore conclude that the ME49 parental infected mice succumb to a severe chronic infection due to a very high cyst burden. The reviewer’s suggestion that the longer chronic infection might reveal a more subtle defect is a good point and is hard to address specifically.

We would argue that a general *in vivo* defect would manifest in a difference in acute virulence, which was not observed (Fig. 8b).

Based on western blot data of tagged strains (Fig 7b-c) both KPHMT-KPR and PBAL protein levels are decreased upon bradyzoite differentiation *in vitro*. Yet, this is no reason to exclude their critical role during chronic infection in the mouse model. The experiment mentioned by the reviewer in line 413 has been removed from the manuscript, due to insufficient replicates which would be difficult to justify in order to obtain animal experiments authorization, and because it did not add significantly to the key findings.

Minor:

Were the parasites per vacuole counted on blinded slides to prevent bias? It doesn't say whether they were blinded in the figure legend or the methods section.

All quantifications were performed blinded, and the methods section has been updated accordingly.

While it is good to have a quantification of the number of parasites per vacuole, it would also be good to have a quantification of the plaque numbers in addition to the representative image.

Plaque size and plaques numbers are different assays. We have included a quantification of plaque size next to the representative images of all plaque assays, except for Fig. 8i, where a qualitative rather than quantitative experiment was set-up (recrudescence - yes or no). We have chosen to quantify plaque size, as this is representative of the whole lytic cycle of the parasite over the 7 days of culture (roughly 3 cycles). The number of lysis plaques represents the fitness of the parasites (number of live parasites in a population) that was used to control the number of live parasites for infection in mice.

The manuscript would be aided by breaking up some of the thoughts and experiments into paragraphs. For example, the first three paragraphs of the results are 2+ pages long, cover several different steps in the pathways, and should be broken up.

We have considerably remodeled the manuscript by increasing the numbers of figures (8 instead of 4 Figures) and expending the text accordingly. Specifically, we followed the reviewer's advice and broke up the CoA pathway investigations into 3 Figures and 3 corresponding results sections.

For figure 2, it would be good to note what the molecular weight cutoff was for the FBS dialysis so that it is clear that it did not contain pantothenate.

We thank the reviewer for this pertinent comment, which we have also answered for Reviewer 3, point 5. The membrane pore size was 10,000 kDa, sufficient to dialyze Pan (218 Da). The absence of Pan in the custom-made medium supplemented with dialyzed FCS was also tested quantitatively by gas chromatography-mass spectrometry and the data is shown in Fig. S3.

For lines 240-245, the authors need to make it clear that they are discussing host transcripts not *T. gondii* transcripts as their RNAseq will capture both.

We have fixed this potential source of confusion by changing the text accordingly (lines 232-240, now 454-463).

For the Fig 3D legend, the authors need to explain that M0 is the normal mass and that M4 is the mass +4 with the incorporation of the stable heavy isotopes.

We have revised the figure legend for 3D (now 6c-d) for more clarity, describing M0 as the unlabeled parental ion vs. the heavy stable-isotope labeled ion (M4).

Reviewer #2 (Remarks to the Author):

In this manuscript Lunghi et al. examined the pantothenate synthesis pathway in order to investigate the importance and role of CoA metabolism in *Toxoplasma gondii*, the agent of human toxoplasmosis. The molecular biology of the manuscript is very comprehensive and mutants of a near full CoA synthesis pathway has been generated. Using genetic, biochemical and metabolomics approaches, they conclude that Pan synthesis is not essential during the acute life stages of the parasite, rather PAn seemed scavenged from the host. On the other hand, they claim that Pan synthesis is active and essential for the development and survival of bradyzoites, parasite life stages responsible for ecystement and chronic phase of the disease. The topic is of high interest and the results interesting and novel. However, the manuscript is currently a mix of very good or excellent science (especially molecular biology), and poorly controlled experiments, insufficient statistical analyses/consistency, and several conclusions/interpretations not fully supported (see details below). Due to the numerous flaws that the manuscript currently suffers from, it is not possible to conclude that the claims made by the authors are fully supported by the presented data. At this point, I would suggest the manuscript to be rejected and resubmitted when all issues are fixed and claims fully supported.

Figure 1:

-There is almost NEVER a parental control included in the experiments. For the auxin system

there is no IAA control for the Ku80 O₅TIR parasites, nor is there a control for anhydro-tetracycline on the Tet inducible DCPK mutant. Almost nothing has been examined on the impact of IAA on toxoplasma metabolism and this control is needed. Similarly, Tetracycline has been known to impact parasite fitness at high doses (usually above 1.0ug/ml) and no control of such potentially impact has been included or tested, especially at the metabolomic level.

We are profoundly aware of the importance of including parental controls in experiments. In our view all the critical controls were present at the time of the first submission, including in the crucial metabolomics experiments. We kindly ask the reviewer to note that the parental control lines differ between the strains used (e.g., RH as control for tet-repressive promoter system experiments but TIR1 as control for mini auxin degron system experiments). For the metabolomic experiments in the previous and this submission, we have included measurements of the parental line as well as the modified strain +/- the inducer. Note that not all studies go to this length and commonly analyzes the modified strain +/- inducer (see for example Fig. 2a in Fairweather SJ *et al.*, PLoS Pathog., 2021). Some additional experiments and figure labeling were included for clarity: in Fig. 2b a Tir1 parental control was included in the western blot; in Fig. 2c and 3d a Tir1 parental was included in the immunofluorescence; a parental RH -Atc condition was added in the intracellular growth assay Fig. 4l. We did realize that the labeling of the tetracycline treatment was missing in Fig. 1m (now Fig. 4h). This has now been fixed, and additional labels (- IAA, parental) were added for clarity in the other panels of the figures.

-In figures 1d, 1g 1k, the IFAs are poor quality. The authors claim the localization is cytosolic but the signal is barely above background. In 1d and there is no DAPI stain to eliminate the nucleus as a site of localization. Moreover, the IMC is over-saturated and it seems the parasites are sick and "melting". Is this the case? What are the parasites like after disrupting the various enzymes of the pathway? This is a glaring omission of figure 1 in which there is no phenotypic characterisation of the mutants at all. Previous recent work of the team showed a wonderful job characterizing KO mutants (such as in Kloehn et al 2020 for the ACS/ACL KO mutants). This sort of characterization is completely missing in this manuscript. Does the IMC get disrupted? Is the mitochondrion intact? Is the nucleus intact? We don't know.

We thank the reviewer for pointing out the omissions. In this submission we included immunofluorescences of all the mAID-HA tagged strains compared with Tir1 parental parasites (Fig. 2c and 3d). It can be appreciated that the signal deriving from the HA tagged PanK1, PanK2 and PPCDC is weak, but well above background. Moreover, we included an immunofluorescence of PPCDC-Ty tagged parasites clearly showing that PPCDC is also present in the nucleoplasm of the parasite (Fig. 3c). We conclude that CoA synthesis occurring in the cytoplasm of the parasite.

Numerous phenotypical analyses through immunofluorescence have been added to complete the phenotyping of the generated mutants. Figures 2d, 2h, 3e and 4g show how the IMC and

morphology of the parasite are disrupted upon loss of either CoA synthesis genes. In Fig. 2h all the observed deformations are shown, with parasites blocked during cell division, abnormal vacuoles, and complete loss of parasite morphology. Nuclear signal (DAPI staining) is not lost, but fragmented nuclei and polyploid cells can be clearly seen. Apicoplast and mitochondrion, main metabolic compartments of the parasite, are imaged by immunofluorescence as shown in Fig. 2i-j, 3h-i, 4j-k. No defects in the apicoplast or mitochondrion are to be seen prior to the major morphology defect in the IMC.

-The statistics of the paper are poor to say the least. In figure 1n, the replication assay of DPCK, the statistics are done as a one-way ANOVA. Why? Two-tailed tests should always be the norm.

We agree with the reviewer that the ANOVA test used for the statistical analysis in this experiment was inappropriate. In the revised manuscript, statistical comparisons of all intercellular growth assays were made by two-tailed student t-tests. We wish to highlight, however, that appropriate statistical tests were applied in all other cases and p-values reported according to scientific standard.

-Minor point: the Tati system requires a fusion promoter swap strategy which would change the promoter. If some of these genes are using the Tet-SAG1 which is a tachyzoite promoter, this would alter the ability of these parasites to convert into bradyzoites. Similarly, the SAG4 promoter is a bradyzoite marker expressed in moderate levels in tachyzoites so would affect any bradyzoite work in figure 4. However, this would be a minor point if the phenotype was similar to the other PANK and PPCDC mutants but there is no phenotypic characterization! So we cannot say for sure.

Both 7tetOp-SAG1 and the 7TetOp-Tet-SAG4 are based on minimal inactive SAG1 and SAG4 promoters, respectively. These chimeric promoters contain only a 70 bp sequence upstream of the initiation of transcription of SAG1 or SAG4 fused to 7 tet-operator sequences and are tet-responsive but are NOT tachyzoite or bradyzoite specific promoters (Soldati D and Boothroyd J, Science, 1993). In sum, stage specificity is not an issue with these minimal tet promoters.

A detailed phenotypic characterization of each mutant in the pathway is now presented in Figures 2,3 and 4.

Why replication assay have been performed after 24h of IAA treatment and not after 12h to be uniform with the WB analysis?

While samples for western blots have been collected to highlight the minimum time necessary for downregulation of the proteins of interest, the growth assays were performed to clearly show the resulting growth and morphology defect. The new western blot (Fig. 2b) shows quick down-regulation 1 hour post IAA treatment. New intracellular growth assays (Fig. 2k, 3j) have

been performed following 8 hours IAA addition and 24 hours total growth informing us for the best time point to perform metabolomic analyses.

The authors have performed replication assay after 72h of ATc treatment, which is too long and usually done after 24-36h.

We appreciate the reviewer's remarks. We have revised several sections in the manuscript (Figure legend, results, and material and methods) to be clearer and more precise. We have performed new intracellular growth assays at 32, 48 and 72 hours, informing us for the suitable time point for metabolomic analyses. In all of the intracellular growth assays the number of parasites per vacuole were counted 24 h after infection. Pretreatments with ATc led to 32, 48, and 72 hours of total ATc treatment. As by Western blot, we observed considerable residual levels of protein after 24 hours and none left after 48 hours (now Fig. 4f), we did not perform the assay at 24 hours of ATc treatment. Parasites were scored as "deformed" following the examples in Fig. 2h. The suitable duration of Atc treatment varies for each protein/construct depending on RNA and protein stability.

Most importantly, the metabolomics of Fig1 have also been performed at 72h of ATc, which is clearly not the good timing as based on replication assay, a lot of the parasites are "abnormal" and probably dead. So it seems normal that all metabolites homeostasis are affected. We cannot conclude that this is specifically due to the loss of each enzymes of the pathway as no control has been performed on other metabolites to see whether this is due to pleiotropic effect and parasite death.

The reviewer raises a critical point about the timing to perform phenotypic experiments. When characterizing mutants upon downregulation of an essential protein, it is challenging and absolutely critical to identify the best time point for analysis. That time-point must be late enough so that the cells are suffering due to loss of the protein and exhibit a metabolic phenotype, but also early enough, so that parasites are not exhibiting a general death-phenotype. The newly performed quantification of the intracellular growth / morphological defect (Fig. 4l) revealed only minor defects in parasite morphology at 32 hours ATc treatment. Based on this, we performed a new metabolomics analysis at 32 hours of ATc treatment to detect the earliest metabolomics changes (Fig. 4m). After 32 hours of down-regulation no significant changes to the CoA pool was observed, but the substrate dephospho-CoA increased to 3-fold higher levels. This reveals a block in the DPCK-reaction. Crucially, we still consider our metabolomic results at 72 hours of ATc (Fig. 4n) as valid and informative and therefore chose to include this data. Parasites at this time-point show an accumulation of the precursors Pan (2-fold) and dePCoA (14-fold) but a severe decrease in the products (CoA and acetyl-CoA). This is the metabolic phenotype expected with a blockage of DPCK in viable parasites. A general death-phenotype would instead show absence or reduction of all metabolites due to absence of enzymatic activity and leakage of cells. Although showing a strong morphological defect after 72 hours, the continued metabolic activity of these mutant parasites (metabolite accumulation) leads us to conclude that the cells are viable. Accumulation of 2 precursors and reduction of 2

products points to a specific block consistent with the enzyme's function in CoA synthesis rather than a pleiotropic effect. New metabolomics performed for the mAID-tagged strains were also performed at 8 hours IAA treatment, where our quantifications (Fig. 2l, 3k) show almost no morphological defect.

Also a complementation with the products of each enzyme is required to demonstrate the the reduction are specifically due to enzyme disruption.

The essentiality of all CoA synthesis enzymes assessed (PanK1/Pank2, PPCDC and DPCK) indicate that these metabolites cannot be salvaged under standard culture conditions. To test if excess exogenous CoA itself can be salvaged or if CoA is hydrolyzed and phospho-pantetheine can be salvaged, we performed plaque assays of PPCDC mutant supplemented with 250 μ M, 500 μ M, and 1mM CoA, pantetheine and pantothenate (Fig. 3m-l, supplementary Fig. 1d). No supplementation rescued the PPCDC loss phenotype. The same results are expected for the PanK and DPCK mutants. We chose the PPCDC mutant as the enzyme is at the critical position in the pathway for pantetheine salvage (just upstream).

Furthermore and this throughout the manuscript there is no explanation on what is the normalization criteria for metabolomics analyses. This is a crucial point that needs to be mentioned and results re-interpreted based on this.

The abundance of each metabolite was normalized to that of the internal standard added at the time of extraction ($^{13}\text{C}_3\text{ }^{15}\text{N}$ -pantothenate or $^{13}\text{C}_6/^{15}\text{N}$ -isoleucine for LC-MS analyses and scyllo-inositol for GC-MS analyses), as indicated for each experiment. Equal number of parasites (10^8) were analyzed and metabolite levels are displayed relative to those in the parental control line. This is now stated clearly in the figure legend and in the material and methods section. The metabolite levels in RH -ATc were compared to those in RH +ATc, the mutant (DPCK-iKD) + and - Atc. This is also critical to point out, to address the earlier comment by reviewer 2:

Similarly, Tetracycline has been known to impact parasite fitness at high doses (usually above 1.0ug/ml) and no control of such potentially impact has been included or tested, especially at the metabolomic level.

We would like to point out that this control was included and that no effect of Atc on the level of the measured metabolites were observed in RH parasites in contrast to the dramatic changes in the ATc-inducible mutant upon ATc treatment (see Figures 4m and 4n). These relevant controls were included in all metabolomic analyses in this revised manuscript as well as in the previous submission.

Also there is no mention whether metabolic quenching was performed prior to metabolomics analyses (based on the authors expertise, I guess that this is the case but this also needs to be precised, or metabolomics re-conducted if not properly performed).

We appreciate the reviewer pointing this out. This information was missing in the material and methods section. The infected monolayer was rinsed and scraped on ice with ice-cold PBS and

cells filtered, with the filters rinsed/washed with 2 volumes of ice-cold PBS. All subsequent steps were carried out on ice or at 4 °C to minimize metabolic activity while swiftly prepping the cells. Metabolites were extracted rapidly after the harvest using solvents, to denature enzymes and further reduce metabolic activity and samples were stored at -80 °C until the time of analysis. The methods section has been updated accordingly.

Figure 2 and 3:

-Again, as in figure 1, there is zero phenotypic characterization of the mutants KHPMT and PBAL. Do they have an egress defect? This is very important for a new mutant to know how they are defective.

Importantly, the plaque assay recapitulates the whole lytic cycle of the parasite. Both *kphmt-kpr-ko* and *pbal-ko* parasites showed no defect in plaque assays and the quantifications of these experiments are presented (Fig. 5e, 5i). In consequence and logically, these two mutants are not affected in any individual steps of the lytic cycle including invasion, intracellular growth and egress. Given the metabolic role of the genes under scrutiny, we opted to include the intracellular growth assay in the manuscript to confirm the absence of defect (Fig. 5j). We also performed the egress assay for all generated knock-out strains, both in RH and ME49 strains. Since egress is not immediately relevant in this study and a defect in egress would be detectable in plaque assay, we have opted to share the data with the reviewer but we do not see the pertinence to include it in the revision.

RH and ME49 *kphmt-kpr-ko* and *pbal-ko* present no induced egress defect. Indicated strains were grown for 30 hours (RH strains) and 72 hours (ME49 strains) before washing with unsupplemented DMEM (no FBS). Egress was induced for 8 minutes with 20µM Bippo, or DMSO

control. Quantification was performed by immunofluorescence (antibodies anti GAP45, and anti GRA3). Mean, SD, and single replicates are plotted, two-sided t-test.

- There is no replication assay of the KHPMT mutant 2b. Why did the authors choose to analyse the PBAL in fig. 2g but not KHPMT?

We have filled this gap of information. Plaque assays and intracellular growth assays, all with or without Pan depletion, were performed for all mutants and are now included in this submission (Fig. 5d-e, 5h-j).

-For what little phenotypic characterization that has been done, there is seemingly a slight replication defect in figure 2g, but with such a small sample size (prevalent throughout the manuscript) there might be a small growth defect and this should be quantified.

The intracellular replication assay has been repeated, side by side with the *kphmt-kpr-ko* mutant (Fig. 5j). As in the first submission, no defect in intracellular growth was detected for parasites grown in Pan depletion conditions, or in the Pan-synthesis mutants. We expect a sample size of 3 independent biological replicates sufficient for this experiment. Please note that a minimum of 3 independent replicates has been analyzed throughout the manuscript.

-Also a problem throughout the manuscript is the poor labelling of the y axes with “metabolite labelling”. The paper is complex enough already and it would make the paper much easier to read if the metabolite in question was listed in the y axis without having to delve into the figure to find the answer. But the y axis on figure 2e should have the description of the compound, not just “% metabolite labelling”. Also, where is the label for natural abundance? Is it so low that it cannot be seen?

Following the reviewer’s comment, we have adjusted the labeling of the y axes to make the data presented clearer. Of note, the generic term “metabolite labelling” has been kept in the instances where in the same graph several metabolites are being presented. Metabolites are then indicated just below the graph. The reviewer is correct, the natural abundance M4 is too small to be detected, although in the 4th bar from the left (the M+4 is visible in the absence of labeled metabolites in the medium). The detected peak is barely above noise levels though.

-why is figure 3h y axis listed as “Peak Height” instead of mol% or total mass etc.? Why choose this method of quantification? Quantifying metabolites by peak height is clearly not acceptable. Use of standards and quantification if the only accepted way of conducting proper metabolomics. I hope this is just a labelling issue

This is an important point raised by the reviewer. We had corrected this by normalizing to an internal standard, however, the outcome was identical as in these simple enzyme activity assays (not complex mix of metabolites), the peak height was directly proportional to the relative amount (after normalization to the standard). Kindly note that for all metabolomics

experiments presented, the metabolite levels were quantified relative to an internal standard and normalized to the corresponding parental strain, analyzing equal numbers of cells. Of note the data presented in this figure has been removed and replaced by evidence of PBAL enzymatic activity in *T. gondii* cells rather than recombinant enzyme (see comments to Reviewer 3).

-The statistics in figure 3 are especially poor with many experiments of only n=2 sample size. The bare minimum of n=3 is already very low and this is unacceptable. An experiment being difficult is not an excuse for inadequate statistics.

The experiments the reviewer refers to have been substituted. In the revision all of the experiments presented have been performed in a minimum of 3 independent biological replicates. Replicates are also shown in main figures, or, when not possible, in supplementary figures and described figure legend. Of note, the experiments referred to, have been performed several times throughout the manuscript with different methods (GC-MS and LC-MS, or using different tracers for the labeling). We argue that this use of different platforms and experiments, all of which come to the same conclusion make the data presented here especially convincing and robust. We have deleted independently two enzymes in the Pan synthesis pathway to show compellingly that Pan synthesis is dispensable under standard tachyzoite culture conditions. In addition, the experiments in Fig. 5k, 5l, 6b, 6d and 6e, while addressing different questions and using different approaches, all support the finding that Pan synthesis is inactive in *T. gondii* parasites, contrary to what has previously been reported in the literature.

-Line 298-302:

“Intracellular Pan and its derived metabolites CoA and acetyl-CoA 297 remained unlabeled following the incubation with ¹³C₃/¹⁵N-β-Ala (Fig. 3d). In contrast, incubation 298 of the parasites in medium containing ¹³C₃/¹⁵N-Pan (highlighted in the cartoon Fig. 3d) resulted in 299 labeled intracellular Pan, which was readily incorporated into CoA and acetyl-CoA (Fig. 3d), 300 confirming active Pan salvage and its metabolization as described above (Fig 2d-e). In 301 consequence, Pan synthesis is inactive in *T. gondii* tachyzoites, raising the question whether 302 these enzymes are functional.”

So why didn't the authors try and complement the DPCK, PPCDC and PANK mutants with exogenous Pan in figure 1? This seemed systematically done in previous publication from the team (such as with acetate in Oppenheim et al. 2015). The authors mention in figure 3 that Pan is most likely scavenged, so why no complementation done?

Pan is the first precursor of CoA synthesis. The inducible mutants (DPCK, PPCKC and DPCK) all function downstream of Pan salvage and in consequence their block is not expected to be rescued by Pan salvage. Moreover, Pan is present in the culture medium. Nevertheless, we

have performed and included the requested Pan supplementation experiment in Fig. 3l-m, as a control for the supplementations of pantetheine and CoA in PPCDC-mAID knock-down. As predicted Pan supplementation (or CoA or pantetheine) did not rescue PPCDC knock-down.

-Despite suggesting that the downstream Pan enzymes for CoA synthesis are all essential in tachyzoites, there is no experiment on the complementation of Pan in tachyzoites. The authors have done replication assays over 24 hours to quantify replication using with or without pan in PBAL KO, but no such control has been done for the other enzymes in cOA synthesis PANK, DPCK, PPCDC etc..

As mentioned in the response to the previous point, these enzymes function downstream of Pan salvage. Their block is not expected to be overcome through additional Pan. Note that Pan is present in all the experiments carried out when scrutinizing CoA synthesis (present by default in medium). As mentioned in the previous response, rescue with Pan, pantetheine, and CoA has been tested by plaque assay, but failed, as expected (Fig. 3l-m).

Figure 4:

-There is no quantification of the growth defect in the plaque assays in 4a. Also, what are “tachyzoite growth conditions” (which, by the way, is not listed in the methods). Is there a different media used in figure 4i?

Quantification of plaque assays has been performed, and included (Fig. 7e, 7g). We have rephrased the expression “tachyzoite growth conditions” to “standard growth conditions”, as this refers to the media and conditions used throughout the manuscript, if not stated otherwise. The methods section now has a detailed description of the culture conditions.

-What is the difference between fig. 4a and 4i? These look like exactly the same experiment. There is no description of the “tachyzoite conditions” described in the manuscript from figure 4a. Are these the same conditions in 4i? This is not specified.

The mentioned experiments are now shown in Fig. 7d and 8i. While 7d presents a plaque assay of *in vitro* cultured parasites (tachyzoites), in Fig. 8i we tested, by plaque assay, the recrudescence of *in vivo* derived parasites (bradyzoites from brain tissue cysts). We have improved the description for clarity.

-Similarly, in 4i, the ME49 strain there is no quantification of the growth in the plaque assays. Which is prevalent throughout the manuscript.

While all other plaque assays in the manuscript are now quantified, we have not done the same for this particular experiment (Fig. 8i). The readout of the experiment is purely qualitative, as the intent was to test recrudescence *in vivo* tissue cysts derived parasites (they do recrudescence). We have clarified this in the text.

-It could already be seen in the figure 2g that the replication of the PBAL KO might have a mild replication defect (unclear with such a low sample size...). The growth defect might just be more obvious in the ME49 mutant because it simply grows slower.

Although we have shown that in figure 2g (now 5j) there is no defect in intracellular growth, we thank the reviewer for pointing out this missing experiment. Intracellular growth assays have been performed for the ME49 mutants (Fig. 7h), and, as for the RH strains, no defects have been shown.

-Where is the Pan complementation in the ME49 mutants was done in Figure 2g for PBAL? The authors tried to complement the PBAL mutant with an *E. coli* copy of the enzyme but with no quantification we can't really see the real difference/(or not).

In Fig. 2g (now Fig. 5j) Pan depletion experiments were performed, in an intracellular growth assay format. As mentioned previously, the experiments have been performed with the ME49 strains (Fig. 7h). The plaque sizes of the *T. gondii* and *E. coli* PBAL complemented strains (ME49 *pbal-ko::cTgPBAL-myc* and ME49 *pbal-ko::cEcPBAL-myc*) have been quantified in Fig. 8f.

-The conclusion drawn from the central nervous system requires further investigation for a manuscript of this calibre. The cyst burden is an indicator of infection, can the authors rule out the possibility that some other defect has occurred that would make it more difficult for the parasite to traverse the mouse body to reach the brain? This goes back to the poor phenotypic characterization of the PBCL and KPHMT mutants in which there is no phenotypic characterization of motility, egress, invasion or the daughter segregation etc. Do the PBAL and KHPMT mutants have motility/egress defects and thus cannot REACH the brain?

In response to this, and previous comments, we have improved the phenotyping of the mentioned mutants throughout the manuscript. In detail: quantification of plaque assays (Fig. 5e, 5i, 7e, 7g), improved phenotyping of *kphmt-kpr-ko* mutants (5d-e, 5h-j, 7d-h). Moreover, we have included in the rebuttal an egress assay demonstrating that these mutants have no defect in spreading. We would like to point out that in Fig. 8b, an *in vivo* experiment performed with virulent RH parental and mutants clearly shows no defect in virulence by the *kphmt-kpr-ko* and *pbal-ko* parasites. We conclude that there is no indication of a defect of these parasites to colonize the tissues of the host, and therefore drop in cyst numbers is to be attributed to defects in the bradyzoite stage.

-The statistics in this figure are quite bad. There are seemingly large enough sample sizes of mouse infections in the brain, but which are all completely different (which is fine), n=25, n=9 n=7 etc. But, the authors feel that performing statistics between these pools is appropriate. In this case, they should all be on separate graphs with their respective individual WT controls in the set-up of a classic paired t test with treated (KO) and untreated (WT) control.

We appreciate the reviewers concern, and we have included a table (Supplementary table 4) showing all individual *in vivo* experiments. Yet for the purpose of clear representation, we decided to select and pool the experiments into one graph (Fig. 8g-h). Given the high reproducibility of these experiments, we are confident in performing a joint statistical analysis on them. We chose to perform a non-parametric test (Mann-Whitney test) over t-tests, as the dataset is non-normal (having many outlier values), having different number of replicates, and this test being the most used for this type of data analysis in the literature.

-Minor point, Also, have the authors investigated an *in vitro* system for CNS infection? Previously in one of their papers, they argued that gene essentiality might be dependent on cell type and I was wondering if they have performed this experiment in a neuronal model? In their previous publication the authors suggest that genes might be essential in different host cell types. Have they considered such a neuronal *in vitro* model for the CoA pathway/bradyzoite infection?

We thank the reviewer for this very appropriate comment. It has been demonstrated that cystogenic parasites strains (such as our ME49) are prone to spontaneous differentiation when cultured in *in vitro* differentiated neurons and myoblasts (Lüder CG *et al.*, Experimental Parasitology, 1999; Ferreira-da-Silva MDF *et al.*, International Journal of Medical Microbiology, 2009). We tried to culture our ME49 strains in differentiated neuronal cell-lines (not shown), and myoblasts, but could detect only a small percentage of differentiation into bradyzoites (as shown by little DBA and P21 staining in the figure below), if compared to our established alkaline differentiation method. We did not further pursue alternative cell lines for parasite differentiation.

To address the point from a different angle, we make use of the recently published manuscript which identified BFD1, a transcription factor that is essential for bradyzoite differentiation, and its overexpression induces strong *in vitro* differentiation (Waldman BS *et al.*, Cell, 2020). We have tested the ME49 dBFD1 differentiation and have found it to be comparable to the pH-induced method in use in the lab (Fig. 7k). Pan synthesis experiments, using ¹³C3/¹⁵N-β-Alanine as tracer, have shown that this novel differentiation method does not lead to synthesis of Pan without pantoate supplementation (Fig. 7l). By supplementing both precursors of Pan, namely β-Alanine and pantoate, we were able to demonstrate that *T. gondii* is capable of Pan synthesis. We speculate on the reasons for which this synthesis is critical only during encysted stage, but this will require further investigation and is beyond the scope of this study.

Comparison of ME49 parental differentiation rates in HFF vs myotubes. Parental ME49 were grown for 7 days, either in HFFs in alkaline medium (left), or in 1-week pre-differentiated KD3 myotubes (on the right), in KD3 differentiation medium (pH7.2). Parasite growth was arrested in the alkaline treated HFFs, but not in KD3, as revealed by GAP45 staining by immunofluorescence. Early bradyzoite cyst-wall marker DBA, and late bradyzoite marker P21 reveal only minimal parasite differentiation in the KD3 myotubes. White scale bar is 25 μ m.

-Also, the reduced brain cyst burden is quite mild and could just be due to a reduced fitness/metabolic efficiency that makes the travel to reach the brain more difficult. Again, if there was adequate phenotypic characterization of these mutants we might have an answer.

As mentioned before, an extended phenotypic characterization has been included in the revised manuscript. There is no data pointing at a reduced fitness of the *kphmt-kpr-ko* and *pbal-ko* mutant parasites. Note that the cyst burden is plotted on a log scale and that the defect in the burden is very substantial (11-14-fold).

-Finally, the authors' model states that Valine and alanine can be used as substrates for Pan synthesis, and the authors have tested ME49 complementation with Alanine. Why have they not done so with Valine? This is required to validate their claimed model.

As pointed out by the reviewer, we have not tested $^{13}\text{C}_5/^{15}\text{N}$ -Valine supplementation in the ME49 strain for detection of Pan synthesis. We argue that β -alanine is the better tracer as it is absent in standard medium and enters Pan synthesis at the final step. Our most recent data (Fig. 6e) demonstrates that Pan synthesis is possible by the parasite upon supplementation with the precursors β -Alanine and pantoate, but not when β -Alanine and valine are provided.

We have adapted the model to the most recent findings. While we expect Pan synthesis to occur from β -Alanine and Valine, as in other model organisms (based on *in vivo* bradyzoite phenotype of *pbal-ko* and *kphmt-kpr-ko* parasites), we could not recreate *in vitro* the conditions by which this synthesis is occurring. We have modified the model for *in vitro vs in vivo* situation accordingly.

Reviewer #3 (Remarks to the Author)

In the manuscript entitled 'Pantothenate biosynthesis is critical for the establishment of chronic infection by the neurotropic parasite *Toxoplasma gondii*' by Lunghi et al, the authors generate and use several knockout strains and a combination of *in vitro* and *in vivo* approaches to investigate the ability of *T. gondii* to synthesize or salvage intermediates of the Pan/CoA pathway. The intent is to establish the importance of several of these biosynthetic enzymes for the fitness of the parasite that causes toxoplasmosis. The authors find that Pan can be taken up by tachyzoites, while *de novo* synthesis is not detected and is dispensable for growth, under the conditions tested. Conversely, Pan synthesis plays a role in *T. gondii* chronic infection. Some of the findings presented in this manuscript are well supported by the data, but some of the data sets include a low number of replicates. Furthermore, figure legends and experiments are not described in sufficient detail, and this makes the paper hard to follow. Specific comments are below.

1. In Figure 1C, top panel: the IP of PanK2-mAID-HA from the singly tagged PanK2-mAID-HA strain yields a band in the input, but not in the actual IP. The same thing is observed in the bottom panel with PanK1. Which antibody was used for the pull down in each of these cases? This information should be clearly provided in the figure or, at the very least, in the figure legend.

The missing annotation has been added. Anti-Ty antibodies were used in the IP, and the mentioned samples (HA-tagged PanKs) served as control, as they do not present the Ty-tagged PanK1 or PanK2, but only the HA-tagged PanK1 or PanK2. The use of anti-Ty antibodies is now indicated in the figure and legend.

2. Depletion of PPCDC seems to be deleterious to *T. gondii* under the experimental conditions used, however, the conclusion that a salvage pathway does not exist would be better supported by the addition of exogenous pantetheine to the medium, which needs to be done.

We thank the reviewer for recommending this relevant experiment. We performed plaque assays of PPCDC knock-down, with the supplementation of CoA, pantetheine and pantothenate (Fig. 3l-m). In other organisms, pantetheine can be salvaged either as such, with the necessity of being phosphorylated by a pantothenate kinase, or as phospho-pantetheine. The latter is a product of CoA degradation, and for this, we tested supplementation of CoA to the growth

media. Pan was supplemented in parallel as control (and upon request of reviewer 2), as we do not expect it to complement PPCDC knock-down. None of the supplementation experiments rescued PPCDC knock-down, demonstrating that *T. gondii* does not salvage pantetheine, phospho-pantetheine or CoA. Uptake of pantothenate, on the other hand, was demonstrated in this manuscript.

3. The position of affinity tags should be clearly indicate for all the constructs. For example, was the Ty tag in PPCDC is at the C-terminus or N-terminus? What about the tags introduced in DPCK?

All of the tagged mutants generated in the study have the epitope positioned the C-terminus of the proteins of interest. This is now stated clearly in the text.

4. The effect on DPCK depletion on the parasite morphology (Fig 1n, see number of abnormal parasites) is modest compared to the depletion of PanK1, PanK2 or PPCDC. The authors should comment on this result. What are the residual CoA and acetyl-CoA levels in the strains with depleted PanK1, PanK2 and PPCDC?

This important comment has been addressed in multiple ways. Unlike PanK1, PanK2 and PPCDC, DPCK-iKD was generated with the swap of the endogenous promoter with a tetracycline inducible system. The regulation of the knock-down is slower, and for this, the strength of the phenotype is reduced at earlier time-points. We have performed one more timepoint (+32 hours ATc) for the intracellular growth assay. Moreover, we made the quantification of “deformed” parasites more stringent, taking in consideration parasite morphology and vacuole organization. Lastly, new quantitative metabolomics analysis was performed at an earlier time-point (32 hours ATc) of DPCK depletion, and early depletions of PanK1 and PPCDC, which show reduced, but not null levels of CoA and accumulation of the corresponding substrates (see completely revised Figure 2,3 and 4).

5. The fact that the pbal-ko parasites grown in Pan-depleted medium maintain fitness could be the result of the presence of residual Pan in the medium and/or dialyzed FBS. The authors should measure it and report it.

The customized medium (-Pan) is expected to be free of Pan according to the manufacturer’s specification. The FBS was dialyzed using a membrane with pores of 10,000 Da exclusion size until glucose levels are below 10 mg/dl (Pan Biotech dialyzed FBS, P30-2102). Pantothenate has a molecular weight (219.23 g/mol) similar to that of glucose (180.156 g/mol) and is much smaller than the exclusion size of 10,000 kDa. We have added this information concerning the exclusion size in the material and methods section.

As recommended by this reviewer, to confirm the absence of Pan in the customized medium + 5% dialyzed FBS, we analyzed the medium using gas chromatography-mass spectrometry. We

were unable to detect Pan in this medium (Supplementary Fig. 3). We cannot rule out that extremely low residual levels of Pan remain present in the medium, below the level of detection. Our data suggests that *T. gondii* tachyzoites rely on Pan uptake and survive when cultured in host cells using medium -Pan (5% dFBS). Therefore, *T. gondii* must be able to access residual Pan, which may be derived from the medium or from a residual pool within the host cells cultured under these Pan-depleted conditions.

6. In Fig. 2h, the *pbal*-ko strain has significantly lower intracellular Pan levels even when Pan is exogenously added. How do the authors explain this result? Is this phenotype rescued by complementation with PBAL?

Thanks to the reviewer's comment we realized that the order of the samples in the legend does not match the one presented in the bar graph, and was the source of this misunderstanding. This has now been fixed in the figure. In Fig. 2h (now 5l) *pbal*-ko has slightly higher Pan levels in respect to the wild-type control in -Pan medium. We are unsure about the biological significance of this. Generation of mutant parasite lines and the associated cloning can lead to the selection of slightly aberrant metabolic phenotypes. But importantly, this experiment tested whether Pan synthesis contributes to the parasite's Pan pool. The fact that PBAL-ko parasites have slightly higher Pan levels than RH, under Pan limiting conditions, is not consistent with Pan synthesis contributing to the pool. This was further scrutinized in other experiments.

7. What is actually shown in Fig 2i? Is it the RH parental strain or *pbal*-ko strain? Also, in Fig 2i and 3a, the intracellular Pan levels in the + Pan conditions are much higher than in the -Pan conditions, but this is different than what shown in Fig 2h. Where is the difference among figures coming from? Along the same lines, what strain is analyzed in Fig. 4c? Is it the wild type strain or one of the 2 ko strains generated? And in this case, which one?

As discussed in point 6, the mis-labeling of figure 2h (now 5l) led to some confusion. Upon Pan depletion, Pan levels are always reduced, but not zero, in all experiments presented in the manuscript (Fig. 5k-l and 6b). In figure 2i (now 5k) and 4c (replaced by other data) wild-type parasites Pan levels were analyzed, RH and ME49, respectively. Additional labels were included in both figures in order to make it more comprehensible.

8. Fig. 3b. Correct the weight of valine, which is not 1117.07898

We thank the reviewer for spotting this typo. The correct accurate mass of valine is: 117.07898. During the experiments performed for the revision of this manuscript, we established that Pan synthesis through PBAL can be observed in *T. gondii* when providing exogenous pantoate and β -Ala (thanks to this reviewer's suggestion – see below). Given the i) absence of β -Ala in culture

medium, ii) the fact that it is readily taken up by *T. gondii*, iii) enters the Pan synthesis pathway at the final step and iv) is used for Pan synthesis through PBAL, we have opted to remove the data previously presented in Figure 3b, demonstrating the absence of Pan-labelling when providing stable isotope labeled valine. We argue that the manuscript is more concise now.

9. The observation that the catalytic domain of PBAL can synthesize Pan, yet the tachyzoites do not, is intriguing. Have the authors tried to supply labeled pantoate and β -alanine to their cultures? This would bypass potential other inactive enzymes in the pathway. Alternatively, residues outside of the expressed PBAL catalytic domain (residues 109-492) could have a regulatory role and keep the enzyme inactive under the culture conditions tested by the authors. Have the authors tried to assay dialyzed (to remove potential endogenous small molecule inhibitors) *T. gondii* lysates for their ability to synthesize Pan?

We are grateful the reviewer's suggestion to supplement pantoate in addition to β -alanine, which turned out to be instrumental. Remarkably, parasites readily salvaged pantoate when supplemented in the culture medium, and supplementation of pantoate in combination with β -alanine (labeled) resulted in Pan (labeled) synthesis by *T. gondii*. This adaptation of culture condition provided compelling evidence that PBAL is active in these parasites. We carefully probed the Pan synthesis pathway by also supplementing valine + β -alanine, and α -ketoisovalerate + β -alanine, but these conditions did not lead to labeling of Pan. The detailed analysis of the pathway including all mutants (*bcat*-ko, *kphmt-kpr*-ko and *pbal*-ko) in the pathway is presented in Figure 6e and was further investigated in ME49 parasites (see Figure 7). Thanks to the pertinent recommendation made by this reviewer we could demonstrate that i) *T. gondii* is capable of Pan synthesis, and ii) regulation of the synthesis is likely happening at the level of the KPHMT-KPR enzyme.

During revision, we also hoped to improve the recombinant expression and purification of the *Tg*PBAL by testing various constructs. While expression and purity of the protein was greatly improved (Supplementary Fig. 6), this preparation presented no Pan-synthesis activity. During chromatography, it was eluting in the void volume, indicating the formation of protein aggregates, likely resulting in the inactivity of this construct. More work will be needed to optimize the heterologous expression of PBAL to be able to characterize the enzyme and screen inhibitors in the future. Given our inability to reproducibly generate active PBAL, following improved purifications, we opted to remove the data associated with the recombinant enzyme (previously Figure 3e-h). Instead, we were able to demonstrate that PBAL is active in *T. gondii* parasites thanks to the reviewer's remarks, making the data of the recombinant enzyme inessential and redundant.

10. The decrease in KPHMT-KPR and PBAL shown in supplemental Fig 4b needs to be confirmed with more than 2 replicates and an actual quantification.

A third replicate of the western blots has been performed, and the quantification of expression levels is now presented in Fig. 7c.

11. In general, the way the data are presented in Fig 4e and f is confusing and poorly described. Furthermore, some of these data were obtained from only 2-3 mice and, if the authors decide to show them, they need to be repeated on larger cohorts. Complementation of the ME49 pbal-ko strain with the cPBAL and cPS partially increases the number of cysts but does not decrease the overall survival of the mice at 30 days post-infection. Is there a known threshold above which the number of cysts become lethal? The authors should expand the discussion of these results. Also, it is unclear whether the pbal coding sequence used in these experiments encodes for the catalytic domain or for the full length enzyme. In the first case, it should be called cdPBAL for consistency with the previous designation. Second, have the authors tried to increase the expression levels of cPABL/cPS?

We thank the reviewer for the constructive comment. We have adjusted Fig. 4 (now Fig. 8) to include only experiments derived from large cohorts of mice. There is no known threshold, after which cyst burden becomes detrimental for mice viability. We speculate that the only partial complementation is to be attributed for this lack of virulence, and have included this in the main text. Both complementation of full-length *T. gondii* PBAL and *E. coli* PBAL (also called PS) have been constructed by regulating transcription with a strong *T. gondii* promoter (tubulin), well characterized and used before. All the information is now highlighted in the text.

Minor:

Fig S1 legend, lines 756: the gel in the Fig S1 seems to be an agarose gel and the labels seem to indicate base pair, not molecular weights

This mistake has been corrected in the figure legends (“base pairs” instead of “molecular weights”).

Fig. 1m is missing the label to mark the panel corresponding to the addition of ATc

This omission has been fixed (Fig. 4h)

Reviewer's comment, after revision

Reviewer #1 (Remarks to the Author):

This manuscript shows that *Toxoplasma gondii* has two PanK that function as a heterodimer to phosphorylate Pan for CoA synthesis. PPCDC and DPCK are essential for CoA synthesis. Pan synthesis is dispensable in tissue culture tachy and brady conditions, but critical during chronic infection. An important point for the field as there are no effective treatments for the chronic stage of infection.

The authors have done an excellent job addressing all of the concerns from the previous reviews. Greatly improved manuscript. Much new data was added including a more careful analysis of the intracellular growth phenotype after ATc treatment and finding that at the 32 hour timepoint, there were only minor defects in the parasite morphology. They then performed metabolomics at this 32 hour timepoint. Also key new experiments were added that showed the parasites can salvage pantoate when they are also supplemented with beta-alanine.

Specifically, in response to review #2, the authors have added information to the materials and methods section, included and labeled the parental controls with and without treatments, growth assays for all mutants and strains, additional phenotypic characterization as well as high-quality immunofluorescent images for figures 2,3 and 4. All figures are more clearly labeled and all experiments are stated to be performed with a minimum of 3 independent biological replicates. Reviewer 2's concern over an egress defect was covered previously by plaque assay, but the authors have included an egress assay in the rebuttal letter to show there was no defect. Reviewer 2's concern that the minimal promoters for SAG1 and SAG4 drive some stage specificity may be shared by other readers. It would be good to include a sentence which states that stage specificity is not an issue with these minimal tet promoters and cite the reference for that. All other points for reviewer 2 were addressed.

A sentence was added in the final manuscript to clarify that minimal promoters do not drive stage specific expression of the gene of interest.